

# Variation in brachiopod microstructure and isotope geochemistry under low pH–ocean acidification–conditions

Facheng Ye[1], Hana Jurikova[2], Lucia Angiolini[1], Uwe Brand[3], Gaia Crippa[1], Daniela Henkel[2], Jürgen Laudien[4], Claas Hiebenthal[2], Danijela Šmajgl[5]

[1]Università degli Studi di Milano, Dipartimento di Scienze della Terra 'A. Desio', Milan, 20133, Italy
[2]GEOMAR Helmholtz–Zentrum für Ozeanforschung Kiel, 24148, Kiel, Germany
[3]Department of Earth Sciences, Brock University, St. Catharines, L2S 3A1, Canada
[4]Alfred-Wegener–Institut Helmholtz–Zentrum für Polar–und Meeresforschung, Bremerhaven, 27515, Germany
[5]ThermoFisher Scientific, Hanna–Kunath–Str. 11, Bremen, 28199, Germany

*Correspondence to*: Facheng Ye (facheng.ye@unimi.it)

**Abstract.** Throughout the last few decades and in the near future $CO_2$–induced ocean acidification is potentially a big threat to marine calcite-shelled animals (e.g., brachiopods, bivalves, corals and gastropods). Despite the great number of studies focusing on the effects of acidification on shell growth, metabolism, shell dissolution and shell repair, the consequences on biomineral formation remain poorly understood, and only few studies addressed contemporarily the impact of acidification on shell microstructure and geochemistry. In this study, a detailed microstructure and stable isotope geochemistry investigation was performed on nine adult brachiopod specimens of *Magellania venosa* (Dixon, 1789), grown in the natural environment as well as in controlled culturing experiments at different pH conditions (ranging 7.35 to 8.15 $\pm 0.05$) over different time intervals (214 to 335 days). Details of shell microstructural features, such as thickness of the primary layer, density and size of endopunctae and morphology of the basic structural unit of the secondary layer were analysed using scanning electron microscopy (SEM). Stable isotope compositions ($\delta^{13}C$ and $\delta^{18}O$) were tested from the secondary shell layer along shell ontogenetic increments in both dorsal and ventral valves. Based on our comprehensive dataset, we observed that, under low pH conditions, *M. venosa* produced a more organic–rich shell with higher density of and larger endopunctae, and smaller secondary layer fibres, when subjected to about one year of culturing. Also, increasingly negative $\delta^{13}C$ and $\delta^{18}O$ values are recorded by the shell produced during culturing and are related to the $CO_2$–source in the culture setup. Both the microstructural changes and the stable isotope results are similar to observations on brachiopods from the fossil record and strongly support the value of brachiopods as robust archives of proxies for studying ocean acidification events in the geologic past.

**Key words:**

*Magellania venosa*, biomineral, $\delta^{13}C$, $\delta^{18}O$, primary layer, secondary layer, endopunctae, scanning electron microscopy, culturing



# 1 Introduction

Since the industrial revolution the surface ocean pH has dropped by 0.1 units and will probably drop another 0.3–0.5 units by 2100 (Caldeira and Wickett, 2005; Orr et al., 2005; IPCC, 2013). This is due to the increasing amount of atmospheric carbon dioxide ($CO_2$) absorbed by the ocean that extensively affects sea water carbonate chemistry (e.g., Caldeira and Wickett, 2003, 2005; Feely et al., 2004). Increased concentrations of anthropogenic $CO_2$ are reflected in an elevated concentration of hydrogen ions, which lowers the pH and the available carbonate ions (Orr et al., 2005). Effects on marine organisms is of great scientific interest, both for understanding the geological past and for the consequences in the immediate future (e.g., Ries et al., 2009), as the decrease in calcium carbonate saturation potentially threatens marine organisms forming biogenic calcium carbonate (e.g., Orr et al., 2005; Guinott et al., 2006; Jantzen et al., 2013a, b; McCulloch et al., 2012). This applies to calcium carbonate shell–forming species, such as brachiopods and mollusks, because they are considered excellent archives documenting how changes in environmental conditions can affect marine organisms (e.g., Kurihara. 2008; Comeau et al., 2009; Watson et al., 2012, Hahn et al., 2012, 2014; Cross et al., 2015, 2016, 2018; Crippa et al., 2016a; Milano et al., 2016; Garbelli et al., 2017; Jurikova et al., in review).

Recently, several experiments were performed to investigate if a change of seawater pH may affect growth rate, shell repair and oxygen consumption of calcifying organisms, and how they respond to ocean acidification (Supplementary Table 1). However, despite the great number of studies, the consequences on biomineral formation remain not well understood, as most studies focused mainly on growth, metabolic rates, shell dissolution and shell repair (Supplementary Table 1, and references therein). Only a few studies deal with the effect of acidification on microstructure (Beniash et al., 2010; Hahn et al., 2012; Stemmer et al., 2013; Fitzer et al., 2014a, b; Milano et al., 2016), and all of them focused on bivalves and show that neither microstructure, nor shell hardness seem to be affected by seawater pH.

The few studies that examined brachiopods or brachiopod shells suggest that the latter suffered increased dissolution under lower seawater pH conditions, whereas the organism either exhibited no changes, or an increase in shell density [calculated as dry mass of the shell (g)/shell volume ($cm^3$)], but otherwise no changes in shell morphology and trace chemistry (Table 1). Overall, there appears to be little to no effect on brachiopod morphology or chemistry with lower seawater pH (Cross et al., 2015, 2016, 2018).

Table 1. Culturing, dissolution experiments and natural variation on several brachiopod species and shells.

| Species N (number of sample) | Growth Parameters | Shell repair/Microstructure/Oxygen consumption/Dissolution of shell/Microstructure | Method & Material | Environment/conditions T=Temperature (℃) S=Salinity (PSU) $pCO_2$ (µatm) | Duration of experiment | Source |
|---|---|---|---|---|---|---|
| *Calloria inconspicua* (Sowerby, 1846) N = 123 | 1) >3 mm in length undamaged individuals were not affected by lower pH; 2) <3 mm in length undamaged individuals grew faster | Not affected by lower pH (>80% of all damaged individuals repaired after 12 weeks) | Culture experiment | a) pH 8.16, T 16.5, S 33.9, $pCO_2$ 465, Ω calcite 3.5 b) pH 7.79, T 16.9, S 33.9, $pCO_2$ 1130, Ω calcite 1.6 c) pH 7.62, T 16.6, S 33.9, $pCO_2$ 1536, Ω calcite 1.3 | 12 weeks | Cross et al., 2016 |





| | | | | | |
|---|---|---|---|---|---|
| | | at pH 7.62 than the control conditions | | | |
| *Calloria inconspicua* (Sowerby, 1846) N = 389 (adults) | | Punctae width decreased by 8.26%, shell density increased by 3.43%, no change in shell morphology, punctae density, shell thickness, and shell elemental composition (Ca, Mg, Na, Sr and P) | One specimen collected every decade from one locality | Last two decades pH reduced 0.1 unit Temperature varied from 10.7–13.0 ℃ $pCO_2$ varied from 320–400 Salinity and Ω of calcite not provided | 120–year record | Cross et al., 2018 |
| *Liothyrella uva* (Broderip, 1833) N = 156 | Not affected by lower pH | Not affected by either low pH conditions or temperature. (>83% of individuals repaired after 7 months) | Culture experiment | a) pH 7.98, T -0.3, S 35, $pCO_2$ 417, Ω calcite 1.20 b) pH 8.05, T 1.7, S 35, $pCO_2$ 365, Ω calcite 1.49 c) pH 7.75, T 1.9, S 35, $pCO_2$ 725, Ω calcite 0.78 d) pH 7.54, T 2.2, S 35, $pCO_2$ 1221, Ω calcite 0.50 | 7 months | Cross et al., 2015 |
| *Liothyrella uva* (Broderip, 1833) $N_{post-mortem}$ = 5 | Not applicable | Higher dissolution in gastropods and brachiopods at lower pH after 14 days | Empty shells | a) pH 7.4, T 4, S 35, Ω calcite 0.74 b) pH 8.2, T 4, S 35, Ω calcite 4.22 $pCO_2$ Not provided | 14 to 63 days | McClintock et al., 2009 |

Brachiopods possess a low–magnesium calcite shell, which should be more resistant to elevated $pCO_2$ compared to the more soluble forms of $CaCO_3$, aragonite and high-Mg calcite (Morse et al., 2007). The shell microstructure of Rhynchonelliformean brachiopods has been used as a powerful tool to understand the biomineral response to modern global

acidification and similar events in the past (Payne and Clapham, 2012; Cross et al., 2015, 2016; Garbelli et al., 2017). A comprehensive study focusing on fossil brachiopods during the end-Permian extinction showed that brachiopods tend to produce shells with higher organic components during ocean acidification events (Garbelli et al., 2017).

Here, the microstructure and stable isotope geochemistry are described of the shells of adult brachiopod specimens of the cold-temperate brachiopod species *M. venosa* (Dixon, 1789) are described. The organisms grew in the natural environment

and in culture under different pH conditions. *M. venosa* represents the largest recent brachiopod species, and locally may be abundant (Försterra et al., 2008), and it has the highest growth rate recorded for recent brachiopods (Baumgarten et al., 2014). Its low-magnesium calcite shell consists of a microgranular primary layer and a fibrous secondary layer (Smirnova et al., 1991; Baumgarten et al., 2014; Romanin et al., 2018; Casella et al. 2018) crossed by perforations, the endopunctae.

Since little is known about morphological and geochemical responses to increased ocean acidification in brachiopods (cf,

Table 1), the main goal of this study is to document any changes in this highly important archival marine organism. It will be described if and how shell microstructural features such as the primary layer thickness, density of endopunctae and fibre morphology, and their stable carbon ($\delta^{13}$C) and oxygen ($\delta^{18}$O) isotope compositions respond to low seawater pH conditions.



## 2 Materials and methods

### 2.1 Brachiopod samples and culturing set–up

A thorough description of the brachiopod sampling and culturing is provided in Jurikova et al. (in review), but an abbreviated version here is provided. Nine adult individuals of *M. venosa* (Dixon, 1789) were chosen for microstructure

5  investigation and an evaluation of their $\delta^{13}C$ and $\delta^{18}O$ values (Table 2). All specimens were collected by scientific SCUBA divers alive from appr. 20 m water depth of Comau Fjord (Chile) at different localities (Figure 1). Specimens #158 and #223 did not experience any treatment after collection from Comau Fjord. All other specimens, #43 ($pH_3$), #63 ($pH_4$), #8004 ($pH_0$), #8005 ($pH_0$), #9004 ($pH_1$ and $pH_2$), #9005 ($pH_1$ and $pH_2$) and #9006 ($pH_1$ and $pH_2$), were cultured under different pH conditions (Table 2 and Table 3) at either AWI in Bremerhaven or GEOMAR (at KIMOCC–Kiel Marine Organisms Culture

10  Centre) in Kiel, Germany.

Table 2. Specimens of *M. venosa* sampled from Comau Fjord, Chile, and natural and experimental culturing conditions.

| Sample ID | Sample locality at Comau Fjord (Chile)[①] | Sample seawater conditions[②] | Date of collection | Length of ventral valve (mm) | Duration of experiment | Experimental conditions |
|---|---|---|---|---|---|---|
| #43 | Lilliguapi | pH: ~7.9 T: ~13 S: ~32 D: 20 | Feb. 2012 | 37 | 214 days[③] | $pCO_2$: 1391, pH: 7.66 ±0.04 T: 11.62 ±0.54, S: 32.58 $\Omega$cal: 1.97 |
| #63 | Lilliguapi | pH: ~7.9 T: ~13 S: ~32 D: 20 | Feb. 2012 | 23 | 214 days[③] | $pCO_2$: 2611, pH: 7.44 ±0.08 T: 11.69 ±0.45, S: 32.65 $\Omega$cal: 1.37 |
| #158 | Huinay Dock | pH: ~7.9 T: ~13 S: ~32 D: 20 | Dec. 2011 | 36 | no | |
| #223 | Cahuelmó | pH: ~7.9 T: ~13 S: ~32 D: 23 | Feb. 2012 | 30 | no | |
| #8004 | Comau Fjord | pH: ~7.9 T: ~13 S: ~32 D: 21 | Apr. 2016 | 31 | 335 days[④] | $pCO_2$: 600 ,pH: 8.00–8.15 ± 0.05 T: ~10, S: 30, $\Omega$cal: 2.0–3.5 |
| #8005 | Comau Fjord | pH: ~7.9 T: ~13 S: ~32 D: 21 | Apr. 2016 | 46 | 335 days[④] | $pCO_2$: 600, pH: 8.00–8.15 ± 0.05 T: ~10, S: 30, $\Omega$cal: 2.0–3.5 |




| #9004 | Comau Fjord | pH: ~7.9 T: ~13 S: ~32 D: 21 | Apr. 2016 | 41 | 335 days[4] | $p$CO$_2$: 2000–4000[5] pH: 7.60 ±0.05 to 7.35 ± 0.054 T: ~10, S: 30, Ωcal: 0.6–1.1 |
| #9005 | Comau Fjord | pH: ~7.9 T: ~13 S: ~32 D: 21 | Apr. 2016 | 25 | 335 days[4] | $p$CO$_2$: 2000–4000[5] pH: 7.60 ±0.05 to 7.35 ± 0.054 T: ~10, S: 30, Ωcal: 0.6–1.1 |
| #9006 | Comau Fjord | pH: ~7.9 T: ~13 S: ~32 D: 21 | Apr. 2016 | 43 | 335 days[4] | $p$CO$_2$: 2000–4000[5] pH: 7.60 ±0.05 to 7.35 ± 0.054 T: ~10, S: 30, Ωcal: 0.6–1.1 |

Note: D: Depth (m), T: temperature (°C), S: salinity (PSU – practical salinity units), $p$CO$_2$ (µatm).

[1] Cahuelmó 42°15'23" S, 72°26'42" W, Cross–Huinay 42°23'28" S, 72°27'27" W, Jetty (Huinay Dock) 42°22'47" S, 72°24'56" W, Lilliguapy 42°9'43" S, 72°35'55" W, samples #8004, #8005, #9004, #9005, #9006 were harvested from three sites in Comau Fjord (Cross–Huinay, Jetty, and Liliguapy), Chilean Patagonia

[2] Reference: Laudien et al. (2014) and Jantzen et al. (2017)

[3] Culture experiments conducted at the Alfred–Wegener–Institut Helmholtz–Zentrum für Polar–und Meeresforschung, Bremerhaven, Germany

[4] Culture experiments conducted at GEOMAR Helmholtz–Zentrum für Ozeanforschung Kiel, Germany (Jurikova et al., in review)

[5] CO$_2$ concentration was changed during the experiment: from 4 August 2016 to 18 April 2017 at 2000 µatm and from 18 April 2017 till 5 July 2017 at 4000 µatm

Table 3. Culture and sensor systems for *M. venosa* specimens (#43, #63, #8004, #8005, #9004, #9005 and #9006). Operated under controlled experimental settings in a climate control laboratory at the Alfred–Wegener–Institut Helmholtz–Zentrum

für Polar–und Meeresforschung, Bremerhaven, Germany and at GEOMAR Helmholtz–Zentrum für Ozeanforschung Kiel, Germany.

|  | Culture system at AWI | Automated sensor Systems at AWI | Culture system at GEOMAR | Automated sensor Systems at GEOMAR |
|---|---|---|---|---|
|  | Aquarium (150 L/each pH treatment) |  | Aquarium (150 L/each pH treatment) |  |
|  | Supplied from a reservoir tank (twice a week 20 % water was replaced) |  | Supplied from a reservoir tank (twice a month 10 % water was replaced) |  |
| Temperature | Controlled in temperature |  | Controlled using | Temperature |




| | constant room | | heaters or coolers | Sensor Pond |
|---|---|---|---|---|
| $p$CO$_2$ | Bubbling of CO$_2$ pH 7.66 ± 0.04, pH 7.44 ± 0.08 | COMPORT, Dennerle, Vinningen; IKS aquastar Aquarium computer V2.xx with Aquapilot 2011 | Bubbling of CO$_2$ enriched air | CONTROS HydroC® underwater CO$_2$ sensor |
| Salinity | Mixing Reef commercial sea–salt (until October: Aqua Medic, Bissendorf, Germany, thereafter Dupla Marin Reef Salt, Dohse Aquaristik, Grafschaft–Gelsdorf, Germany) with deionized water (Atkinson and Bingman, 1998) | Conductivity Electrode | Mixing Tropic Marin Pro–Reef commercial sea-salt with deionized water (Atkinson and Bingman, 1998) | Conductivity Electrode |
| Filtering | Biofilter, protein skimmer and UV sterilizer | | Biofilter, protein skimmer and UV sterilizer | |
| Food | Regularly fed (typically 5 times per week) with Dupla Rin, Coral Food, Reef Pearls 5–200 μm, alive *Thalassiosira weissflogii,* and 1d old nauplii of *Artemia salina* | | Regularly fed (typically 5 times per week) with *Rhodomonas baltica* | |
| Substrate | Sabia Corallina, 7–8mm, Dohse Aquaristik, Grafschaft–Gelsdorf, Germany | | No | |



Figure 1. Map of Comau Fjord. Upper left map: Overview of Chilean Patagonia. Lower left map: Gulf of Ancud with connection in the North and South to the Pacific Ocean. Right hand map: Fjord Comau with localities of brachiopod sample collection. In both maps the rectangle marks the location of Comau Fjord.

In summary, *M. venosa* individuals sampled in Chile were transported to Germany and cultured under controlled environmental setting in a climate laboratory. As a culture medium we used artificial seawater, which was prepared by



mixing a commercial salt with deionized water until the desired salinity and chemical composition was achieved. An overview of the culturing setup at both laboratories is available in Table 3. Brachiopods were first left to acclimatize, and prior to the start of experimental treatments labelled using a fluorescent dye – calcein (Sigma, CAS 1461–15–0; 50 mg/l for 3 h) (e.g., Baumgarten et al., 2013; Jurikova et al., in review). Specimens #43 and #63 were cultured at AWI at $pH_3 = 7.66$

($pCO_2 = 1390$ µatm) and $pH_4 = 7.44$ ($pCO_2 = 2610$ µatm) from $29^{th}$ August 2013 to $31^{th}$ March 2014 respectively. Specimens #8004, #8005, #9004, #9005 and #9006 were cultured concurrently at GEOMAR under control or low pH conditions. Specimens #8004 and #8005 were maintained under control settings ($pH_0 = 8.0/8.15$) from $4^{th}$ August 2016 to $5^{th}$ July 2017, conditions similar to the fjord habitat. In contrast, specimens #9004, #9005 and #9006 were cultured under low–pH artificial seawater conditions. Low–pH conditions were mediated by additional bubbling of $CO_2$ at AWI, and $CO_2$-enriched air at

GEOMAR (Table 3). The acidification experiment was performed in two phases; the first one from $4^{th}$ August 2016 to $18^{th}$ April 2017 during which the $pCO_2$ was set to 2000 µatm (corresponding to a $pH_1 = 7.60$), and the second one during which the $pCO_2$ was set to 4000 µatm (corresponding to a $pH_2 = 7.35$) from $18^{th}$ April 2017 to $5^{th}$ July 2017. In order to distinguish between the shell parts participated under the specific pH conditions as well as to allow exact comparison to shells from the control treatment, calcein marking was carried out prior to the second low–pH phase (i.e. before the 4000 µatm experiment).

Parts of the shell grown under specific pH conditions are indicated in Figure 2. In addition to the calcein marking, newly grown shell parts may be distinguished from visible growth lines on the surface of the shell (Figure 2). The total length (defined as maximum distance from the blue line to the anterior margin) of the curved dorsal and ventral valves grown during the 11 months of culturing (Figure 2) varied from < 5 mm to 15.6 mm (Table 4).

## 2.1 Brachiopod samples and culturing set–up




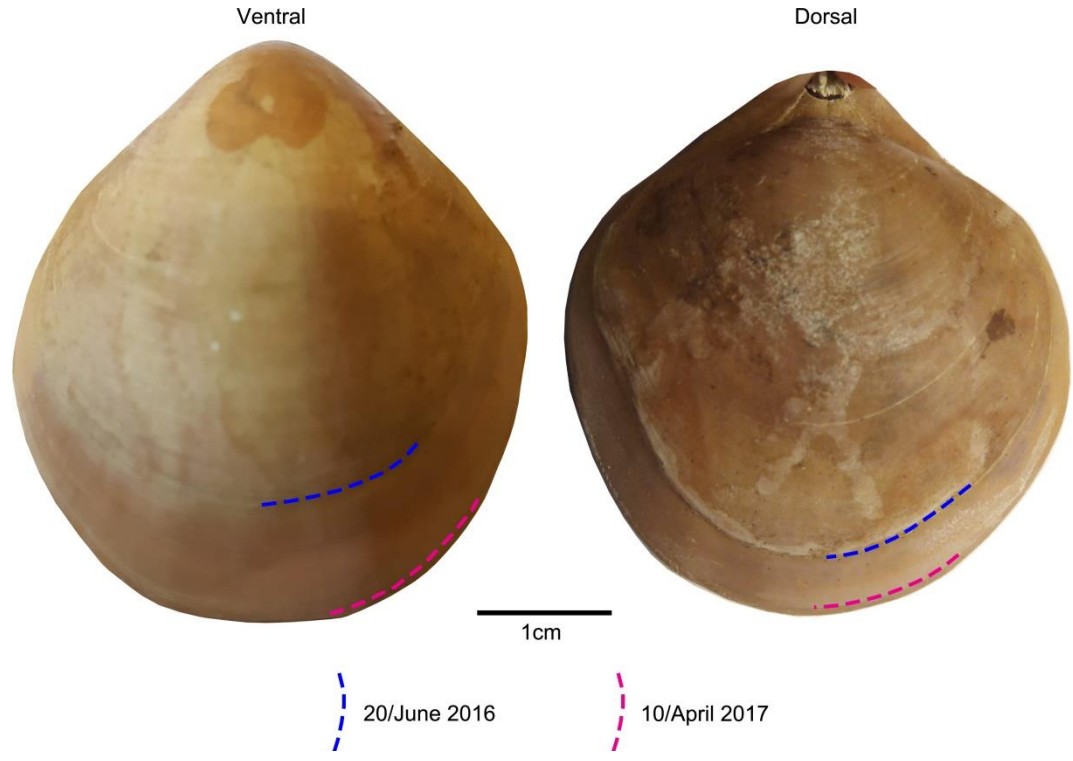

Figure 2. Growth lines marked with calcein on the surface of the brachiopod specimens (#9006).

Table 4. Total shell length of three specimens of *M. venosa* before, during and at the end of the in vitro culturing.

| Sample | Initial anterior–posterior length (mm) | Length–Duration (a) 257 Days (mm) | Length–Duration (b) 78 Days (mm) |
|---|---|---|---|
| #8004 ventral | 31 | 14 ($pH_0$) | 1.6 ($pH_0$) |
| #8005 ventral | 46 | 5 ($pH_0$) | <1 ($pH_0$) |
| #8005 dorsal | 41 | 4 ($pH_0$) | <1 ($pH_0$) |
| #9004 ventral | 41 | 13 ($pH_1$) | 1.2 ($pH_2$) |
| #9005 ventral | 25 | 12 ($pH_1$) | 1.8 ($pH_2$) |
| #9006 ventral | 43 | 9 ($pH_1$) | <1 ($pH_2$) |
| #9006 dorsal | 38 | 8 ($pH_1$) | <1 ($pH_2$) |

5   Note: (a) Culturing from 4 August 2016 to 18 April 2017; (b) Culturing from 18 April 2017 to 5 July 2017; $pH_0 = 8.00–8.14$, $pH_1 = 7.60$, $pH_2 = 7.35$.




## 2.1 Microstructural Analysis

This study followed the sample preparation method for recent shells suggested by Crippa et al. (2016b). In order to obtain more detailed data on microstructural changes, the samples were cut with a diamond blade along different axes and directions (Figure 3A). Subsequently, the samples were immersed in 36 volume hydrogen peroxide ($H_2O_2$) for 24 to 48 hours

5   to remove the organic components. The sectioned surfaces were manually smoothed with 1200 grit sandpaper, then quickly (3 seconds) cleaned with 5% hydrochloric acid (HCl), immediately washed with tap water and air–dried. Finally, the valve sections were gold–coated and analysed by a Cambridge S–360 scanning electron microscope with a lanthanum hexaboride (LaB6) source operating at an acceleration voltage of 20 kV (Dipartimento di Scienze della Terra "A. Desio", Università di Milano).

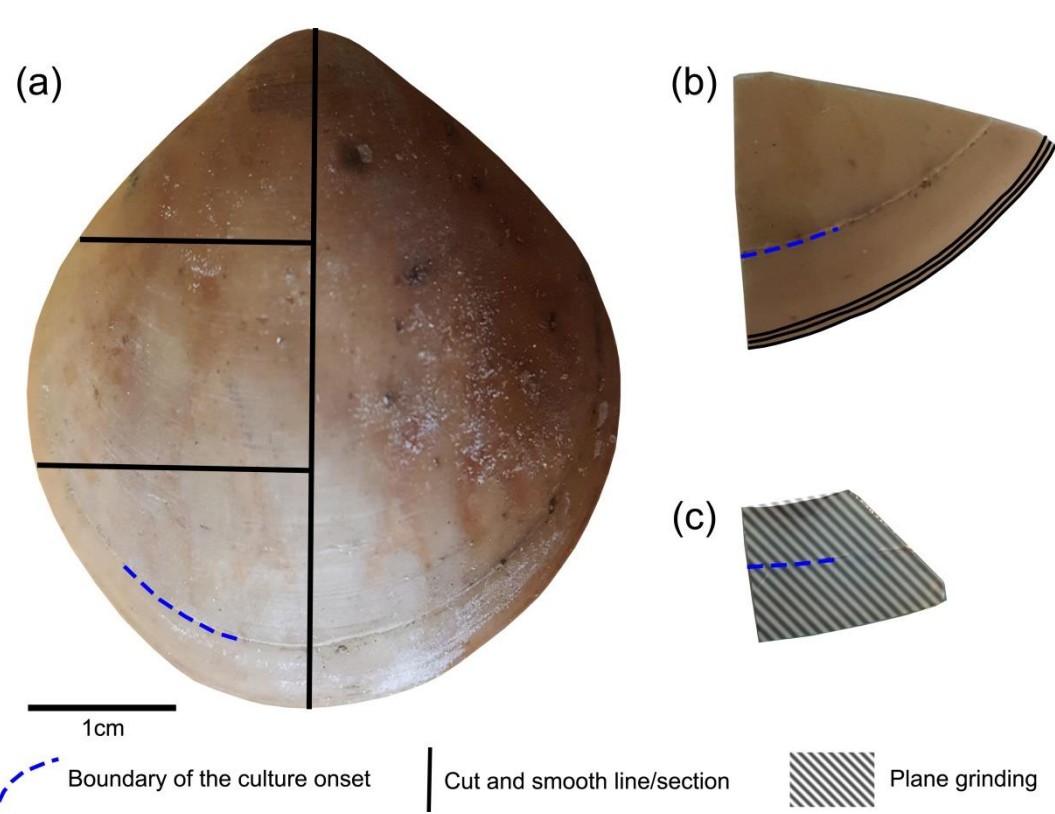

Figure 3. Brachiopod shell sample cut along different axes. A, longitudinal and transverse sections; B, transverse sections at the anterior margin of the shell; C, plane grinding of the external surface of the shell.

15   The methods described by Ye et al. (2018a) were followed to investigate the basic microstructural units (fibres) in SEM images. We focused primarily on the anterior margin of the valves, the part that was produced during culturing (hereinafter referred to as *during–culturing*) under different pH conditions. Therefore, additional transverse sections along the growth



lines were obtained in the most anterior part (black lines in Figure 3B) by manually smoothing with 1200 grit sandpaper. Plane grinding was performed on the external surface of the shell (Figure 3) to investigate the distribution of endopunctae. The thickness of the primary layer was measured on the SEM images of specimens #8005 and #9006 (Figure 4A) in different positions along the longitudinal growth axis (posterior, central and anterior regions). In the vicinity of the transition from

5    natural growth to cultured growth, the region was further subdivided into four sub–zones.

To calculate and measure the density and diameter (max) of endopunctae, squares (800 μm×800 μm) were located randomly over the smoothed external surface of the anterior shell (Figure 4B). Four sub-zones (C2, A1, A2, A3) were defined according to their position along the posterior-anterior direction, while distinguishing the part of the shell produced *before– culturing* and that produced *during–culturing*.





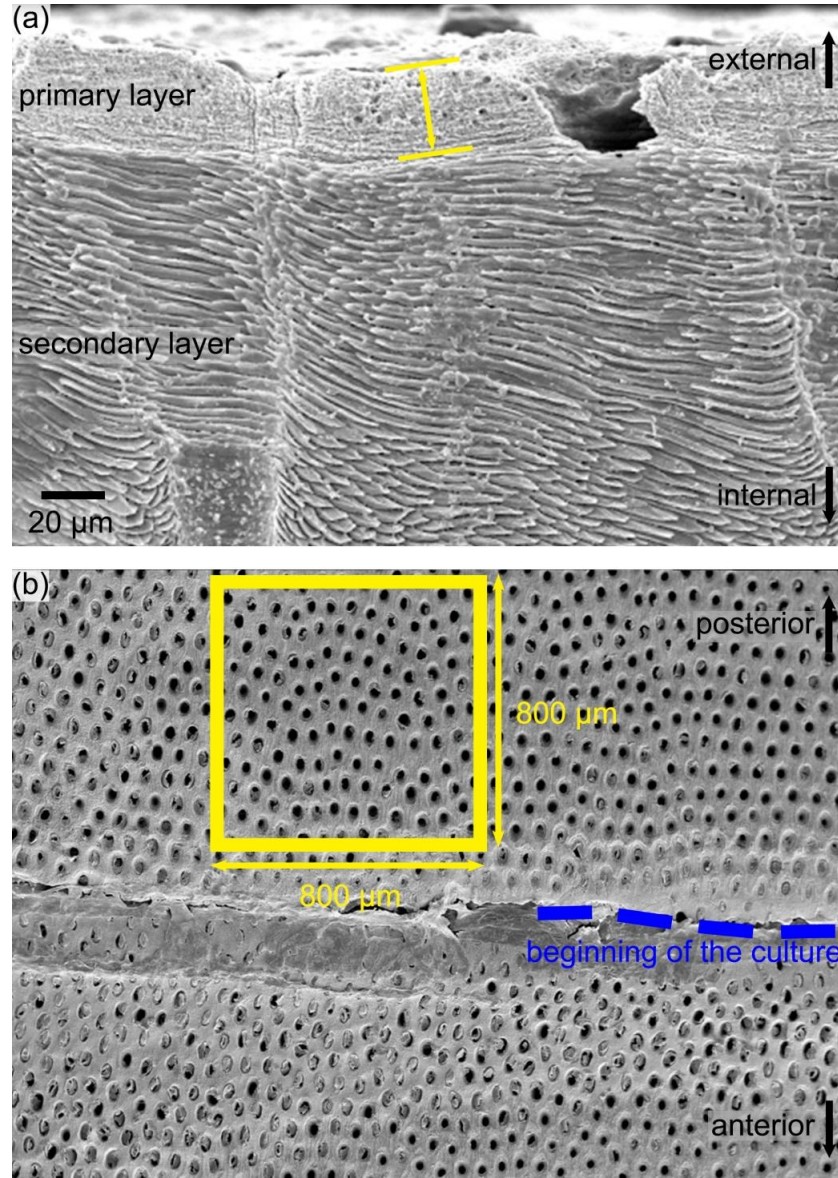

Figure 4. Measurement methods used for the thickness of primary layer and the density of the endopunctae. Note that for the latter, endopunctae were counted when included for more than their half diameter inside the square.

5   For morphometric analyses, fibres were manually outlined using polygonal lasso in Adobe Photoshop CS6, and size and shape parameters were measured with Image-Pro Plus 6.0 and ImageJ (for convexity). In particular, following Ye et al. (2018a, b) we measured/calculated the Feret diameter (max), Area, Roundness [$4Area/\pi \times$ Feret diameter $(max)^2$] and Convexity (Convex Perimeter/Perimeter). The width of an individual fibre roughly corresponds to the Feret diameter (max), whereas its height corresponds to the Feret diameter (min) (see Figure 6 in Ye et al., 2018a).





As individual fibres are irregular in shape in the most anterior sections of brachiopods, the morphometric measurement method proposed by Ye et al. (2018a, b) is not always suitable. Thus, modifications had to be made to Ye et al. (2018a, b) measurement method to make the comparative morphometric analysis of the fibres more robust (Figure 5A, 5B). First, all SEM images were oriented in the same direction with the base of the primary layer facing upwards. Then a uniform size

5   zone (20 μm × 20 μm) was selected for additional measurements with the upper side of the square always placed at the boundary between the primary and the secondary layers (Figure 5C). Two new methods were developed and were then applied: for Method 1, the width of fibres crossed by two standard lines was measured, which were always located in the same position and at the same distance in all the selected zones (yellow and orange lines in Figure 5 method 1). For Method 2, we calculated the number of boundaries based on the number of fibres crossed by the two standard lines (Figure 5 method

10   2). Samples were named according to the following nomenclature, the most anterior transection zone of the ventral valve was named Z1, the second most anterior transection zone of the ventral valve Z2 and so on, the most anterior transection zone of the dorsal valve was named Z4; The standard line facing towards the primary layer was named "1" and the second standard line "2" (example: "Z1–1" is the sample of the standard line facing towards the primary layer at the most anterior transection zone of the ventral valve).





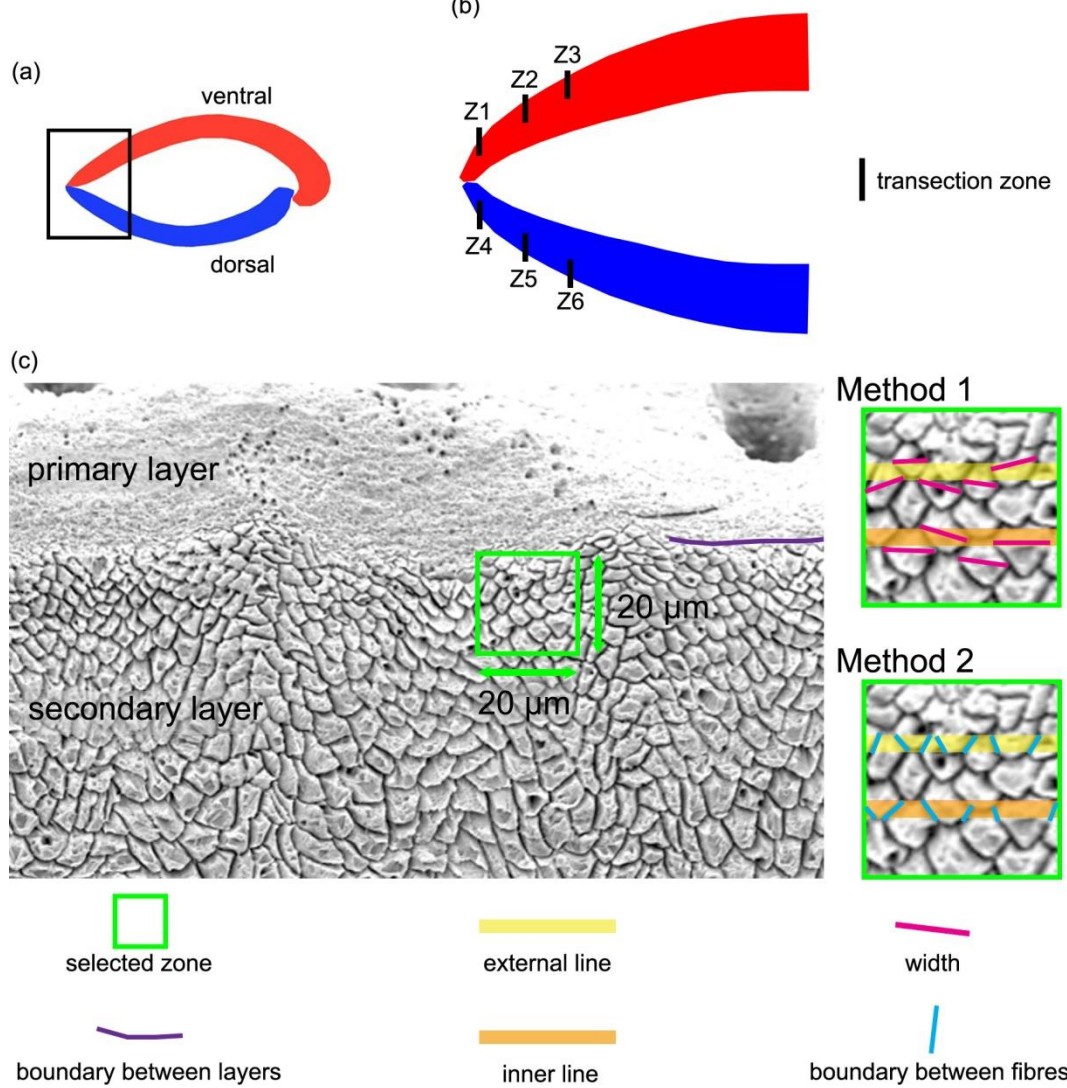

Figure 5. Methods of measurements used in the anterior transverse sections. All SEM images are oriented in the same direction: base of the primary layer facing upwards. A square (20 μm × 20 μm) with its upper side just overlapping the boundary between the primary and secondary layer was analysed. Method 1, refers to the measurement of the width of the

5  fibres crossed by two standard lines, which were located in the same position and at the same distance in all 194 squares analysed (yellow and orange lines); Method 2, calculation of the numbers of boundaries between the fibres, which are crossed by two standard lines were carried out.

## 2.3 Carbonate stable isotopes analyses

Cleaned shells of specimens #8004, #8005, #9004, #9005 and #9006 were chosen for carbon and oxygen isotope analyses.

10  For specimens #8005 and #9006, the primary layer and surface contaminants were manually and chemically removed by





leaching with 10 % HCl, rinsed with distilled water and air–dried. Individual growth increments exclusively come from the secondary layer, and were separated from the shell in both dorsal and ventral valves using a WECHEER (WE 248) microdrill with tungsten–carbide milling bit. Shell increment fragments, of similar width, were then powdered using an agate mortar and pestle. For carbon and oxygen isotope analyses about 250 μg of powdered calcite of each sample was analysed

with a Finnigan GasBench connected to a Delta V (Thermo Fisher Scientific Inc., Waltham, Massachusetts, USA) mass spectrometer at the Dipartimento di Scienze della Terra, Università degli Studi di Milano, Italy. Isotope values ($\delta^{18}$O, $\delta^{13}$C) are reported as per mil (‰) deviations of the isotopic ratios ($^{18}$O/$^{16}$O, $^{13}$C/$^{12}$C) calculated to the V–PDB scale using a within-run laboratory internal standard (MAMI) calibrated against the International Atomic Energy Agency 603 (IAEA-603; $\delta^{18}$O: -2.37 ± 0.04 ‰, $\delta^{13}$C: +2.46 ± 0.01 ‰) and NBS 18 ($\delta^{18}$O: -23.2 ± 0.1 ‰, $\delta^{13}$C: -5.014 ± 0.035 ‰) standards. Analytical

reproducibility (1σ) for these analyses was better than 0.04‰ for $\delta^{13}$C and 0.1‰ for $\delta^{18}$O (Appendix 1). Another set of shells, #8004, #9004 and #9005, were gently rinsed with ultra pure water (Milli–Q) and dried for a few days on a hotplate at 40 ℃ in a clean flow hood. Targeted parts of the shell were sampled for powder under binoculars using a precision drill (Proxxon) with a mounted dental tip. Stable isotope analyses of powders of these specimens were performed at GEOMAR, Kiel on a Thermo Finnigan MAT 252 mass spectrometer coupled online to an automated Kiel carbonate preparation line. The external

reproducibility (1σ) of in–house carbonate standards was better than ± 0.1 ‰ and ± 0.08 ‰ for $\delta^{13}$C and $\delta^{18}$O, respectively (Appendix 2).

### 2.4 Stable isotopes analyses of water samples

In addition to carbon and oxygen isotope analyses of shells, analyses were also carried out on seawater samples collected from the culturing tanks. Measurements of $\delta^{13}$C$_{DIC}$ and $\delta^{18}$O$_{H2O}$ were performed using Thermo Scientific™ Delta Ray™ IRIS

with URI Connect.

Isotope values ($\delta^{13}$C, $\delta^{18}$O) are reported as per mil (‰) deviations of the isotopic ratios ($^{13}$C/$^{12}$C, $^{18}$O/$^{16}$O) calculated to the VPDB scale for $\delta^{13}$C and VSMOW scale for $\delta^{18}$O values. Analytical reproducibility (1σ) on 3 aliquots of each water sample, was ≤ 0.03‰ for both $\delta^{13}$C and $\delta^{18}$O values (Appendix 3).

### 3.1 Primary layer thickness

The thickness of the primary layer was measured at different positions along the shell from the posterior (umbonal) region to the *before–culturing* portion and finally to the anterior valve margin (Figure 6). Generally, in the posterior part of *M. venosa,* the primary layer is missing, or it has the lowest recorded thickness. Then the primary layer progressively thickens toward the central and anterior parts. The thickest primary layer within the same valve is always located just before the beginning of the culture (*before–culturing* portion, Table 5). During culturing the thickness of the primary layer decreases. A most distinct

change was observed in specimen #9006 cultured at the lowest pH condition (pH$_1$: pH 7.6, and pH$_2$: pH 7.35) followed by another progressive increase in both valves *during–culturing*. In contrast, the thickness of the primary layer of the control condition specimen (#8005) remained stable (dorsal valve) or slightly decreased (Figure 6, ventral valve; Table 5).

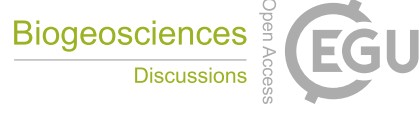

Table 5. Statistical comparison of thickness of the primary layer (μm) along the ontogenetic direction of both valves of specimens #8005 and #9006. ①: Specific positions see Figure 6. N = number of measurement. Significant values ($p$–value ≤ 0.05) are marked in bold style.

| Sample | Position[①] | N | Mean | STD | Min | Max | $p$-values | $p$-values |
|---|---|---|---|---|---|---|---|---|
| #8005 dorsal | P | 4 | 11.82 | 1.05 | 10.55 | 13.02 | | |
| | C1 | 8 | 11.40 | 2.29 | 8.50 | 15.05 | P vs C1 0.755 | |
| | C2 | 10 | 28.99 | 4.79 | 22.15 | 36.65 | C1 vs C2 < 0.001 | |
| | A1 | 8 | 24.36 | 2.52 | 19.80 | 27.06 | C2 vs A1 0.033 | |
| | A2 | 7 | 24.83 | 2.15 | 21.67 | 27.94 | A1 vs A2 0.726 | |
| | A3 | 1 | 21.77 | NA | NA | NA | A2 vs A3 NA | |
| | | | | | | | | #8005DP vs #9006DP 0.120 |
| #8005 ventral | P | 2 | 17.64 | 2.36 | 15.28 | 20 | | #8005DC1 vs #9006DC1 < 0.001 |
| | C1 | 6 | 13.68 | 3.96 | 8.50 | 20.52 | P vs C1 NA | |
| | C2 | 8 | 47.57 | 2.49 | 42.55 | 50.27 | C1 vs C2 < 0.001 | #8005DC2 vs #9006DC2 < 0.001 |
| | A1 | 8 | 44.18 | 2.68 | 38.33 | 47.98 | C2 vs A1 0.028 | |
| | A2 | 6 | 42.09 | 3.85 | 36.06 | 45.04 | A1 vs A2 0.289 | #8005DA1 vs #9006DA1 0.088 |
| | A3 | 4 | 34.09 | 3.51 | 29.63 | 37.52 | A2 vs A3 0.017 | |
| | | | | | | | | #8005DA2 vs #9006DA2 0.101 |
| #9006 dorsal | P | 7 | 9.08 | 2.77 | 5.56 | 14.64 | | #8005DA3 vs #9006DA3 NA |
| | C1 | 10 | 18.78 | 2.04 | 16.90 | 22.50 | P vs C1 < 0.001 | #8005VP vs #9006VP NA |
| | C2 | 11 | 46.91 | 5.22 | 35.92 | 55.86 | C1 vs C2 < 0.001 | |
| | A1 | 10 | 28.83 | 6.65 | 19.04 | 39.93 | C2 vs A1 < 0.001 | #8005VC1 vs #9006VC1 0.123 |
| | A2 | 8 | 28.06 | 4.03 | 22.50 | 36.69 | A1 vs A2 0.779 | |
| | A3 | 4 | 32.84 | 3.55 | 29.10 | 38.65 | A2 vs A3 0.096 | #8005VC2 vs #9006VC2 0.194 |
| | | | | | | | | #8005VA1 vs #9006VA1 < 0.001 |
| #9006 ventral | P | 7 | 9.78 | 1.72 | 6.07 | 11.79 | | #8005VA2 vs #9006VA2 0.007 |
| | C1 | 9 | 16.75 | 2.77 | 12.61 | 21.29 | P vs C1 < 0.001 | |
| | C2 | 12 | 45.16 | 4.34 | 35.09 | 51.40 | C1 vs C2 < 0.001 | #8005VA3 vs #9006VA3 0.027 |
| | A1 | 11 | 36.92 | 3.82 | 26.62 | 42.54 | C2 vs A1 < 0.001 | |
| | A2 | 4 | 32.95 | 2.91 | 30.84 | 37.95 | A1 vs A2 0.102 | |
| | A3 | 5 | 40.55 | 2.63 | 37.78 | 45.23 | A2 vs A3 0.008 | |





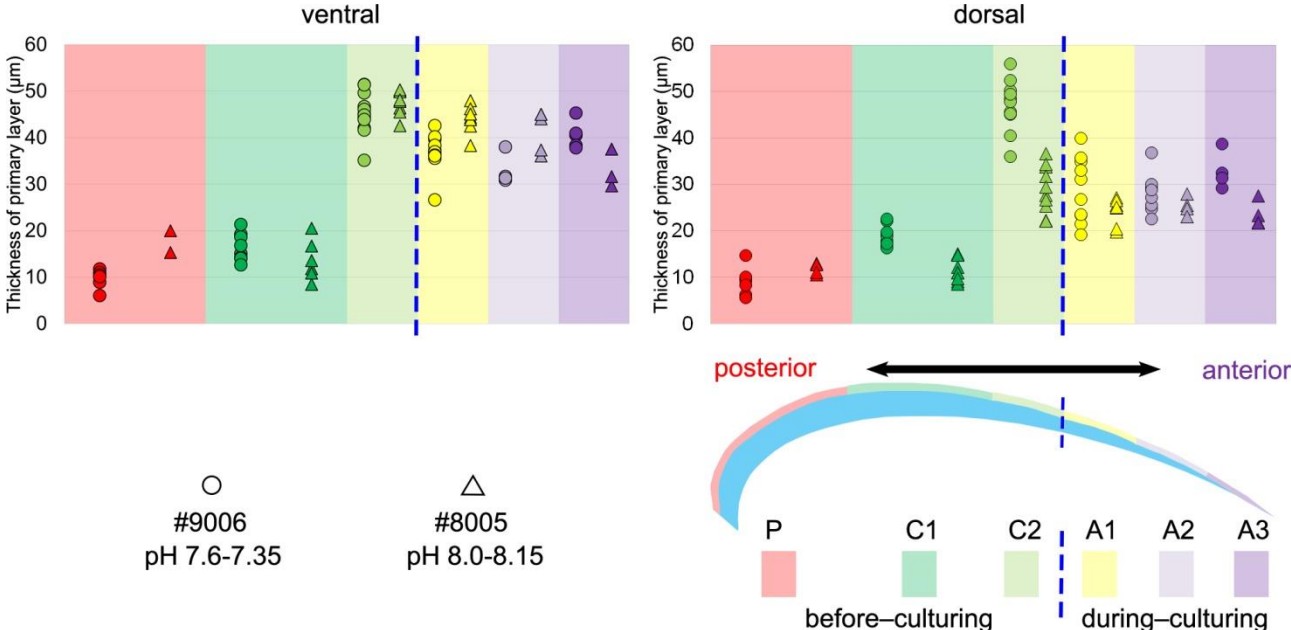

Figure 6. Variations of the thickness of the primary layer (ventral and dorsal valve) of a *M. venosa* specimen cultured at pH 7.35 and 7.6 (#9006) and a specimen cultured at pH 8.0–8.15 (#8005).

## 3.2 Endopunctae density and size

On the externally–ground surface of the anterior part, the total number and the diameter (max) of endopunctae in a squared frame (800 μm × 800 μm) was measured in four zones of the *before-culturing* and of the *during–culturing* parts of the shell (Figure 7). Generally, the density of endopunctae gradually increases along the selected transect (from ca. 185 /mm$^2$ to ca. 305 /mm$^2$ in ventral valve and from ca. 220 /mm$^2$ to ca. 280 /mm$^2$; Table 6). The size of endopunctae increases along the selected transect in the ventral valve (from ca. 17 μm to 33 μm; Table 7), but it slightly decreases in the dorsal valve (from ca. 36 μm to ca. 21 μm; Table 7). These trends are observed in both specimens cultured at different pH conditions. However, it is worth noting that in the most anterior part (*during–culturing*) of the ventral valve of #9006 (cultured at pH$_2$: pH 7.35), the density of endopunctae sharply increases and their diameter reaches the maximum recorded values (Table 6).

Table 6. Statistical comparison of the number of endopunctae (per mm$^2$) on both valves of #8005 and #9006. ①: Specific zones see to Figure 7. N = number of measurement.

| Sample | Zone① | N | Mean | STD | Min | Max |
|---|---|---|---|---|---|---|
| | C2 | 3 | 236 | 8.6 | 225 | 250 |
| #8005 dorsal | A1 | 1 | 280 | NA | NA | NA |
| | A2 | 2 | 244 | 12.5 | 231 | 256 |
| | A3 | 2 | 281 | 14 | 267 | 295 |





| | | | | | |
|---|---|---|---|---|---|
| #8005 ventral | C2 | 2 | 225 | 1.6 | 223 | 227 |
| | A1 | 1 | 242 | NA | NA | NA |
| | A2 | 2 | 241 | 5.5 | 236 | 247 |
| | A3 | 2 | 269 | 6.3 | 263 | 275 |
| #9006 dorsal | C2 | 2 | 221 | 8.6 | 213 | 230 |
| | A1 | 1 | 269 | NA | NA | NA |
| | A2 | 2 | 250 | 3.1 | 247 | 253 |
| | A3 | 2 | 266 | 3.1 | 263 | 269 |
| #9006 ventral | C2 | 2 | 186 | 3.1 | 183 | 189 |
| | A1 | 1 | 234 | NA | NA | NA |
| | A2 | 2 | 230 | 4.7 | 225 | 234 |
| | A3 | 2 | 308 | 1.6 | 306 | 309 |

Table 7. Statistical comparison of the diameter (max) (µm) of endopunctae on both valves of #8005 and #9006. ①: Specific zones see Figure 7. N = number of measurement. Significant values ($p$-value ≤ 0.05) are marked in bold style.

| Sample | Zone① | N | Mean | STD | Min | Max | $p$-values | $p$-values |
|---|---|---|---|---|---|---|---|---|
| #8005D | C2 | 21 | 36.04 | 1.78 | 33.2 | 40.4 | C2 vs A1 < **0.001** | #8005DC2 vs #9006DC2 |
| | A1 | 10 | 28.36 | 2.33 | 25 | 32.1 | A1 vs A2 < **0.001** | **0.025** |
| | A2 | 15 | 18.77 | 1.10 | 17 | 21.1 | A2 vs A3 **0.001** | |
| | A3 | 13 | 21.8 | 2.53 | 18.2 | 26.2 | | #8005DA1 vs #9006DA1 < |
| #8005V | C2 | 11 | 17.07 | 1.42 | 13.6 | 18.9 | C2 vs A1 < **0.001** | **0.001** |
| | A1 | 13 | 20.88 | 2.22 | 17.1 | 24.3 | A1 vs A2 **0.007** | |
| | A2 | 12 | 18.74 | 0.84 | 18 | 20.9 | A2 vs A3 < **0.001** | #8005DA2 vs #9006DA2 < |
| | A3 | 14 | 26.83 | 2.83 | 23 | 33.1 | | **0.001** |
| #9006D | C2 | 12 | 32.54 | 4.39 | 26.2 | 40 | C2 vs A1 0.178 | |
| | A1 | 13 | 34.63 | 2.33 | 29 | 37.2 | A1 vs A2 **0.012** | #8005DA3 vs #9006DA3 < |
| | A2 | 11 | 32.02 | 2.12 | 27.5 | 36.1 | A2 vs A3 **0.005** | **0.001** |
| | A3 | 19 | 28.75 | 3.51 | 23 | 34.4 | | |
| #9006V | C2 | 13 | 29.98 | 2.04 | 24.3 | 33 | C2 vs A1 < **0.001** | #8005VC2 vs #9006VC2 |
| | A1 | 12 | 38.66 | 2.41 | 35.5 | 42.6 | A1 vs A2 < **0.001** | < **0.001** |
| | A2 | 14 | 32.51 | 4.08 | 25.3 | 40.3 | A2 vs A3 0.516 | |
| | A3 | 24 | 33.70 | 5.82 | 22 | 44.3 | | #8005VA1 vs #9006VA1 < |





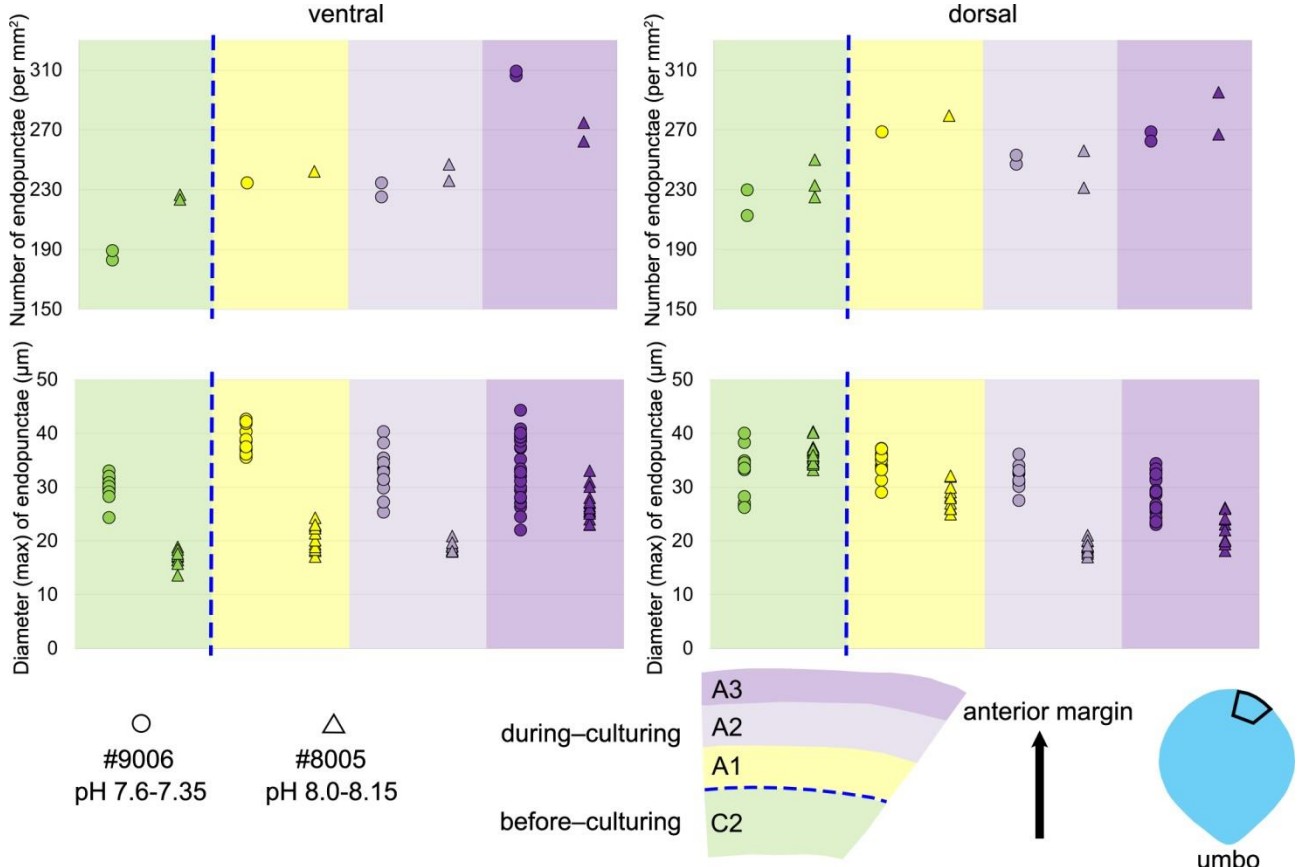

Figure 7. Variations in the number and diameter (max) of endopunctae in the ventral and dorsal valve from a specimen of *M. venosa* cultured at pH 7.35 and 7.6 (#9006) and a specimen cultured at pH 8.0–8.15 (#8005).

## 3.3 Shell morphometrics

### 3.3.1 Before–culturing

Ontogenetic variation in fibre morphometry is not obvious when all six adult specimens are considered (Table 8). However, clearer growth trends can be observed when considering the data from each single specimen separately, where *t*-tests on morphometric data from specimens #8005 and #9006 show that there are significant differences in Feret diameter (max) and Roundness between the posterior and the middle part of the shell (Table 9). Overall, in specimens #8005 and #9006 fibres become wider from the posterior to mid–shell. In contrast, #63 shows an opposite trend along the posterior to mid-shell direction (Figure 8). The fibre size and shape in the other specimens are rather constant.

Table 8. Statistical comparison of fibres size and shape data of the posterior external vs central middle parts of both the ventral valve and the dorsal valve. NC: non-cultured samples #158, #223; CU: cultured samples #43, #63, #8005, #9006;



Vpe: ventral posterior external, Vcm: ventral central middle, Dpe: dorsal posterior external, Dcm: dorsal central middle, N: number of measurement. Significant values ($p$–value $\leq 0.05$) are marked in bold style.

| Sample | Position | N | Mean | STD | Min | Max | $p$-values |
|---|---|---|---|---|---|---|---|
| Feret diameter (max) (µm): | | | | | | | |
| NC | Vpe | 7 | 13.79 | 3.22 | 6.97 | 17.33 | NC Vpe vs CU Vpe 0.486 |
| CU | Vpe | 26 | 12.47 | 6.58 | 4.59 | 24.78 | NC Vcm vs CU Vcm 0.633<br>NC Vpe vs NC Vcm 0.533 |
| NC | Vcm | 32 | 12.98 | 2.91 | 7.09 | 20.61 | CU Vpe vs CU Vcm 0.572 |
| CU | Vcm | 65 | 13.24 | 2.15 | 8.68 | 18.84 | |
| NC | Dpe | 8 | 18.36 | 4.22 | 13.30 | 24.46 | NC Vpe vs CU Vpe 0.486 |
| CU | Dpe | 12 | 10.78 | 6.36 | 4.85 | 22.29 | NC Vcm vs CU Vcm 0.633<br>NC Vpe vs NC Vcm 0.533 |
| NC | Dcm | 12 | 12.14 | 1.13 | 9.84 | 14.42 | CU Vpe vs CU Vcm 0.572 |
| CU | Dcm | 46 | 12.51 | 1.57 | 9.45 | 15.89 | |
| Roundness: | | | | | | | |
| NC | Vpe | 7 | 0.308 | 0.077 | 0.239 | 0.475 | NC Vpe vs CU Vpe 0.717 |
| CU | Vpe | 26 | 0.296 | 0.074 | 0.172 | 0.446 | NC Vcm vs CU Vcm 0.396<br>NC Vpe vs NC Vcm 0.296 |
| NC | Vcm | 29 | 0.282 | 0.051 | 0.179 | 0.389 | CU Vpe vs CU Vcm 0.146 |
| CU | Vcm | 65 | 0.272 | 0.051 | 0.180 | 0.421 | |
| NC | Dpe | 8 | 0.220 | 0.034 | 0.169 | 0.268 | **NC Dpe vs CU Dpe 0.003** |
| CU | Dpe | 12 | 0.337 | 0.100 | 0.155 | 0.500 | **NC Dcm vs CU Dcm 0.028**<br>**NC Dpe vs NC Dcm 0.005** |
| NC | Dcm | 11 | 0.311 | 0.068 | 0.192 | 0.416 | **CU Dpe vs CU Dcm 0.048** |
| CU | Dcm | 48 | 0.269 | 0.051 | 0.162 | 0.378 | |
| Convexity: | | | | | | | |
| NC | Vpe | 7 | 0.985 | 0.004 | 0.979 | 0.991 | NC Vpe vs CU Vpe 0.309 |
| CU | Vpe | 26 | 0.982 | 0.008 | 0.968 | 0.999 | NC Vcm vs CU Vcm 0.655<br>NC Vpe vs NC Vcm 0.823 |
| NC | Vcm | 32 | 0.984 | 0.005 | 0.975 | 1.000 | CU Vpe vs CU Vcm 0.257 |
| CU | Vcm | 62 | 0.984 | 0.008 | 0.965 | 1.008 | |
| NC | Dpe | 8 | 0.987 | 0.006 | 0.979 | 0.998 | NC Dpe vs CU Dpe 0.604 |
| CU | Dpe | 11 | 0.985 | 0.007 | 0.973 | 0.998 | NC Dcm vs CU Dcm 0.273<br>NC Dpe vs NC Dcm 0.543 |
| NC | Dcm | 12 | 0.984 | 0.008 | 0.973 | 1.000 | CU Dpe vs CU Dcm 0.207 |
| CU | Dcm | 48 | 0.982 | 0.008 | 0.967 | 1.001 | |



Table 9. Statistical comparison of fibres size and shape data of the posterior external vs central middle area for #8005 and #9006, considering both valves together. pe: posterior external, cm: central middle, N: number of measurement. Significant values ($p$–value $\leq 0.05$) are marked in bold style.

| Sample | Position | N | Mean | STD | Min | Max | *p*-values |
|---|---|---|---|---|---|---|---|
| Feret diameter (max) (µm): | | | | | | | |
| #8005 | pe | 10 | 7.92 | 3.30 | 4.85 | 14.97 | #8005 pe vs #9006 pe 0.265 |
| #8005 | cm | 36 | 12.29 | 1.64 | 9.63 | 15.89 | #8005 cm vs #9006 cm 0.171 |
| #9006 | pe | 10 | 6.45 | 1.95 | 4.59 | 11.41 | #8005 pe vs #8005 cm 0.003 |
| #9006 | cm | 25 | 11.73 | 1.39 | 8.68 | 15.24 | #9006 pe vs #9006 cm < 0.001 |
| Roundness: | | | | | | | |
| #8005 | pe | 10 | 0.33 | 0.097 | 0.155 | 0.446 | #8005 pe vs #9006 pe 0.547 |
| #8005 | cm | 36 | 0.25 | 0.045 | 0.162 | 0.374 | #8005 cm vs #9006 cm 0.012 |
| #9006 | pe | 10 | 0.35 | 0.079 | 0.232 | 0.500 | #8005 pe vs #8005 cm 0.040 |
| #9006 | cm | 26 | 0.28 | 0.043 | 0.195 | 0.369 | #9006 pe vs #9006 cm 0.022 |
| Convexity: | | | | | | | |
| #8005 | pe | 10 | 0.981 | 0.007 | 0.973 | 0.994 | #8005 pe vs #9006 pe 0.308 |
| #8005 | cm | 35 | 0.982 | 0.008 | 0.968 | 1.001 | #8005 cm vs #9006 cm 0.277 |
| #9006 | pe | 9 | 0.985 | 0.007 | 0.975 | 0.999 | #8005 pe vs #8005 cm 0.829 |
| #9006 | cm | 26 | 0.984 | 0.007 | 0.967 | 1.001 | #9006 pe vs #9006 cm 0.775 |



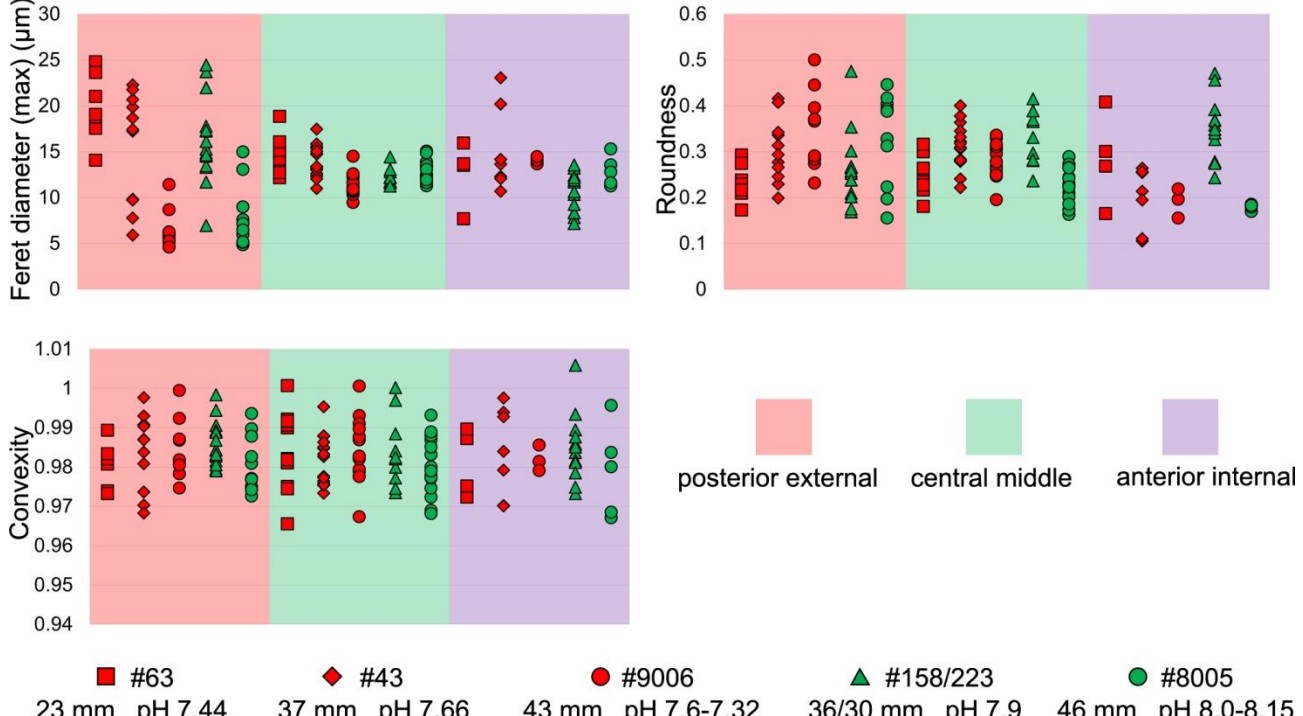

Figure 8. Comparisons of the fibre size and shape of *M. venosa* (ventral and dorsal valve) at different positions along the posterior-anterior axis; pH conditions of culturing or natural environment are reported. One circle point represents one measurement. Outliers have been removed, the latter were identified with Tukey's fences (Tukey, 1977), when falling outside the fences F1 and F2 [F1 = Q1 - 1.5IQR; F2 = Q3 + 1.5IQR; Q1/Q3 = first/third quartiles; IQR (interquartile range) = Q3 - Q1].

### 3.3.2 During–culturing

Transverse sections obtained by smoothing of the anterior part of the shell allowed to measure the width of 1392 fibres [Max Ferret diameter (max) see in Method 1], and select 388 sub–zones for fibre boundary calculation. In addition, they allowed us to focus on the parts that were produced under the different low–pH treatments ($pH_1$, $pH_2$, $pH_3$ and $pH_4$, respectively).

In all six specimens, the width of fibres increases and the number of boundaries decreases along a transect from the more external subzone to the immediately inner subzone (e.g., Z1–1 to Z1–2; Z2–1 to Z2–2; and Z3–1 to Z3–2 in Figure 9A, B, C, D). That means, even within less than 10 μm distance, the size of fibres become larger from the exterior to the interior part of the shell with growth.

Results from #9006 were compared to those of control specimen #8005 ($pH_0$). Specimen #9006 cultured under low–pH conditions ($pH_1$ and $pH_2$) had narrower fibres and a higher number of fibre boundaries when compared to that of control



specimen #8005 (Figure 9A, C). It is worth noting that, in comparison between the two specimens, the fibres from Z1–2 and Z2–2 of #9006 are significantly smaller than those of #8005. However, there is no significant difference in the size of fibres from subzone Z3–2 between the two specimens (Table 10).

The results from specimens (#43 and #63) grown under low pH conditions ($pH_3$ and $pH_4$) for a short time interval of 214 days are difficult to interpret, as in this case, there is no direct control experiment sample to compare with the cultured specimens (Figure 9B, D). The specimens grown in the natural environment (#158, #223) have a different size and age and so different growth rates may affect the size of the fibres.

Table 10. Statistical comparison of fibres size of *M. venosa* (ventral and dorsal valve) in the anterior transverse sections. ①: specific zones see Figure 9. N: number of measurement. Significant values ($p$–value ≤ 0.05) are marked in bold style.

| Sample | position ① | N | Mean (μm) | STD | Min (μm) | Max (μm) | Difference between means (μm) and ($p$–values) | Difference between means (μm) and ($p$–values) | Difference between means (μm) and ($p$–values) |
|---|---|---|---|---|---|---|---|---|---|
| | | | | | | | #9006 vs #8005 for the same zone | Z1 vs Z2, Z2 vs Z3 for the same vertical position in the same specimen | Z1vs Z2, Z2 vs Z3 for the same transverse position in the same specimen |
| #9006 | Z1–1 | 26 | 4.43 | 1.06 | 2.86 | 6.74 | 0.23 (0.402) | #9006 Z1–1 vs Z2–1 | |
| #8005 | Z1–1 | 49 | 4.66 | 1.13 | 1.89 | 7.37 | | **0.60 (0.013)** | |
| #9006 | Z2–1 | 53 | 3.83 | 0.66 | 2.76 | 5.30 | 0.12 (0.419) | #9006 Z2–1 vs Z3–1 0.07 (0.650) | #9006 Z1 vs Z2 |
| #8005 | Z2–1 | 65 | 3.95 | 1.03 | 2.06 | 6.46 | | | **0.48 (0.011)** |
| #9006 | Z3–1 | 38 | 3.76 | 0.80 | 2.32 | 5.55 | 0.32 (0.134) | #8005 Z1–1 vs Z2–1 **0.71 (0.001)** | #9006 Z2 vs Z3 |
| #8005 | Z3–1 | 44 | 4.08 | 1.05 | 2.22 | 7.53 | | #8005 Z2–1 vs Z3–1 0.13 (0.554) | 0.14 (0.323) |
| #9006 | Z1–2 | 26 | 4.71 | 1.27 | 2.76 | 8.38 | **0.74 (0.024)** | #9006 Z1–2 vs Z2–2 0.33 (0.200) | #8005 Z1 vs Z2 |
| #8005 | Z1–2 | 46 | 5.45 | 1.29 | 2.94 | 10.43 | | | **0.59 (< 0.001)** |
| #9006 | Z2–2 | 48 | 4.38 | 0.90 | 2.87 | 7.00 | **0.62 (0.001)** | #9006 Z2–2 vs Z3–2 0.30 (0.144) | |
| #8005 | Z2–2 | 59 | 5.00 | 0.97 | 2.94 | 7.16 | | | #8005 Z2 vs Z3 0.09 (0.595) |
| #9006 | Z3–2 | 40 | 4.68 | 1.01 | 2.57 | 7.76 | 0.40 (0.087) | #8005 Z1–2 vs Z2–2 **0.45 (0.048)** | |
| #8005 | Z3–2 | 38 | 5.08 | 1.00 | 3.02 | 7.78 | | #8005 Z2–2 vs Z3–2 0.08 (0.720) | |
| #9006 | Z4–1 | 23 | 3.79 | 0.71 | 2.72 | 4.99 | **0.72 (0.003)** | | |
| #8005 | Z4–1 | 58 | 4.51 | 1.02 | 2.15 | 7.11 | | #9006 Z4–1 vs Z5–1 0.11 (0.594) | |
| #9006 | Z5–1 | 24 | 3.68 | 0.72 | 2.54 | 5.19 | NA | | #9006 Z4 vs Z5 0.09 (0.615) |
| #9006 | Z4–2 | 33 | 4.61 | 0.89 | 3.15 | 6.55 | 0.24 (0.272) | | |
| #8005 | Z4–2 | 52 | 4.85 | 1.01 | 3.07 | 6.90 | | #9006 Z4–2 vs Z5–2 0.06 (0.811) | |
| #9006 | Z5–2 | 24 | 4.67 | 1.08 | 2.79 | 7.48 | NA | | |





| | | | | | | | #63 vs #43 vs #158/223 for the same zone | Z1 vs Z2 for the same vertical position in the same specimen | Z1 vs Z2 for the same transverse position in the same specimen |
|---|---|---|---|---|---|---|---|---|---|
| #63 | Z1–1 | 36 | 3.37 | 0.59 | 2.39 | 4.97 | #63 vs #158/223 **0.40 (0.013)** | #63 Z1–1 vs Z2–1 **0.72 (< 0.001)** | |
| #43 | Z1–1 | 70 | 3.73 | 0.98 | 1.63 | 6.94 | | | |
| #158/223 | Z1–1 | 29 | 2.97 | 0.66 | 2.03 | 4.52 | #43 vs #158/223 0.17 (0.404) | #43 Z1–1 vs Z2–1 0.26 (0.109) | |
| #63 | Z2–1 | 24 | 4.09 | 0.75 | 2.84 | 5.85 | #43 vs #158/223 0.07 (0.691) | #158/223 Z1–1 vs Z2–1 **0.95 (< 0.001)** | #63 Z1 vs Z2 **0.80 (< 0.001)** |
| #43 | Z2–1 | 61 | 3.99 | 0.82 | 1.95 | 5.88 | | | |
| #158/223 | Z2–1 | 56 | 3.92 | 0.83 | 2.17 | 6.14 | | | #43 Z1 vs Z2 **0.40 (0.001)** |
| #63 | Z1–2 | 35 | 4.02 | 0.87 | 2.56 | 6.19 | #63 vs #158/223 **0.73 (0.001)** | | #158/223 Z1 vs Z2 **1.2 (< 0.001)** |
| #43 | Z1–2 | 71 | 4.04 | 0.87 | 2.16 | 7.24 | | #63 Z1–2 vs Z2–2 **0.95 (< 0.001)** | |
| #158/223 | Z1–2 | 25 | 3.29 | 0.67 | 2.04 | 4.73 | #43 vs #158/223 **0.75 (< 0.001)** | #43 Z1–2 vs Z2–2 **0.58 (0.001)** | |
| #63 | Z2–2 | 20 | 4.97 | 0.95 | 3.64 | 7.19 | #63 vs #158/223 0.28 (0.234) | #158/223 Z1–2 vs Z2–2 **1.4 (< 0.001)** | |
| #43 | Z2–2 | 56 | 4.62 | 1.10 | 2.68 | 7.67 | | | |
| #158/223 | Z2–2 | 55 | 4.69 | 0.85 | 3.02 | 7.09 | #43 vs #158/223 0.07 (0.688) | | |





Figure 9. Differences in sizes of fibres of *M. venosa* (ventral and dorsal valve) in the anterior transverse sections of specimens cultured at different pH conditions. A, B: The bottom/top of the box and the band inside the box are the first/third quartiles and the median of the data respectively; ends of the whiskers represent the minimum and maximum of results. C, D: Circle point represents average data, $N_m$: number of measurement.





### 3.4 Stable isotopes

The $\delta^{13}$C and $\delta^{18}$O data were measured along the shell growth increments in the dorsal and ventral valves (Figure 10). In the *before–culturing* part, $\delta^{13}$C values varied between -2.02 ‰ and +0.45 ‰ in the control group specimens #8004 and #8005, whereas they varied between -9.24 ‰ and -0.53 ‰ in the low pH group specimens #9004, #9005 and #9006. $\delta^{18}$O values

5  varied between -2.39 ‰ and +0.21 ‰ in the control group specimens #8004 and #8005, but varied between -4.92 ‰ and +0.05 ‰ in the low pH group specimens #9004, #9005 and #9006.

In the *during–culturing* part, $\delta^{13}$C values varied between -6.80 ‰ and -1.34 ‰ in the control group specimens #8004 and #8005, whereas they varied between -27.09 ‰ and -9.69 ‰ in the low pH group specimens #9004, #9005 and #9006 (Figure 10). $\delta^{18}$O values varied between -6.80 ‰ and -1.34 ‰ in the control group specimens #8004 and #8005, but varied between -

10  6.97 ‰ and -5.29 ‰ in the low pH group specimens #9004, #9005 and #9006 (Figure 10).

A marked drop in $\delta^{13}$C and $\delta^{18}$O is recorded in the shell increments produced *during–culturing*, particularly so in the specimens grown under low pH conditions (pH$_1$ and pH$_2$), where $\delta^{13}$C values decreased to -27.09 ‰ (Figure 10).





Figure 10. Plots of $\delta^{13}$C and $\delta^{18}$O of the ventral and dorsal valves of *M. venosa* specimens along their growth axis. Different colour backgrounds represent different pH conditions during growth. When few data were available, data-points were joined by dashed lines.



## 4 Discussion

### 4.1 Microstructure and organic components relationship

Before discussing whether and how acidification may affect the microstructure of the brachiopod shell, it is important to examine the relationship between the microstructure and the amount of organic components within the shell. It has already

been stated that, in fossil and recent brachiopods, different shell microstructures have different amounts of shell organic components (Garbelli et al., 2014, 2017; Ye et al., 2018a; Casella et al., 2018).

This holds true for most rhynchonelliformean brachiopods, the primary layer of *M. venosa* consists of finely acicular and granular calcite (Williams, 1968, 1973, 1997; MacKinnon and Williams, 1974; Williams and Cusack, 2007; Casella et al., 2018). Analyses of electron back scattering diffraction show that the primary layer is produced in a thin nanocrystalline film

with higher micro–hardness and smaller–sized calcite crystallites compared to those of the secondary layer (Griesshaber et al., 2004). In addition, each spherical and small unit is coated by a mixture of organics and amorphous calcium carbonate (Cusack et al., 2010). This, *per se*, may suggest a higher amount of organic components than other shell layers, but it has never been proven. In fossils, the primary layer is likely to be diagenetically altered and luminescent (Grossman et al., 1991), suggesting that higher amounts of organic components may be present. However, this has been also ascribed to the

incorporation of magnesium into the lattice (Popov, et al., 2007; Cusack et al. 2008). A report of higher sulphur concentration in the primary layer of the brachiopod *Terebratulina retusa* may suggest the presence of a sulphur-rich organic components, but backscatter electron imaging revealed contradictory results (England et al., 2007). Cusack et al. (2008) showed that, in the same species, the sulphate concentration is higher in the primary layer than in the secondary layer. Depleted $\delta^{18}$O and $\delta^{13}$C values in the primary layer caused by kinetic effects have been reported by Carpenter and Lohmann

(1995), Auclair et al. (2003), and Parkinson et al. (2005). May this indicate a greater amount of organic components in this part of the shell? Since there is no conclusive evidence for this observation, we cannot relate the increase in thickness of the primary layer to changes in organic components within the shell. With respect to previous findings (Williams, 1966; Parkinson et al., 2005), our results show that the thickness of the primary layer of *M. venosa* is much less uniform and shows an increase with growth, which is more evident during culturing at low pH conditions. However, disturbances (stress

condition with handling before and at the start of the culturing) may cause an abrupt change in thickness.

Endopunctae, which in life are filled with mantle expansion, are widely distributed in the shell of *M. venosa* and show the superficial hexagonal close-packing pattern documented by Cowen (1966). The biological function of endopunctae is still controversially discussed, with some suggesting that generally, in living organism they serve as support and protection structures (Williams, 1956, 1997), as sensors, or as storage and respiration features (Pérez–Huerta et al., 2009). With more

endopunctae filled by mantle expansions, the amount of organic components would increase in the same volume of shell. The density of endopunctae has been related to temperature, as species living at higher temperatures have greater endopunctae density (Campbell, 1965; Foster, 1974; Peck et al., 1987; Ackerly et al., 1993). The present analyses suggest that the increase in endopunctae density may be related in part to ontogeny; it is higher in the specimen cultured at low pH



condition. This may be expected, as specimens living under low pH conditions have to up-regulate their internal pH to be able to calcify as shown for instance in corals by McCulloch et al. (2012) and Movilla et al. (2014). This would demand a higher energetic cost and thus a larger respiration/storage surface would be favourable to cope with it.

The punctal pattern detected here is different from that observed by Cross et al. (2018), who recorded no change in the

punctal density of the ventral valve of *C. inconspicua* on specimens from the last 120 years. Also different is the trend in size of the endopunctae, which measured in the dorsal valve only by Cross et al. (2018), seems to decrease in size. However, the environmental conditions of the natural ambient of 0.1 pH unit decrease and 2 °C increase over the last two decades (refs. in Cross et al., 2018) are very different from those of our culturing experiments. Further, the size of the endopunctae was measured from the dorsal valve only by Cross et al. (2018), whereas the increase in size we report was observed only from

the ventral valve of *M. venosa*.

In addition to the thickness of the primary layer and the density of the endopunctae, the size changes of the individual fibres within the fibrous secondary layer may also contribute to the variability in organic components. Most of the recent rhynchonelliformean brachiopods, and *M. venosa* in particular, possess a shell mainly made by a fibrous secondary layer (Williams, 1997; Parkinson et al. 2005; Williams and Cusack, 2007). Each fibre of this layer is secreted by the mantle and it

is ensheathed by organic membrane (e.g., Jope, 1965; Williams, 1968; MacKinnon, 1974; Williams and Cusack, 2007; Cusack et al., 2008; Casella et al., 2018; Romanin et al., 2018). Thus, with a decrease in size but within the same shell volume the surface area increases and with it the amount of organic components. Recently, the relationship between the size of fibres and the shell organic components was discussed in detail (Garbelli, 2017; Garbelli et al. 2017; Ye et al., 2018a). The main conclusion is that the smaller the calcite fibres, the higher the organic components in the shell (cf. Figure 11). Thus,

smaller fibres, and a greater endopunctae density may lead to higher organic components content per shell volume (Figure 11).





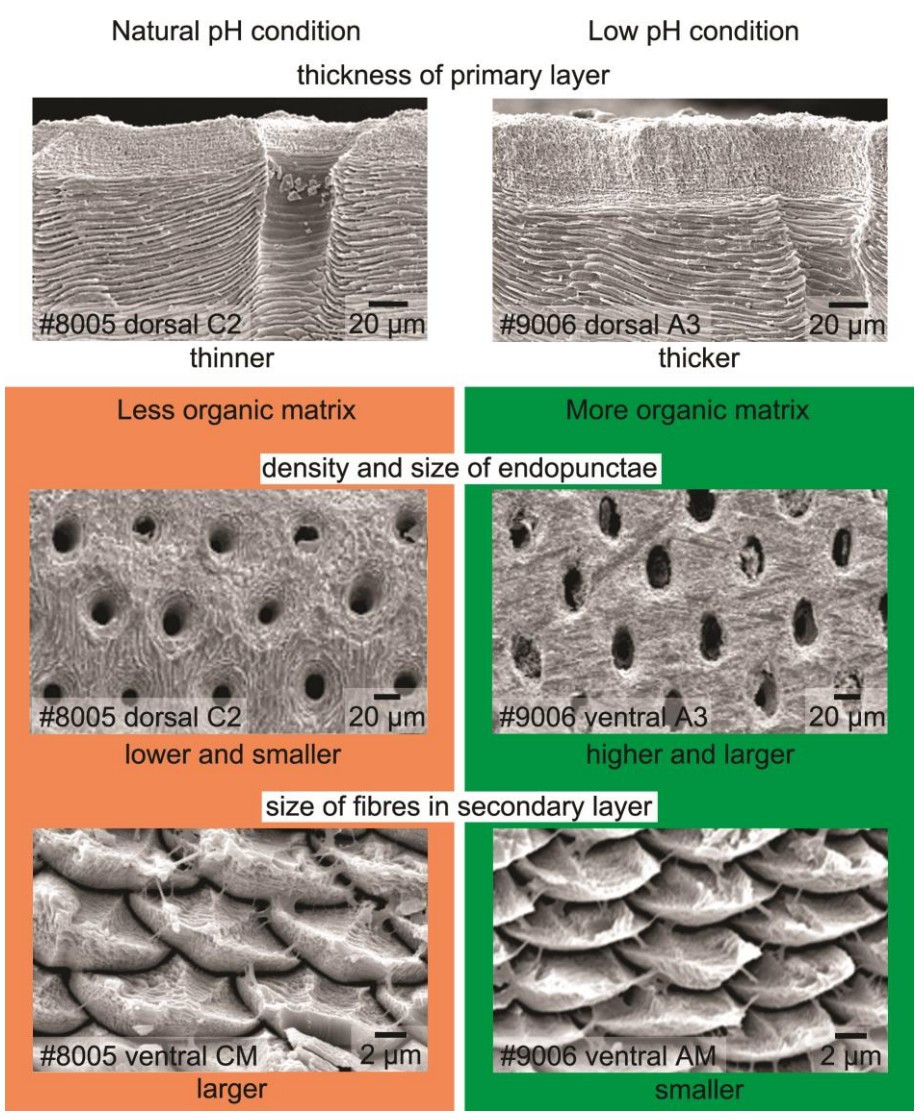

Figure 11. Relationship between the microstructure and the organic components of calcified shells of brachiopods. Position information see Figure 6 and Figure 7; CM: central middle part; AM: anterior middle part.

## 4.2 Low pH and brachiopod microstructure

Several studies tried to understand how marine carbonate shelled animals respond to ocean acidification, such as brachiopods (McClintock et al., 2009; Cross et al., 2015, 2016, 2018), bivalves (e.g., Berge et al., 2006; McClintock et al., 2009; Beniash et al., 2010; Parker et al., 2010; Melzner et al., 2011; Talmage and Gobler, 2011; Amaral et al., 2012; Hiebenthal et al., 2013; Coleman et al., 2014; Gobler et al., 2014; Milano et al., 2016), cold–water scleractinian corals (e.g., McCulloch et al., 2012; Form and Riebesell, 2012; Jantzen et al., 2013b; Büscher et al., 2017) and sea urchins (Suckling et





al., 2015) (Supplementary Table 1). The results of these studies show that, in general, seawater acidification reduces the growth rates of marine calcifiers (Michaelidis et al., 2005; Shirayama and Thornton, 2005; Berge et al., 2006; Bibby et al., 2007; Beniash et al., 2010; Nienhuis et al., 2010; Thomsen and Melzner, 2010; Fern ández–Reiriz et al., 2011; Melzner et al., 2011; Mingliang et al., 2011; Talmage and Gobler, 2011; Parker et al., 2011, 2012; Liu and He, 2012; Navarro et al., 2013;

Milano et al., 2016).

For the Antarctic brachiopod *Liothyrella uva* and the New Zealand brachiopod *Calloria inconspicua* no ocean acidification effects on shell growth were detected by Cross et al. (2015, 2016, 2018), although, shells of the former species may rapidly dissolve in acidified waters (McClintock et al., 2009). One response, however, appears to reinforce the shells of *C. inconspicua* by laying down a denser shell compared to specimens from New Zealand over the last 120 years while

subjected to a slight decrease in pH (by 0.1) and 2 ℃ increase in temperature over the last two decades (Cross et al., 2018).

The present experiment showed that growth of specimen was not affected by the low pH conditions, instead their growth was similar of that of the specimen cultured under control conditions (#9006, ~0.9 cm in the ventral valve, ~0.8 cm in the dorsal valve; 8005, ~0.5 cm in the ventral valve, ~0.4 cm in the dorsal valve). Based on the growth von Bertalanffy growth function calculated by Baumgarten et al. (2013), the expected growth increment was calculated and compared with the measured one.

Figure 12 demonstrates that the measured individual growth rates are within the range of the ones of naturally growing individuals (Fig. 12).




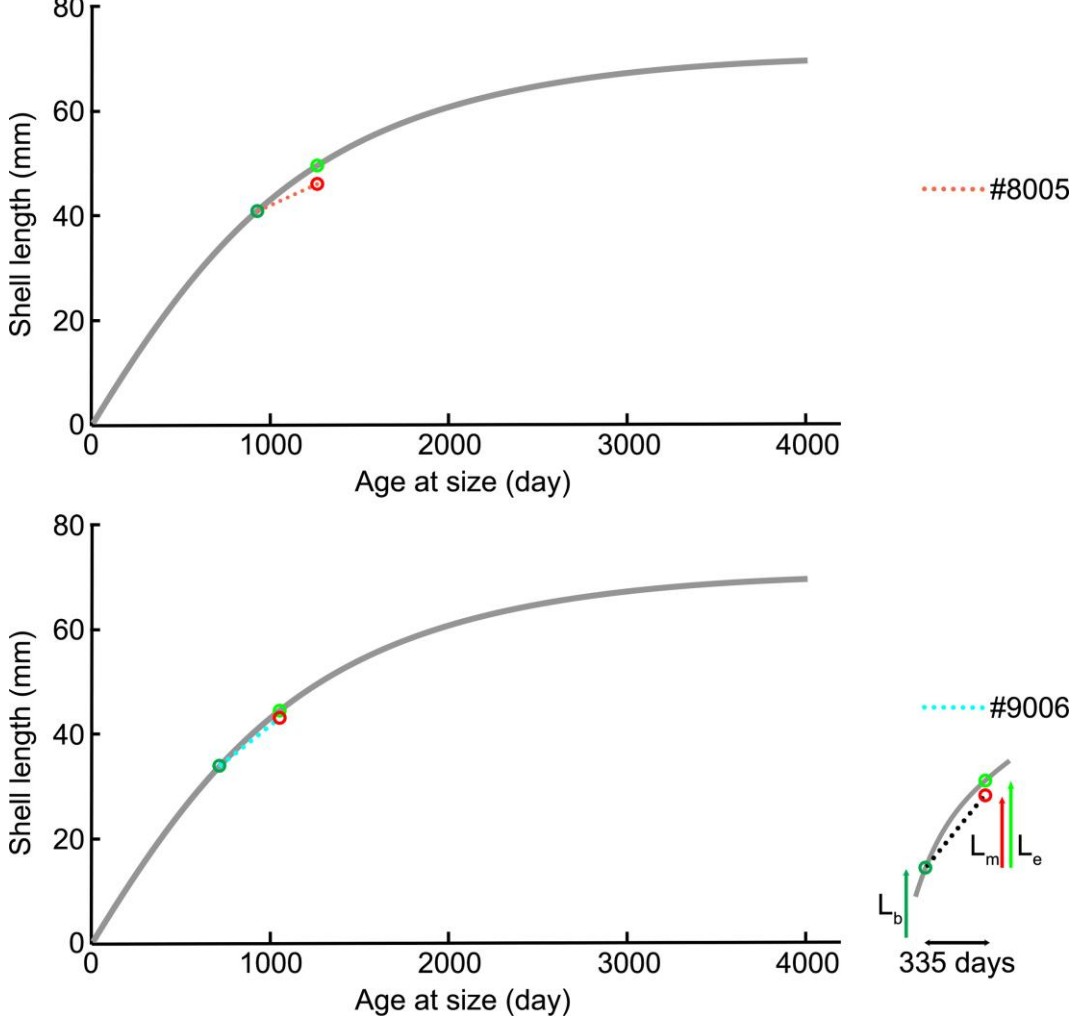

Figure 12. Projection of shell length of ventral valve on the von Bertalanffy growth function (grey line) $L_t = 71.53 [1 - e^{-0.336(t-t0)}]$, source from Baumgarten et al. (2013), $L_b$: shell length at the beginning of culturing; $L_m$: measured shell growth at the end of culturing; Le: expected shell growth.

5  A limiting factor of this assessment is the limited database, but the present observations agree with studies that show no or little impact of acidification on brachiopod growth rates (Marchant et al., 2010; Thomsen et al., 2010; Talmage and Gobler, 2011; Range et al., 2011, 2012; Dickinson et al., 2012; Fernández–Reiriz et al., 2012; Liu and He, 2012; Hiebenthal et al., 2013; Cross et al., 2015, 2016, 2018) or, even an increase in respiration, shell growth or metabolic rates after having experienced low pH condition (Wood et al., 2008; Cummings et al., 2011; Parker et al., 2012).

10  Therefore, the observations of marine calcifiers to seawater acidification in terms of growth rates are complex. The response of marine organisms to the interplay of several stressors such as low pH, lower dissolved oxygen and higher temperature is even more complex. Steckbauer et al. (2015) reported that hypoxia and increased $p$CO$_2$ could significantly reduce the





respiration rate of marine invertebrates (Anthozoa, Gastropoda, Echinoidea and Crustacea). Highest growth rate in the bivalve *Macoma balthica* [= *Limecola balthica* (Linnaeus, 1758)] was observed in a combination of low $O_2$ and high pH conditions (Jansson et al., 2015). Gobler et al. (2014) reported that juveniles of the bivalves *Argopecten irradians* (Lamarck, 1819) and *Mercenaria mercenaria* (Linnaeus, 1758) are not affected when hypoxia or acidification if applied separately, but

the growth rate decreases when juveniles are exposed to both conditions simultaneously.

To explore the effects of acidification on brachiopod biomineralization, the microstructures of the specimens cultured for 214 days (#43 and #63) at $pH_3$ and $pH_4$ and the other population cultured for 335 days (#8005 and #9006) at $pH_0$ and $pH_1$ to $pH_2$ were investigated in detail. No conclusive consideration can be carried out on the specimens cultured for 214 days, but when the culturing experiment is conducted for a time interval of 335 days, the microstructure produced by the specimen

cultured at low pH conditions ($pH_1$ to $pH_2$) is different from that produced under control condition ($pH_0$): 1) the thickness of the primary layer increases with culturing; 2) the density and size of the endopunctae are higher; and 3) the fibres of the secondary layer are smaller. Thus, the length of culturing time–in terms of months–under low pH conditions seems to be an important control factor.

This is in line with the few data available in the literature on microstructural changes during acidification. Milano et al.

(2016) reported no significant difference in the prismatic microstructure of the cockle *Cerastoderma edule* when cultured under low pH conditions for about 2 months, except for dissolution of ontogenetically younger parts of the shell. Similarly, a study by Stemmer et al. (2013) on the clam *Arctica islandica* revealed that there was no effect on the shape and size of the crystals in the homogeneous microstructure after three months of culturing at low pH (Supplementary Table 1). However, the experiments conducted by Fitzer et al. (2014a, b) for six months on the blue mussel *Mytilus edulis* showed that the

animals exposed to low pH and high $p$CO$_2$ tend to produce less organised, disorientated calcite crystals and an unordered layer structure.

Thus, in bivalves, similarly to our observations, the duration of culturing may be crucial in recording significant effects. The present results lend support to the microstructure variation observed in brachiopods during the end-Permian extinction event and concomitant ocean acidification (Garbelli et al., 2017). During this event, both Strophomenata and Rhynchonellata

produced more organic rich shells to cope with the long term and protracted seawater acidification effects (Garbelli et al., 2017).

### 4.3 Stable isotope variation at low pH condition

Brachiopod shells are the archives commonly used for deep-time paleoenvironmental reconstructions as they potentially record the original geochemical composition of the seawater they lived in (Grossman et al., 1993; Banner and Kaufman,

1994; Mii and Grossman, 1994; Mii et al., 2001; Brand et al., 2003, 2011, 2016; Jurikova et al., in review). Several studies suggest that oxygen and carbon isotopic compositions of the secondary layer of brachiopod shells–especially its innermost part–tend to be in equilibrium with the seawater chemistry (e.g., Popp et al., 1986; Carpenter and Lohmann, 1995; Parkinson et al., 2005; Brand et al., 2013, 2015, 2016; Takayanagi et al., 2013; Yamamoto et al., 2013).



The measured $\delta^{13}C$ and $\delta^{18}O$ values of the secondary layer produced during growth in the natural environment (Figure 10) are similar to previous results from the shells of *M. venosa* (Penman et al., 2013; Ullmann et al., 2017; Romanin et al. 2018). Furthermore, the present results show that there are no significant differences in $\delta^{13}C$ and $\delta^{18}O$ values between the dorsal and ventral valves (*p*–values in $\delta^{13}C$ and $\delta^{18}O$ of #8005 are 0.437 and 0.491 respectively, *p*-values in $\delta^{13}C$ and $\delta^{18}O$ of #9006 are

0.862 and 0.910 respectively), which are in agreement with previous findings (e.g., Parkinson et al., 2005; Brand et al., 2015; Romanin et al., 2018).

Generally, in the naturally grown shell *before–culturing*, $\delta^{13}C$ and $\delta^{18}O$ values are relatively stable along the ontogenetic direction, except for depleted values at approximately mid–shell length in both #8005 and #9006. In particular, in #9006, in this part of the shell values drop to about -6 ‰ for $\delta^{13}C$ and -2 ‰ for $\delta^{18}O$ values (Figure 10). We exclude that this drop may

be produced by shell material added later, during the *during–culturing* shell thickening, as the samples were taken from the mid-shell layer and not from the shell interior. Also, negative isotope excursions of similar magnitude were recorded in *M. venosa* specimens from the South America shelf by Ullmann et al. (2017) and Romanin et al. (2018). Ullmann et al. (2017) implied that these variable $\delta^{13}C$ and $\delta^{18}O$ values indicate isotope disequilibrium with ambient waters in Terebratellids. In contrast, Romanin et al. (2018), who also analysed specimens collected from Comau Fjord, attributed the negative isotope

excursion to environmental perturbations, in particular, to changes in seawater productivity and temperature, and/or to anthropogenic activities. Negative shifts in both, $\delta^{13}C$ and $\delta^{18}O$ values during ontogeny have also been observed also in in the brachiopod *Terebratella dorsata*, which co–occurs with *M. venosa* and have been explained by the effect of resorption in corresponding muscle scars (Carpenter and Lohmann, 1995). Here, we follow the interpretation of Romanin et al. (2018) to explain the mid–shell excursion observed in our specimens.

In our experiments, oxygen isotope compositions record only a minor depletion *during–culturing* at different pH conditions, a depletion which is in isotope equilibrium with $\delta^{18}O_{H2O}$ during the cultivation process [$\delta^{18}O$ (VSMOW): -6.88 ‰ for the low pH conditions and -6.69 ‰ for the control conditions].

However, a sharp drop in $\delta^{13}C$ values was observed in the secondary layer produced *during–culturing* under low pH conditions. $\delta^{13}C$ values are depleted by more than 20 ‰ in the specimens cultured at low pH conditions (pH$_1$ and pH$_2$; #9004,

#9005 and #9006) (Figure 10 and Appendix), whereas the depletion is lower by just a few per mil (ca. 0.9–1.2 ‰) in the control specimens (pH$_0$; #8004 and #8005). Our results are comparable with those of other studies. Hahn et al. (2014) reported a decreasing trend of about 10 ‰ in $\delta^{13}C$ values in the blue mussel *Mytilus edulis* when exposed to seawater conditions of pH 8.03 ($pCO_2$ 612 µatm) and pH of 7.21 ($pCO_2$ 4237 µatm). In corals, a species–specific $\delta^{13}C$ response to high $pCO_2$ conditions was reported by Krief et al. (2010) of more negative 2.3‰ and 1.5‰ $\delta^{13}C$ values in *Porites* sp. after

14 months of culturing at low pH conditions (pH 7.49, $pCO_2$ 1908 µatm and 7.19 $pCO_2$, 3976 µatm), whereas no significant difference was found in other coral species, such as *Stylophora pistillata* Esper, 1797. Given that the $\delta^{13}C_{DIC}$ in the water during the cultivation process of our specimens was low ($\delta^{13}C$ VPDB: -23.63 ‰ for the low pH conditions and -2.03 ‰ for the control conditions, which corresponds to the pH$_2$ phase), we can conclude that the negative shift is probably related to the




C–source in the carbon dioxide gas used in culture setup. This was also previously suggested by McConnaughey et al. (2008), Poulain et al. (2010), and Hahn et al. (2014).

The $\delta^{13}$C and $\delta^{18}$O composition of *M. venosa* shells produced *during-culturing* is summarized in Table 11. The fractionation of carbon and oxygen isotopes between the seawater and calcite phase, is defined as $\Delta^{13}C_{cal–DIC}$ or $\Delta^{18}O_{cal–sw} = 1000 \times \ln\alpha_{cal–DIC/sw}$, where $\alpha_{cal–DIC/sw} = [^{13}C/^{12}C]_{cal} / [^{13}C/^{12}C]_{DIC}$ or $[^{18}O/^{16}O]_{cal} / [^{18}O/^{16}O]_{sw}$, respectively.

Table 11. Carbon and oxygen fractionation in our cultured *M. venosa* specimens.

| Sample #ID | Treatment | Avg. $\Delta^{13}C_{cal-DIC}$ | Avg. $\Delta^{18}O_{cal-sw}$ |
| --- | --- | --- | --- |
| #8004 | Control | -4.06 | 29.99 |
| #9005 | Acidification pH$_2$ | -1.21 | 30.92 |
| #9004 | Acidification pH$_2$ | -2.23 | 30.70 |

For carbon isotopes, we observe a variability in $\Delta^{13}C_{cal-DIC}$ between the different specimens, and it is inconclusive if this is linked to an ontogenetic variations or to differences between the individuals. It appears that there is about 2 ‰ difference between the control specimen and samples from the acidification (pH$_2$) treatments, with the last one being, strikingly, more close to equilibrium with seawater DIC. Possibly, this illustrates the variability in kinetic effects, but may also be linked to a more changeable $\delta^{13}C_{DIC}$ in the control treatment. More measurements are however needed to fully answer this.

The $\Delta^{18}O_{cal–sw}$ values show little variability between the specimens, with similar fractionation to that of inorganically precipitated carbonates (Watkins et al., 2013; around 30 per mille at similar seawater conditions). In addition, alike in the experiment of Watkins et al. (2013), we observe a slight trend in pH, with higher $\Delta^{18}O_{cal–sw}$ at lower pH. This suggests that the $\Delta^{18}O_{cal–sw}$ behaviour of *M. venosa* is not far from that of inorganic calcite.

Thus, we think that large part of the secondary layer isotope record may reflect the environmental conditions supporting the interpretation of brachiopod shells as good archives of geochemical proxies, even when stressed by ocean acidification.

## 5 Conclusions

This study combined the analysis of shell microstructure and stable isotope geochemistry on brachiopods cultured at low pH conditions for different time intervals, and suggests the following conclusions.

In brachiopod specimens cultured for a period of 11 months, the microstructure produced by the specimen cultured at low pH is different from that produced under control conditions. In particular, the microstructure produced at low pH tends to be more organic components-rich. A result that lends strong support to the brachiopod microstructure variations observed in the fossil record and related to the effect of ocean acidification.

Low pH conditions on brachiopod shell parts precipitated during culture conditions for about one year record a change in the microstructure but not in the growth rate.

$\delta^{13}$C and $\delta^{18}$O values are rather constant during growth but experience a sharp drop during culturing. In particular, the $\delta^{13}$C values of specimens cultured for one year at low pH conditions dropped abruptly. This was related to the source of carbon dioxide gas used in the culture setup



Brachiopods are thus faithful recorders of the ambient O and C isotope composition, even when stressed by environmental perturbations such as ocean acidification.

The present observations are invaluable in using specific proxies and shell morphologic features for studying ocean acidification events and changes in atmospheric $CO_2$ contents in the geologic past.

**Acknowledgements:**

This project has received funding from the European Union's Horizon 2020 research and innovation programme under the Marie Skłodowska–Curie grant agreement No 643084 (BASE–LiNE Earth). We would like to express our thanks to the scientific divers and staff of the Huinay Field Station, Chile. Vreni Haussermann is thanked for logist support. We thank Nina Hörner and Ulrike Holtz for help during culturing of the brachiopods at AWI, Dirk Nürnberg at GEOMAR for assistance with isotope analyses and Curzio Malinverno and Agostino Rizzi for technical support at Universit à di Milano.

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

Table 1. Culturing, dissolution experiments and natural variation on several brachiopod species and shells.

| Species N (number of sample) | Growth Parameters | Shell repair/Microstructure/Oxygen consumption/Dissolution of shell/Microstructure | Method & Material | Environment/conditions T=Temperature (℃) S=Salinity (PSU) $p$CO$_2$ (µatm) | Duration of experiment | Source |
|---|---|---|---|---|---|---|
| *Calloria inconspicua* (Sowerby, 1846) N = 123 | 1) >3 mm in length undamaged individuals were not affected by lower pH; 2) <3 mm in length undamaged individuals grew faster at pH 7.62 than the control conditions | Not affected by lower pH (>80% of all damaged individuals repaired after 12 weeks) | Culture experiment | a) pH 8.16, T 16.5, S 33.9, $p$CO$_2$ 465, Ω calcite 3.5 b) pH 7.79, T 16.9, S 33.9, $p$CO$_2$ 1130, Ω calcite 1.6 c) pH 7.62, T 16.6, S 33.9, $p$CO$_2$ 1536, Ω calcite 1.3 | 12 weeks | Cross et al., 2016 |
| *Calloria inconspicua* (Sowerby, 1846) N = 389 (adults) | | Punctae width decreased by 8.26%, shell density increased by 3.43%, no change in shell morphology, punctae density, shell thickness, and shell elemental composition (Ca, Mg, Na, Sr and P) | One specimen collected every decade from one locality | Last two decades pH reduced 0.1 unit Temperature varied from 10.7–13.0 ℃ $p$CO$_2$ varied from 320-400 Salinity and Ω of calcite not provided | 120–year record | Cross et al., 2018 |
| *Liothyrella uva* (Broderip, 1833) N = 156 | Not affected by lower pH | Not affected by either low pH conditions or temperature. (>83% of individuals repaired after 7 months) | Culture experiment | a) pH 7.98, T -0.3, S 35, $p$CO$_2$ 417, Ω calcite 1.20 b) pH 8.05, T 1.7, S 35, $p$CO$_2$ 365, Ω calcite 1.49 c) pH 7.75, T 1.9, S 35, $p$CO$_2$ 725, Ω calcite 0.78 d) pH 7.54, T 2.2, S 35, $p$CO$_2$ 1221, Ω calcite 0.50 | 7 months | Cross et al., 2015 |
| *Liothyrella uva* (Broderip, 1833) N$_{post-mortem}$ = 5 | Not applicable | Higher dissolution in gastropods and brachiopods at lower pH after 14 days | Empty shells | a) pH 7.4, T 4, S 35, Ω calcite 0.74 b) pH 8.2, T 4, S 35, Ω calcite 4.22 $p$CO$_2$ Not provided | 14 to 63 days | McClintock et al., 2009 |

Table 2. Specimens of *M. venosa* sampled from Comau Fjord, Chile, and natural and experimental culturing conditions.




| Sample ID | Sample locality at Comau Fjord (Chile)[1] | Sample seawater conditions[2] | Date of collection | Length of ventral valve (mm) | Duration of experiment | Experimental conditions |
|---|---|---|---|---|---|---|
| #43 | Lilliguapi | pH: ~7.9<br>T: ~13<br>S: ~32<br>D: 20 | Feb. 2012 | 37 | 214 days[3] | $p$CO$_2$: 1391, pH: 7.66 ±0.04<br>T: 11.62 ±0.54, S: 32.58<br>Ωcal: 1.97 |
| #63 | Lilliguapi | pH: ~7.9<br>T: ~13<br>S: ~32<br>D: 20 | Feb. 2012 | 23 | 214 days[3] | $p$CO$_2$: 2611, pH: 7.44 ±0.08<br>T: 11.69 ±0.45, S: 32.65<br>Ωcal: 1.37 |
| #158 | Huinay Dock | pH: ~7.9<br>T: ~13<br>S: ~32<br>D: 20 | Dec. 2011 | 36 | no | |
| #223 | Cahuelmó | pH: ~7.9<br>T: ~13<br>S: ~32<br>D: 23 | Feb. 2012 | 30 | no | |
| #8004 | Comau Fjord | pH: ~7.9<br>T: ~13<br>S: ~32<br>D: 21 | Apr. 2016 | 31 | 335 days[4] | $p$CO$_2$: 600 ,pH: 8.00–8.15 ± 0.05<br>T: ~10, S: 30, Ωcal: 2.0–3.5 |
| #8005 | Comau Fjord | pH: ~7.9<br>T: ~13<br>S: ~32<br>D: 21 | Apr. 2016 | 46 | 335 days[4] | $p$CO$_2$: 600, pH: 8.00–8.15 ± 0.05<br>T: ~10, S: 30, Ωcal: 2.0–3.5 |
| #9004 | Comau Fjord | pH: ~7.9<br>T: ~13<br>S: ~32<br>D: 21 | Apr. 2016 | 41 | 335 days[4] | $p$CO$_2$: 2000–4000[5]<br>pH: 7.60 ±0.05 to 7.35 ± 0.054<br>T: ~10, S: 30, Ωcal: 0.6–1.1 |
| #9005 | Comau Fjord | pH: ~7.9<br>T: ~13<br>S: ~32<br>D: 21 | Apr. 2016 | 25 | 335 days[4] | $p$CO$_2$: 2000–4000[5]<br>pH: 7.60 ±0.05 to 7.35 ± 0.054<br>T: ~10, S: 30, Ωcal: 0.6–1.1 |
| #9006 | Comau Fjord | pH: ~7.9<br>T: ~13<br>S: ~32<br>D: 21 | Apr. 2016 | 43 | 335 days[4] | $p$CO$_2$: 2000–4000[5]<br>pH: 7.60 ±0.05 to 7.35 ± 0.054<br>T: ~10, S: 30, Ωcal: 0.6–1.1 |

Note: D: Depth (m), T: temperature (°C), S: salinity (PSU – practical salinity units), $p$CO$_2$ (µatm).

[1] Cahuelmó 42°15'23" S, 72°26'42" W, Cross–Huinay 42°23'28" S, 72°27'27" W, Jetty (Huinay Dock) 42°22'47" S, 72°24'56" W, Lilliguapy 42°9'43" S, 72°35'55" W, samples #8004, #8005, #9004, #9005, #9006 were harvested from three sites in Comau Fjord (Cross–Huinay, Jetty, and Liliguapy), Chilean Patagonia

[2] Reference: Laudien et al. (2014) and Jantzen et al. (2017)




[3]Culture experiments conducted at the Alfred–Wegener–Institut Helmholtz–Zentrum für Polar–und Meeresforschung, Bremerhaven, Germany

[4]Culture experiments conducted at GEOMAR Helmholtz–Zentrum für Ozeanforschung Kiel, Germany (Jurikova et al., in review)

[5]$CO_2$ concentration was changed during the experiment: from 4 August 2016 to 18 April 2017 at 2000 µatm and from 18 April 2017 till 5 July 2017 at 4000 µatm

Table 3. Culture and sensor systems for *M. venosa* specimens (#43, #63, #8004, #8005, #9004, #9005 and #9006). Operated under controlled experimental settings in a climate control laboratory at the Alfred–Wegener–Institut Helmholtz–Zentrum für Polar–und Meeresforschung, Bremerhaven, Germany and at GEOMAR Helmholtz–Zentrum für Ozeanforschung Kiel, Germany.

| | Culture system at AWI | Automated sensor Systems at AWI | Culture system at GEOMAR | Automated sensor Systems at GEOMAR |
|---|---|---|---|---|
| | Aquarium (150 L/each pH treatment) | | Aquarium (150 L/each pH treatment) | |
| | Supplied from a reservoir tank (twice a week 20 % water was replaced) | | Supplied from a reservoir tank (twice a month 10 % water was replaced) | |
| Temperature | Controlled in temperature constant room | | Controlled using heaters or coolers | Temperature Sensor Pond |
| $p$CO$_2$ | Bubbling of CO$_2$ pH 7.66 ± 0.04, pH 7.44 ± 0.08 | COMPORT, Dennerle, Vinningen; IKS aquastar Aquarium computer V2.xx with Aquapilot 2011 | Bubbling of CO$_2$ enriched air | CONTROS HydroC® underwater CO$_2$ sensor |
| Salinity | Mixing Reef commercial sea–salt (until October: Aqua Medic, Bissendorf, Germany, thereafter Dupla Marin Reef Salt, Dohse Aquaristik, Grafschaft–Gelsdorf, Germany) with deionized water (Atkinson and Bingman, 1998) | Conductivity Electrode | Mixing Tropic Marin Pro–Reef commercial sea-salt with deionized water (Atkinson and Bingman, 1998) | Conductivity Electrode |
| Filtering | Biofilter, protein skimmer and UV sterilizer | | Biofilter, protein skimmer and UV | |



| | | |
|---|---|---|
| | | sterilizer |
| Food | Regularly fed (typically 5 times per week) with Dupla Rin, Coral Food, Reef Pearls 5–200 µm, alive *Thalassiosira weissflogii,* and 1d old nauplii of *Artemia salina* | Regularly fed (typically 5 times per week) with *Rhodomonas baltica* |
| Substrate | Sabia Corallina, 7–8mm, Dohse Aquaristik, Grafschaft–Gelsdorf, Germany | No |

Table 4. Total shell length of three specimens of *M. venosa* before, during and at the end of the in vitro culturing.

| Sample | Initial anterior–posterior length (mm) | Length–Duration (a) 257 Days (mm) | Length–Duration (b) 78 Days (mm) |
|---|---|---|---|
| #8004 ventral | 31 | 14 ($pH_0$) | 1.6 ($pH_0$) |
| #8005 ventral | 46 | 5 ($pH_0$) | <1 ($pH_0$) |
| #8005 dorsal | 41 | 4 ($pH_0$) | <1 ($pH_0$) |
| #9004 ventral | 41 | 13 ($pH_1$) | 1.2 ($pH_2$) |
| #9005 ventral | 25 | 12 ($pH_1$) | 1.8 ($pH_2$) |
| #9006 ventral | 43 | 9 ($pH_1$) | <1 ($pH_2$) |
| #9006 dorsal | 38 | 8 ($pH_1$) | <1 ($pH_2$) |

Note: (a) Culturing from 4 August 2016 to 18 April 2017; (b) Culturing from 18 April 2017 to 5 July 2017; $pH_0 = 8.00$–$8.14$, $pH_1 = 7.60$, $pH_2 = 7.35$.

Table 5. Statistical comparison of thickness of the primary layer (µm) along the ontogenetic direction of both valves of specimens #8005 and #9006. ①: Specific positions see Figure 6. N = number of measurement. Significant values ($p$–value $\leq$ 0.05) are marked in bold style.

| Sample | Position① | N | Mean | STD | Min | Max | $p$-values | $p$-values |
|---|---|---|---|---|---|---|---|---|
| #8005 dorsal | P | 4 | 11.82 | 1.05 | 10.55 | 13.02 | | |
| | C1 | 8 | 11.40 | 2.29 | 8.50 | 15.05 | P vs C1 0.755 | |
| | C2 | 10 | 28.99 | 4.79 | 22.15 | 36.65 | C1 vs C2 < 0.001 | #8005DP vs 9006DP 0.120 |
| | A1 | 8 | 24.36 | 2.52 | 19.80 | 27.06 | C2 vs A1 0.033 | |
| | A2 | 7 | 24.83 | 2.15 | 21.67 | 27.94 | A1 vs A2 0.726 | #8005DC1 vs #9006DC1 < 0.001 |
| | A3 | 1 | 21.77 | NA | NA | NA | A2 vs A3 NA | |
| | | | | | | | | #8005DC2 vs #9006DC2 < 0.001 |
| #8005 ventral | P | 2 | 17.64 | 2.36 | 15.28 | 20 | P vs C1 NA | #8005DA1 vs #9006DA1 0.088 |
| | C1 | 6 | 13.68 | 3.96 | 8.50 | 20.52 | C1 vs C2 < 0.001 | |



| Sample | Zone | N | Mean | STD | Min | Max | *p*-values | *p*-values |
|---|---|---|---|---|---|---|---|---|
|  | C2 | 8 | 47.57 | 2.49 | 42.55 | 50.27 | C2 vs A1 0.028 | #8005DA2 vs #9006DA2 0.101 |
|  | A1 | 8 | 44.18 | 2.68 | 38.33 | 47.98 | A1 vs A2 0.289 | #8005DA3 vs #9006DA3 NA |
|  | A2 | 6 | 42.09 | 3.85 | 36.06 | 45.04 | A2 vs A3 0.017 | #8005VP vs #9006VP NA |
|  | A3 | 4 | 34.09 | 3.51 | 29.63 | 37.52 |  | #8005VC1 vs #9006VC1 0.123 |
| #9006 dorsal | P | 7 | 9.08 | 2.77 | 5.56 | 14.64 |  | #8005VC2 vs #9006VC2 0.194 |
|  | C1 | 10 | 18.78 | 2.04 | 16.90 | 22.50 | P vs C1 < 0.001 |  |
|  | C2 | 11 | 46.91 | 5.22 | 35.92 | 55.86 | C1 vs C2 < 0.001 | #8005VA1 vs #9006VA1 < 0.001 |
|  | A1 | 10 | 28.83 | 6.65 | 19.04 | 39.93 | C2 vs A1 < 0.001 |  |
|  | A2 | 8 | 28.06 | 4.03 | 22.50 | 36.69 | A1 vs A2 0.779 | #8005VA2 vs #9006VA2 0.007 |
|  | A3 | 4 | 32.84 | 3.55 | 29.10 | 38.65 | A2 vs A3 0.096 | #8005VA3 vs #9006VA3 0.027 |
| #9006 ventral | P | 7 | 9.78 | 1.72 | 6.07 | 11.79 |  |  |
|  | C1 | 9 | 16.75 | 2.77 | 12.61 | 21.29 | P vs C1 < 0.001 |  |
|  | C2 | 12 | 45.16 | 4.34 | 35.09 | 51.40 | C1 vs C2 < 0.001 |  |
|  | A1 | 11 | 36.92 | 3.82 | 26.62 | 42.54 | C2 vs A1 < 0.001 |  |
|  | A2 | 4 | 32.95 | 2.91 | 30.84 | 37.95 | A1 vs A2 0.102 |  |
|  | A3 | 5 | 40.55 | 2.63 | 37.78 | 45.23 | A2 vs A3 0.008 |  |

Table 6. Statistical comparison of the number of endopunctae (per mm$^2$) on both valves of #8005 and #9006. ①: Specific zones see to Figure 7. N = number of measurement.

| Sample | Zone① | N | Mean | STD | Min | Max |
|---|---|---|---|---|---|---|
| #8005 dorsal | C2 | 3 | 236 | 8.6 | 225 | 250 |
|  | A1 | 1 | 280 | NA | NA | NA |
|  | A2 | 2 | 244 | 12.5 | 231 | 256 |
|  | A3 | 2 | 281 | 14 | 267 | 295 |
| #8005 ventral | C2 | 2 | 225 | 1.6 | 223 | 227 |
|  | A1 | 1 | 242 | NA | NA | NA |
|  | A2 | 2 | 241 | 5.5 | 236 | 247 |
|  | A3 | 2 | 269 | 6.3 | 263 | 275 |
| #9006 dorsal | C2 | 2 | 221 | 8.6 | 213 | 230 |
|  | A1 | 1 | 269 | NA | NA | NA |
|  | A2 | 2 | 250 | 3.1 | 247 | 253 |
|  | A3 | 2 | 266 | 3.1 | 263 | 269 |
| #9006 ventral | C2 | 2 | 186 | 3.1 | 183 | 189 |
|  | A1 | 1 | 234 | NA | NA | NA |
|  | A2 | 2 | 230 | 4.7 | 225 | 234 |
|  | A3 | 2 | 308 | 1.6 | 306 | 309 |

5  Table 7. Statistical comparison of the diameter (max) (µm) of endopunctae on both valves of #8005 and #9006. ①: Specific zones see Figure 7. N = number of measurement. Significant values (*p*-value ≤ 0.05) are marked in bold style.

| Sample | Zone① | N | Mean | STD | Min | Max | *p*-values | *p*-values |
|---|---|---|---|---|---|---|---|---|
| #8005D | C2 | 21 | 36.04 | 1.78 | 33.2 | 40.4 | C2 vs A1 **< 0.001** | #8005DC2 vs #9006DC2 |
|  | A1 | 10 | 28.36 | 2.33 | 25 | 32.1 |  |  |



| Sample | Position | N | Mean | STD | Min | Max | p-values | |
|---|---|---|---|---|---|---|---|---|
| | A2 | 15 | 18.77 | 1.10 | 17 | 21.1 | A1 vs A2 < **0.001** | **0.025** |
| | A3 | 13 | 21.8 | 2.53 | 18.2 | 26.2 | A2 vs A3 **0.001** | |
| #8005V | C2 | 11 | 17.07 | 1.42 | 13.6 | 18.9 | C2 vs A1 < **0.001** | #8005DA1 vs #9006DA1 < **0.001** |
| | A1 | 13 | 20.88 | 2.22 | 17.1 | 24.3 | A1 vs A2 **0.007** | |
| | A2 | 12 | 18.74 | 0.84 | 18 | 20.9 | A2 vs A3 < **0.001** | |
| | A3 | 14 | 26.83 | 2.83 | 23 | 33.1 | | #8005DA2 vs #9006DA2 < **0.001** |
| #9006D | C2 | 12 | 32.54 | 4.39 | 26.2 | 40 | C2 vs A1 0.178 | |
| | A1 | 13 | 34.63 | 2.33 | 29 | 37.2 | A1 vs A2 **0.012** | |
| | A2 | 11 | 32.02 | 2.12 | 27.5 | 36.1 | A2 vs A3 **0.005** | #8005DA3 vs #9006DA3 < **0.001** |
| | A3 | 19 | 28.75 | 3.51 | 23 | 34.4 | | |
| #9006V | C2 | 13 | 29.98 | 2.04 | 24.3 | 33 | C2 vs A1 < **0.001** | #8005VC2 vs #9006VC2 < **0.001** |
| | A1 | 12 | 38.66 | 2.41 | 35.5 | 42.6 | A1 vs A2 < **0.001** | |
| | A2 | 14 | 32.51 | 4.08 | 25.3 | 40.3 | A2 vs A3 0.516 | |
| | A3 | 24 | 33.70 | 5.82 | 22 | 44.3 | | |

Table 8. Statistical comparison of fibres size and shape data of the posterior external vs central middle parts of both the ventral valve and the dorsal valve. NC: non-cultured samples #158, #223; CU: cultured samples #43, #63, #8005, #9006; Vpe: ventral posterior external, Vcm: ventral central middle, Dpe: dorsal posterior external, Dcm: dorsal central middle, N:
5    number of measurement. Significant values ($p$–value ≤ 0.05) are marked in bold style.

| Sample | Position | N | Mean | STD | Min | Max | p-values |
|---|---|---|---|---|---|---|---|
| Feret diameter (max) (µm): | | | | | | | |
| NC | Vpe | 7 | 13.79 | 3.22 | 6.97 | 17.33 | NC Vpe vs CU Vpe 0.486 |
| CU | Vpe | 26 | 12.47 | 6.58 | 4.59 | 24.78 | NC Vcm vs CU Vcm 0.633 |
| NC | Vcm | 32 | 12.98 | 2.91 | 7.09 | 20.61 | NC Vpe vs NC Vcm 0.533 |
| CU | Vcm | 65 | 13.24 | 2.15 | 8.68 | 18.84 | CU Vpe vs CU Vcm 0.572 |
| NC | Dpe | 8 | 18.36 | 4.22 | 13.30 | 24.46 | NC Vpe vs CU Vpe  0.486 |
| CU | Dpe | 12 | 10.78 | 6.36 | 4.85 | 22.29 | NC Vcm vs CU Vcm 0.633 |
| NC | Dcm | 12 | 12.14 | 1.13 | 9.84 | 14.42 | NC Vpe vs NC Vcm 0.533 |
| CU | Dcm | 46 | 12.51 | 1.57 | 9.45 | 15.89 | CU Vpe vs CU Vcm 0.572 |
| Roundness: | | | | | | | |
| NC | Vpe | 7 | 0.308 | 0.077 | 0.239 | 0.475 | NC Vpe vs CU Vpe  0.717 |
| CU | Vpe | 26 | 0.296 | 0.074 | 0.172 | 0.446 | NC Vcm vs CU Vcm 0.396 |
| NC | Vcm | 29 | 0.282 | 0.051 | 0.179 | 0.389 | NC Vpe vs NC Vcm 0.296 |
| CU | Vcm | 65 | 0.272 | 0.051 | 0.180 | 0.421 | CU Vpe vs CU Vcm 0.146 |
| NC | Dpe | 8 | 0.220 | 0.034 | 0.169 | 0.268 | NC Dpe vs CU Dpe  **0.003** |




| | | | | | | | |
|---|---|---|---|---|---|---|---|
| CU | Dpe | 12 | 0.337 | 0.100 | 0.155 | 0.500 | NC Dcm vs CU Dcm **0.028** |
| NC | Dcm | 11 | 0.311 | 0.068 | 0.192 | 0.416 | NC Dpe vs NC Dcm **0.005** |
| CU | Dcm | 48 | 0.269 | 0.051 | 0.162 | 0.378 | CU Dpe vs CU Dcm **0.048** |
| Convexity: | | | | | | | |
| NC | Vpe | 7 | 0.985 | 0.004 | 0.979 | 0.991 | NC Vpe vs CU Vpe  0.309 |
| CU | Vpe | 26 | 0.982 | 0.008 | 0.968 | 0.999 | NC Vcm vs CU Vcm 0.655 |
| NC | Vcm | 32 | 0.984 | 0.005 | 0.975 | 1.000 | NC Vpe vs NC Vcm 0.823 |
| CU | Vcm | 62 | 0.984 | 0.008 | 0.965 | 1.008 | CU Vpe vs CU Vcm 0.257 |
| NC | Dpe | 8 | 0.987 | 0.006 | 0.979 | 0.998 | NC Dpe vs CU Dpe  0.604 |
| CU | Dpe | 11 | 0.985 | 0.007 | 0.973 | 0.998 | NC Dcm vs CU Dcm 0.273 |
| NC | Dcm | 12 | 0.984 | 0.008 | 0.973 | 1.000 | NC Dpe vs NC Dcm 0.543 |
| CU | Dcm | 48 | 0.982 | 0.008 | 0.967 | 1.001 | CU Dpe vs CU Dcm 0.207 |

Table 9. Statistical comparison of fibres size and shape data of the posterior external vs central middle area for #8005 and #9006, considering both valves together. pe: posterior external, cm: central middle, N: number of measurement. Significant values ($p$–value ≤ 0.05) are marked in bold style.

| Sample | Position | N | Mean | STD | Min | Max | *p*-values |
|---|---|---|---|---|---|---|---|
| Feret diameter (max) (μm): | | | | | | | |
| #8005 | pe | 10 | 7.92 | 3.30 | 4.85 | 14.97 | #8005 pe vs #9006 pe |
| #8005 | cm | 36 | 12.29 | 1.64 | 9.63 | 15.89 | 0.265 |
| #9006 | pe | 10 | 6.45 | 1.95 | 4.59 | 11.41 | #8005 cm vs #9006 cm |
| | | | | | | | 0.171 |
| | | | | | | | #8005 pe vs #8005 cm |
| #9006 | cm | 25 | 11.73 | 1.39 | 8.68 | 15.24 | **0.003** |
| | | | | | | | #9006 pe vs #9006 cm |
| | | | | | | | **< 0.001** |
| Roundness: | | | | | | | |
| #8005 | pe | 10 | 0.33 | 0.097 | 0.155 | 0.446 | #8005 pe vs #9006 pe |
| #8005 | cm | 36 | 0.25 | 0.045 | 0.162 | 0.374 | 0.547 |
| #9006 | pe | 10 | 0.35 | 0.079 | 0.232 | 0.500 | #8005 cm vs #9006 cm |
| | | | | | | | **0.012** |
| | | | | | | | #8005 pe vs #8005 cm |
| #9006 | cm | 26 | 0.28 | 0.043 | 0.195 | 0.369 | **0.040** |
| | | | | | | | #9006 pe vs #9006 cm |
| | | | | | | | **0.022** |
| Convexity: | | | | | | | |



| Sample | position | N | Mean | STD | Min | Max | | |
|---|---|---|---|---|---|---|---|---|
| #8005 | pe | 10 | 0.981 | 0.007 | 0.973 | 0.994 | #8005 pe vs #9006 pe | 0.308 |
| #8005 | cm | 35 | 0.982 | 0.008 | 0.968 | 1.001 | #8005 cm vs #9006 cm | 0.277 |
| #9006 | pe | 9 | 0.985 | 0.007 | 0.975 | 0.999 | #8005 pe vs #8005 cm | 0.829 |
| #9006 | cm | 26 | 0.984 | 0.007 | 0.967 | 1.001 | #9006 pe vs #9006 cm | 0.775 |

Table 10. Statistical comparison of fibres size of *M. venosa* (ventral and dorsal valve) in the anterior transverse sections. ①: specific zones see Figure 9. N: number of measurement. Significant values ($p$–value ≤ 0.05) are marked in bold style.

| Sample | position ① | N | Mean (μm) | STD | Min (μm) | Max (μm) | Difference between means (μm) and ($p$–values) | Difference between means (μm) and ($p$–values) | Difference between means (μm) and ($p$–values) |
|---|---|---|---|---|---|---|---|---|---|
| | | | | | | | #9006 vs #8005 for the same zone | Z1 vs Z2, Z2 vs Z3 for the same vertical position in the same specimen | Z1vs Z2, Z2 vs Z3 for the same transverse position in the same specimen |
| #9006 | Z1–1 | 26 | 4.43 | 1.06 | 2.86 | 6.74 | 0.23 (0.402) | #9006 Z1–1 vs Z2–1 **0.60 (0.013)** | |
| #8005 | Z1–1 | 49 | 4.66 | 1.13 | 1.89 | 7.37 | | | |
| #9006 | Z2–1 | 53 | 3.83 | 0.66 | 2.76 | 5.30 | 0.12 (0.419) | #9006 Z2–1 vs Z3–1 0.07 (0.650) | #9006 Z1 vs Z2 **0.48 (0.011)** |
| #8005 | Z2–1 | 65 | 3.95 | 1.03 | 2.06 | 6.46 | | | |
| #9006 | Z3–1 | 38 | 3.76 | 0.80 | 2.32 | 5.55 | 0.32 (0.134) | #8005 Z1–1 vs Z2–1 **0.71 (0.001)** #8005 Z2–1 vs Z3–1 0.13 (0.554) | #9006 Z2 vs Z3 0.14 (0.323) |
| #8005 | Z3–1 | 44 | 4.08 | 1.05 | 2.22 | 7.53 | | | |
| #9006 | Z1–2 | 26 | 4.71 | 1.27 | 2.76 | 8.38 | **0.74 (0.024)** | #9006 Z1–2 vs Z2–2 0.33 (0.200) | #8005 Z1 vs Z2 **0.59 (< 0.001)** |
| #8005 | Z1–2 | 46 | 5.45 | 1.29 | 2.94 | 10.43 | | | |
| #9006 | Z2–2 | 48 | 4.38 | 0.90 | 2.87 | 7.00 | **0.62 (0.001)** | #9006 Z2–2 vs Z3–2 0.30 (0.144) | #8005 Z2 vs Z3 0.09 (0.595) |
| #8005 | Z2–2 | 59 | 5.00 | 0.97 | 2.94 | 7.16 | | | |
| #9006 | Z3–2 | 40 | 4.68 | 1.01 | 2.57 | 7.76 | 0.40 (0.087) | #8005 Z1–2 vs Z2–2 **0.45 (0.048)** #8005 Z2–2 vs Z3–2 0.08 (0.720) | |
| #8005 | Z3–2 | 38 | 5.08 | 1.00 | 3.02 | 7.78 | | | |
| #9006 | Z4–1 | 23 | 3.79 | 0.71 | 2.72 | 4.99 | **0.72 (0.003)** | #9006 Z4–1 vs Z5–1 0.11 (0.594) | |
| #8005 | Z4–1 | 58 | 4.51 | 1.02 | 2.15 | 7.11 | | | |
| #9006 | Z5–1 | 24 | 3.68 | 0.72 | 2.54 | 5.19 | NA | | #9006 Z4 vs Z5 0.09 (0.615) |
| #9006 | Z4–2 | 33 | 4.61 | 0.89 | 3.15 | 6.55 | 0.24 (0.272) | #9006 Z4–2 vs Z5–2 0.06 (0.811) | |
| #8005 | Z4–2 | 52 | 4.85 | 1.01 | 3.07 | 6.90 | | | |
| #9006 | Z5–2 | 24 | 4.67 | 1.08 | 2.79 | 7.48 | NA | | |



| Sample | Zone | n | | | | | #63 vs #43 vs #158/223 for the same zone | Z1 vs Z2 for the same vertical position in the same specimen | Z1 vs Z2 for the same transverse position in the same specimen |
|---|---|---|---|---|---|---|---|---|---|
| #63 | Z1–1 | 36 | 3.37 | 0.59 | 2.39 | 4.97 | #63 vs #158/223 **0.40 (0.013)** | #63 Z1–1 vs Z2–1 **0.72 (< 0.001)** | |
| #43 | Z1–1 | 70 | 3.73 | 0.98 | 1.63 | 6.94 | #43 vs #158/223 | | |
| #158/223 | Z1–1 | 29 | 2.97 | 0.66 | 2.03 | 4.52 | | #43 Z1–1 vs Z2–1 0.26 (0.109) | |
| #63 | Z2–1 | 24 | 4.09 | 0.75 | 2.84 | 5.85 | #63 vs #158/223 0.17 (0.404) | | #63 Z1 vs Z2 **0.80 (< 0.001)** |
| #43 | Z2–1 | 61 | 3.99 | 0.82 | 1.95 | 5.88 | #43 vs #158/223 0.07 (0.691) | #158/223 Z1–1 vs Z2–1 **0.95 (< 0.001)** | |
| #158/223 | Z2–1 | 56 | 3.92 | 0.83 | 2.17 | 6.14 | | | #43 Z1 vs Z2 **0.40 (0.001)** |
| #63 | Z1–2 | 35 | 4.02 | 0.87 | 2.56 | 6.19 | #63 vs #158/223 **0.73 (0.001)** | | |
| #43 | Z1–2 | 71 | 4.04 | 0.87 | 2.16 | 7.24 | #43 vs #158/223 **0.75 (< 0.001)** | #63 Z1–2 vs Z2–2 **0.95 (< 0.001)** | #158/223 Z1 vs Z2 **1.2 (< 0.001)** |
| #158/223 | Z1–2 | 25 | 3.29 | 0.67 | 2.04 | 4.73 | | #43 Z1–2 vs Z2–2 **0.58 (0.001)** | |
| #63 | Z2–2 | 20 | 4.97 | 0.95 | 3.64 | 7.19 | #63 vs #158/223 0.28 (0.234) | #158/223 Z1–2 vs Z2–2 **1.4 (< 0.001)** | |
| #43 | Z2–2 | 56 | 4.62 | 1.10 | 2.68 | 7.67 | #43 vs #158/223 0.07 (0.688) | | |
| #158/223 | Z2–2 | 55 | 4.69 | 0.85 | 3.02 | 7.09 | | | |

Table 11. Carbon and oxygen fractionation in our cultured *M. venosa* specimens.

| Sample #ID | Treatment | Avg. $\Delta^{13}C_{cal-DIC}$ | Avg. $\Delta^{18}O_{cal-sw}$ |
|---|---|---|---|
| #8004 | Control | -4.06 | 29.99 |
| #9005 | Acidification $pH_2$ | -1.21 | 30.92 |
| #9004 | Acidification $pH_2$ | -2.23 | 30.70 |




Figure 1. Map of Comau Fjord. Upper left map: Overview of Chilean Patagonia. Lower left map: Gulf of Ancud with connection in the North and South to the Pacific Ocean. Right hand map: Fjord Comau with localities of brachiopod sample collection. In both maps the rectangle marks the location of Comau Fjord.



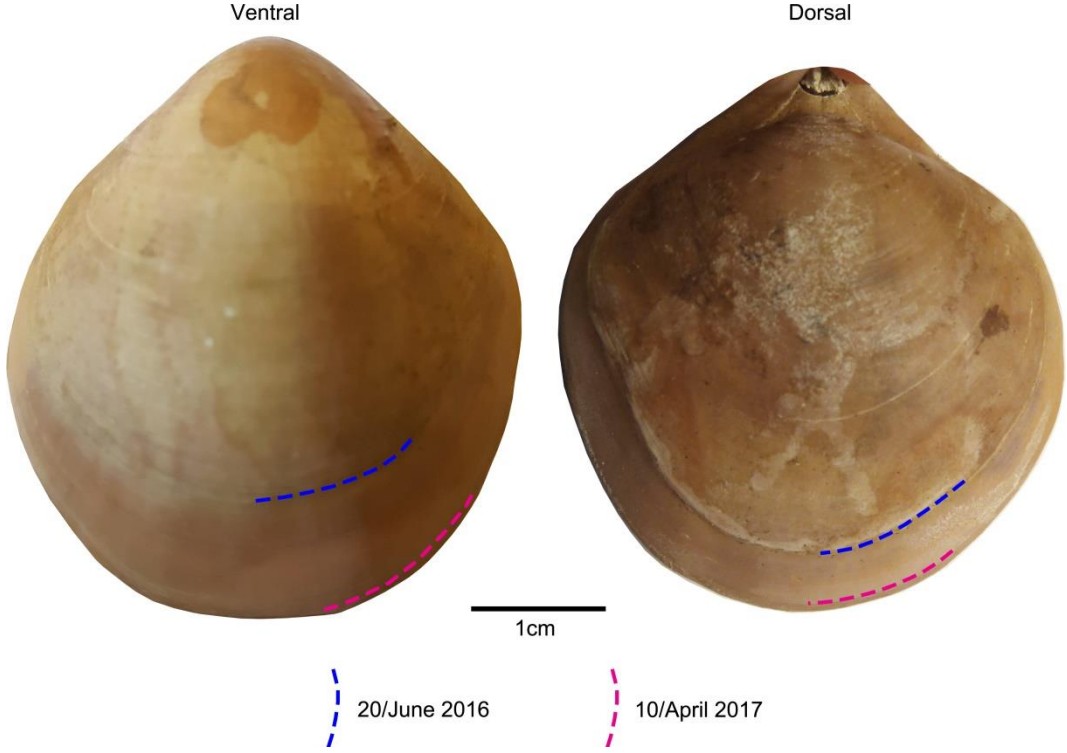

Figure 2. Growth lines marked with calcein on the surface of the brachiopod specimens (#9006).



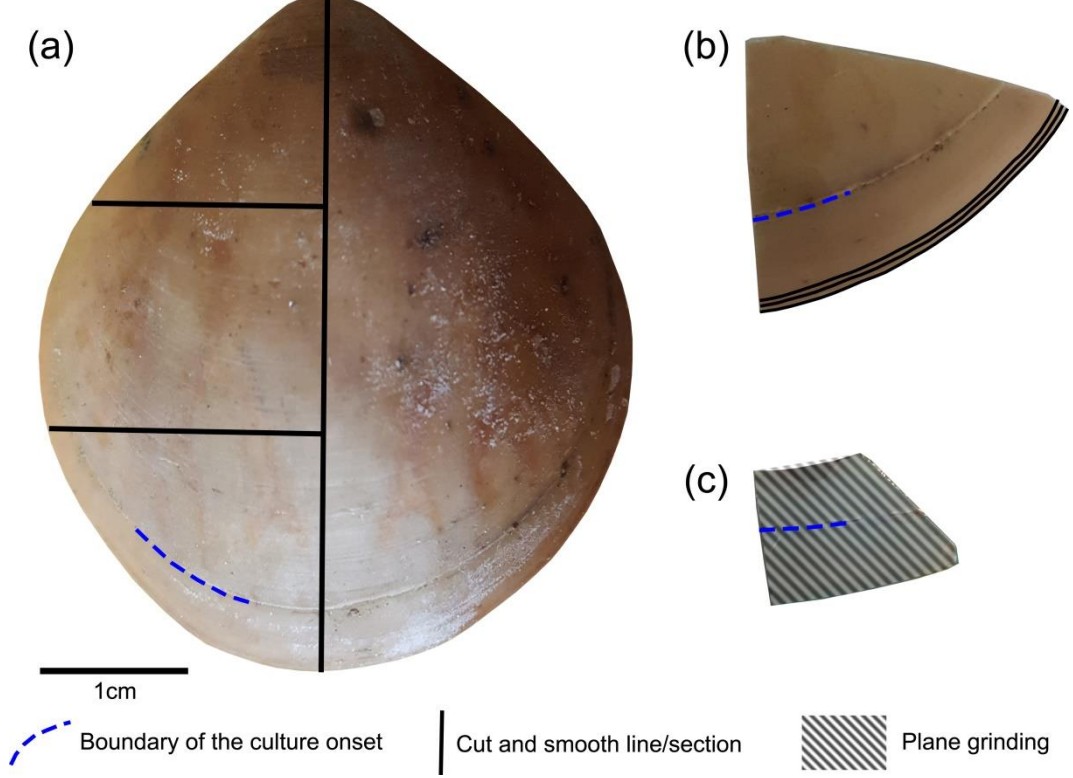

Figure 3. Brachiopod shell sample cut along different axes. A, longitudinal and transverse sections; B, transverse sections at the anterior margin of the shell; C, plane grinding of the external surface of the shell.



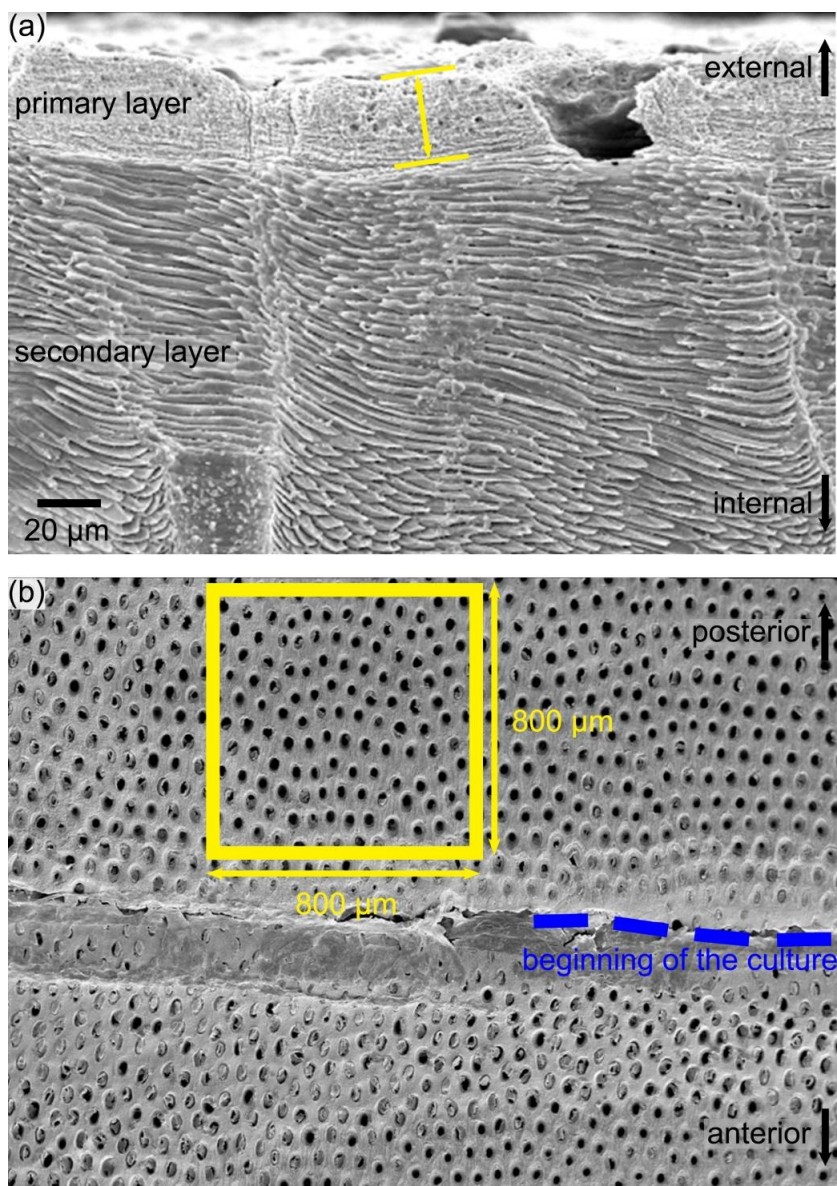

Figure 4. Measurement methods used for the thickness of primary layer and the density of the endopunctae. Note that for the latter, endopunctae were counted when included for more than their half diameter inside the square.



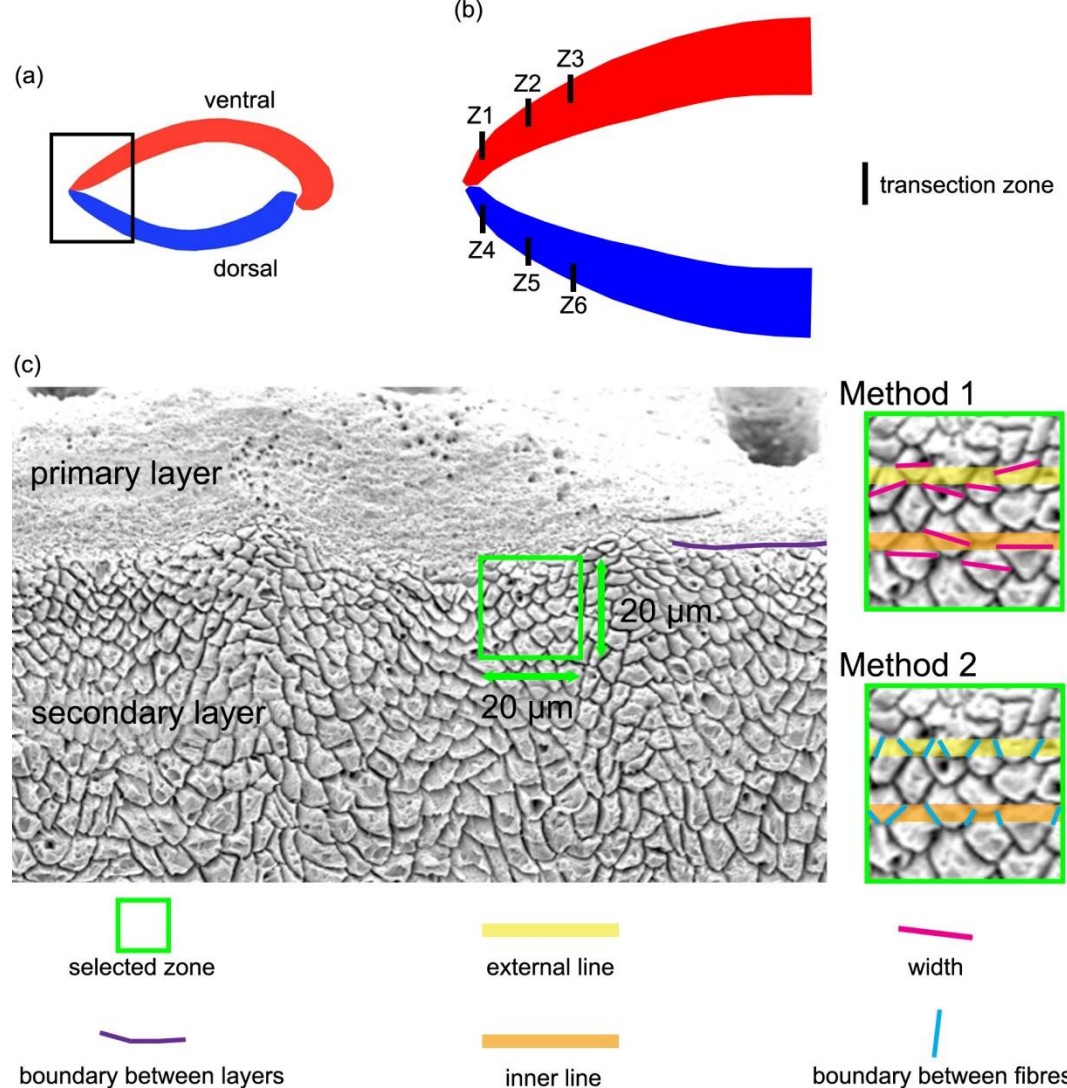

Figure 5. Methods of measurements used in the anterior transverse sections. All SEM images are oriented in the same direction: base of the primary layer facing upwards. A square (20 μm × 20 μm) with its upper side just overlapping the boundary between the primary and secondary layer was analysed. Method 1, refers to the measurement of the width of the fibres crossed by two standard lines, which were located in the same position and at the same distance in all 194 squares analysed (yellow and orange lines); Method 2, calculation of the numbers of boundaries between the fibres, which are crossed by two standard lines were carried out.





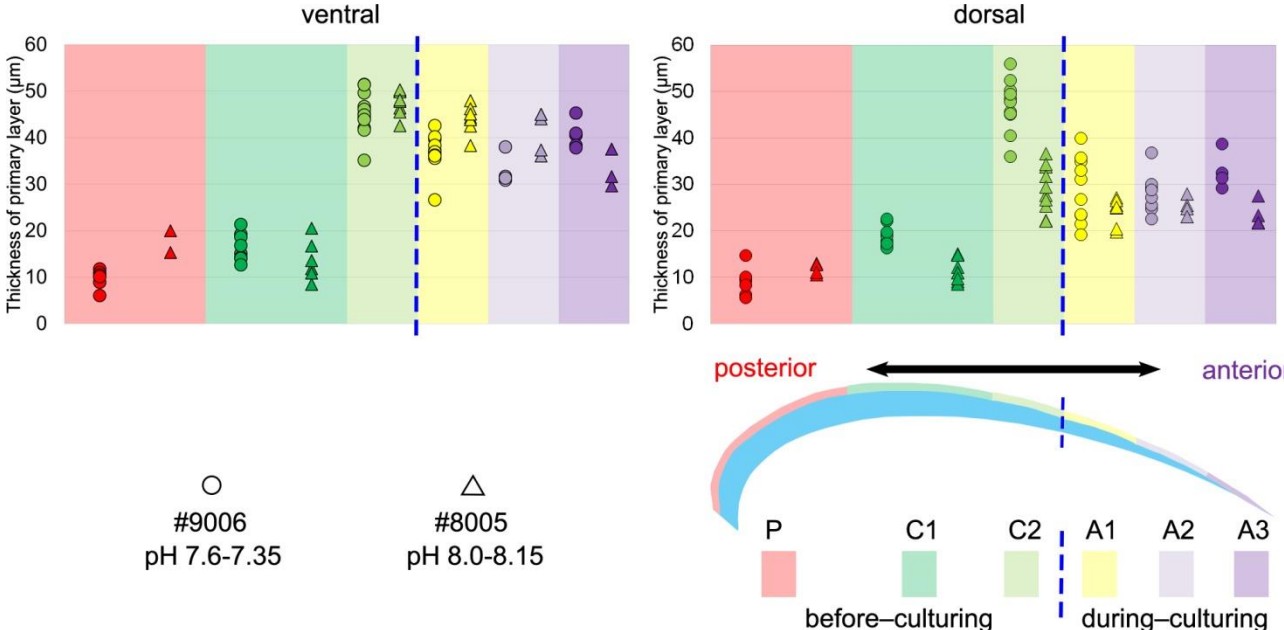

Figure 6. Variations of the thickness of the primary layer (ventral and dorsal valve) of a *M. venosa* specimen cultured at pH 7.35 and 7.6 (#9006) and a specimen cultured at pH 8.0–8.15 (#8005).



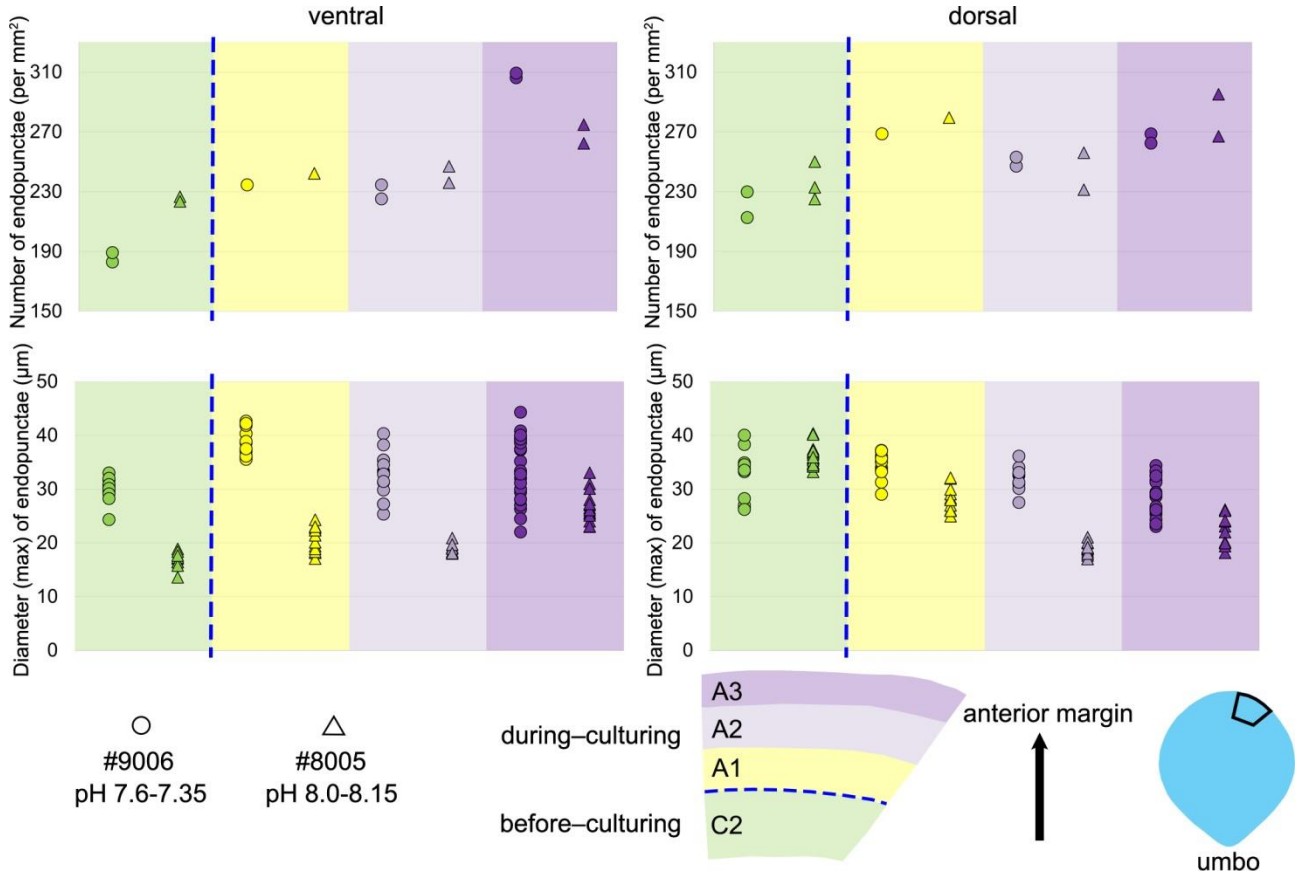

Figure 7. Variations in the number and diameter (max) of endopunctae in the ventral and dorsal valve from a specimen of *M. venosa* cultured at pH 7.35 and 7.6 (#9006) and a specimen cultured at pH 8.0–8.15 (#8005).




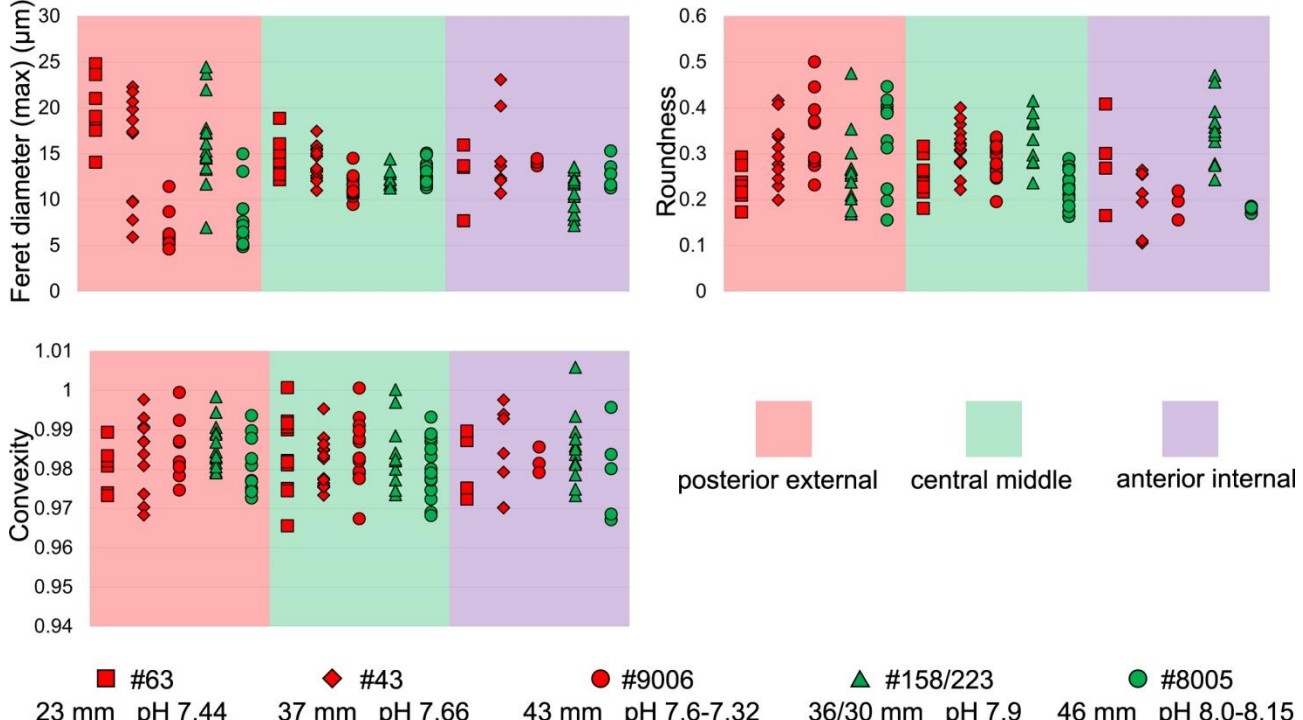

Figure 8. Comparisons of the fibre size and shape of *M. venosa* (ventral and dorsal valve) at different positions along the posterior-anterior axis; pH conditions of culturing or natural environment are reported. One circle point represents one measurement. Outliers have been removed, the latter were identified with Tukey's fences (Tukey, 1977), when falling outside the fences F1 and F2 [F1 = Q1 - 1.5IQR; F2 = Q3 + 1.5IQR; Q1/Q3 = first/third quartiles; IQR (interquartile range) = Q3 - Q1].





Figure 9. Differences in sizes of fibres of *M. venosa* (ventral and dorsal valve) in the anterior transverse sections of specimens cultured at different pH conditions. A, B: The bottom/top of the box and the band inside the box are the first/third quartiles and the median of the data respectively; ends of the whiskers represent the minimum and maximum of results. C, D: Circle point represents average data, $N_m$: number of measurement.





Figure 10. Plots of $\delta^{13}$C and $\delta^{18}$O of the ventral and dorsal valves of *M. venosa* specimens along their growth axis. Different colour backgrounds represent different pH conditions during growth. When few data were available, data-points were joined by dashed lines.



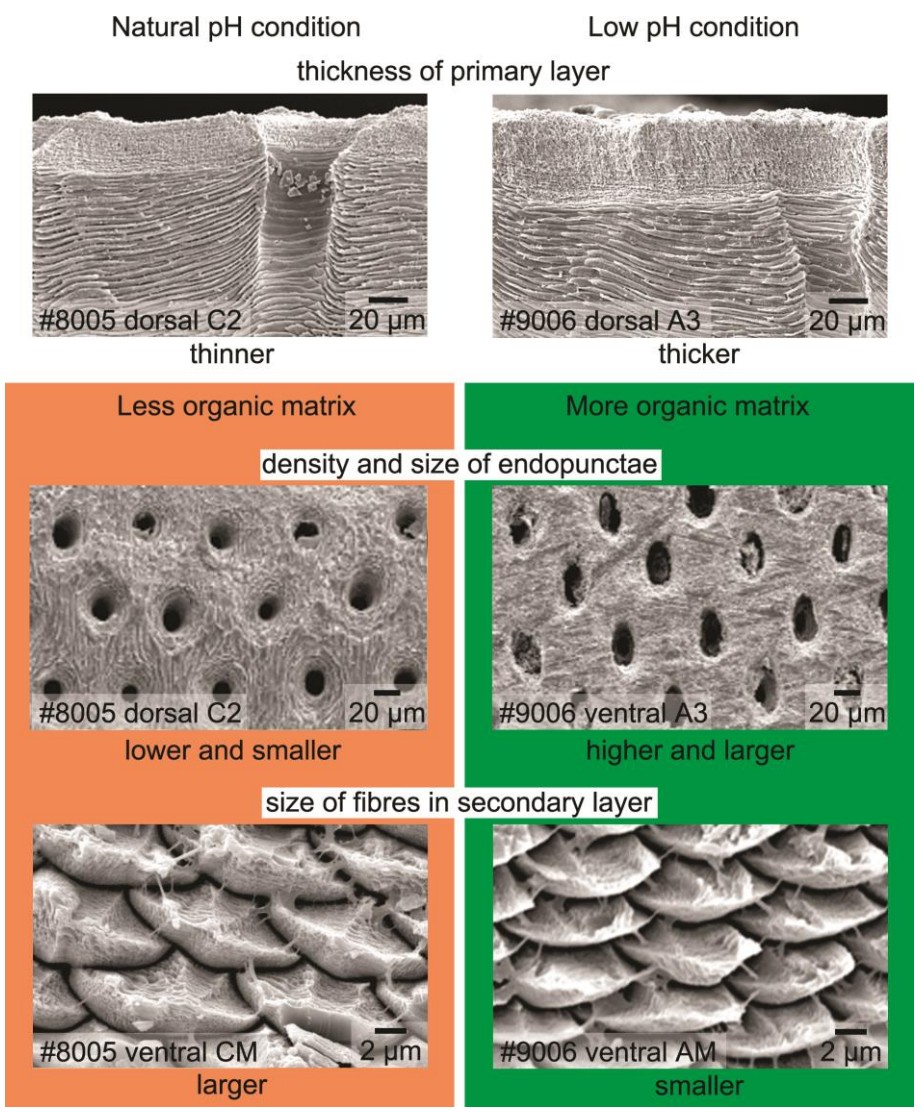

Figure 11. Relationship between the microstructure and the organic components of calcified shells of brachiopods. Position information see Figure 6 and Figure 7; CM: central middle part; AM: anterior middle part.



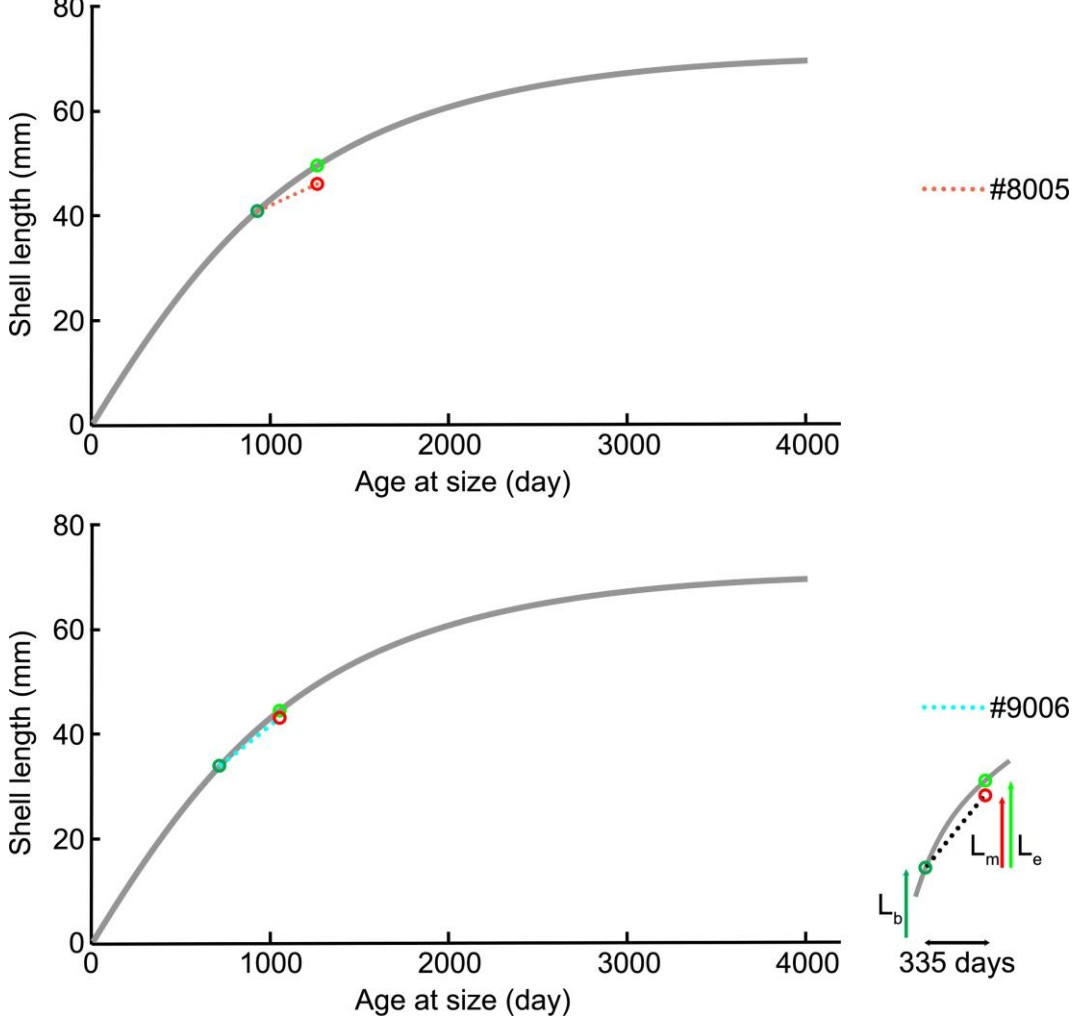

Figure 12. Projection of shell length of ventral valve on the von Bertalanffy growth function (grey line) $L_t = 71.53 [1 - e^{-0.336(t-t0)}]$, source from Baumgarten et al. (2013), $L_b$: shell length at the beginning of culturing; $L_m$: measured shell growth at the end of culturing; Le: expected shell growth.

