# Peer review of "Variation in brachiopod microstructure and isotope geochemistry under low pH-ocean acidification-conditions"

_Biogeosciences, 2018_

## Referee Comment (RC1) · K. Azmy (Referee) · 13 Aug 2018

Dear editor: I have read the manuscript by Ye et al entitled "Variation in brachiopod microstructure and isotope geochemistry under low pH–ocean acidification–conditions", which test the ultrastructural and stable isotope geochemical variations in the shell of M. venosa shells that were partially cultured under different pH conditions. The subject of the manuscript fits well within the scope of BG and the paper present a novel application to brachiopods that are known to archive the climatic and oceanographic conditions of the ambient seawater.

The title reflects the paper contents and the abstract is concise and to the point. The

authors used the proper methodology and utilized the most suitable tools to perform their investigation that lead to clear and reliable and good results that will contribute significantly to the subject of study. The experimental procedure is clear and the statistical analyses utilized are suitable. The references are up to date and the general structure of the manuscript is appropriate.

I have some minor comments that I list hereby below.

Karem Azmy

Page 7 Line 6: How were samples transported? in what type of containers and under what conditions of transportation. Provide couple of lines that describe the conditions during transportation and how they were maintained to simulate the original environment as close as possible.

Page 11 Line 7: "Four sub-zones (C2, A1, A2, A3)", a figure is needed to be cited here to show the positions of those sub-zones.

Page 15 Line 3: after "microdrill" add here at "low speed"

Page 16 caption of Table 5: "are marked in bold style". There nothing is marked in bold font in the table. Mark those values in bold font. Also replace "style" by "font" or you can delete the entire word "style" and leave it as "marked in bold".

Change to "N" into "n" in italic font and fix it in the rest of table headings. This way you do not need to repeat it in the following table captions

Page 17 Line 12: "worth noting" change to "noteworthy" Caption of Table 6: What does this mean? "âŚă: Specific zones see to Figure 7". Something is wrong. Delete "to" Also no specific zones marked on Figure 7. This is confusing and similarly in the Caption of Table 7.

Is STD 1 sigma or 2 sigma. Replace by the correct value and use the Greek proper letter. Fix it in the rest of the manuscript and table captions when applicable

If you used exel for your calculations, then this is 1ïĄş.

Page 28 Line 27: "which in", delete "in" and replace by "during". Also add "s" to "expansion"

Page 31 Line 6, add "," after "inconspicua"

Page 33 Line 22: replace "similarly" by "and similar"

Page 34 Line 9: "We exclude. . . . . . . . . . ." The structure of this sentence is wrong. Rewrite it.

Page 34 Line 11: add "a" before "similar"

Page 34 Line 16: delete "also" and put it after "been" and delete the other "also in" after "observed"

Please also note the supplement to this comment:
https://www.biogeosciences-discuss.net/bg-2018-332/bg-2018-332-RC1-supplement.pdf

---

## Referee Comment (RC2) · Anonymous Referee #2 · 19 Sep 2018

Dear Editor,

The manuscript 'Variation in brachiopod microstructure and isotope geochemistry under low pH–ocean acidification–conditions' examines the change to micro-structure and biogeochemistry of the shell of the brachiopod Magellania venosa in natural conditions versus experimentally cultured brachiopods under low pH environments. The authors apply scanning electron microscopy (SEM) to understand changes to the microstructure in the form of the distribution of endopunctae in the anterior margin and the thickness of the primary layer. In addition, the study applies stable carbon and oxygen isotopes ($\delta18O$, $\delta13C$) to relate the shell growth to the surrounding seawater

chemistry in which the brachiopods were grown. This is a thorough investigation of the changes to the brachiopod shell formation under low pH environments. The authors comment from a biogeochemistry point of view relating the changing microstructure and isotope geochemistry of the shell to environmental carbon isotopes therefore providing further evidence for brachiopods to be useful as an ocean acidification proxy in geological samples. A very good multi-disciplinary approach to determine the impacts on the brachiopods shell growth.

I do have a few concerns in the way the samples were prepared using 5% acid etching, and if this could possibly mask the impacts of the experimental acidification on the microstructure. However, these are minor concerns outlined below for the authors. The use of calcein staining to mark the new growth of the shell distinguishes where natural and low pH environments impact the shell growth in the brachiopod. This should ensure that the authors can identify any similarities in the low pH treatment versus the acid etching. However, this should be discussed in the manuscript and perhaps guide the reader to the nice figures representing this. For example, can the authors provide figures 4 for each treatment for comparison? These are very nice visual representations of how the brachiopod microstructure is affected under low pH treatments versus the natural growth ahead of the calcein staining.

The manuscript is appropriate and well-suited for publication in Biogeosciences Discussions, I would recommend for publication with minor edits as detailed below.

Minor comments to the authors.

Introduction Please re-phrase, 'pH has dropped by 0.1 pH units and will probably drop another 0.3-0.5', these are projections based on modelling of historic data, I would suggest predicted or projected instead of probably. Line 15 states 'calcifying organisms', the table 1 refers only to a few brachiopod studies, please change to brachiopods. I could not comment on the supplementary table here. I would instead suggest including a sentence referencing some of the key papers outlining the consequences of experimental acidification on biomineral formation in other calcifying organisms. There are such studies examining acidification impact on the microstructure of the sea urchin spicules for example Bray et al., 2014 (Med. Mar. Sci.), PUPA Gilberts group including studies by Politi et al., the authors only list here studies applied to molluscs.

Materials and methods. Page 8, line 2, how long were the brachiopods acclimated for prior to calcein staining and CO2 induced acidification? Low-pH culture of several brachiopods was done under two phases, what was the justification for this? Were these two phases comparable in their treatments? In general, the treatments appear clear in the table 2 and 3, however it is difficult to understand the experimental design without the details which are currently not obtainable from Jurikova et al., in review.

Microstructural analyses – this section is much easier to understand with sufficient detail for the reader to reproduce. The authors used 5% hydrochloric acid for etching the shell prior to SEM analyses. Although a standard protocol for SEM imaging, the authors should comment on how they can be sure that this has not affected the microstructure in comparison to the experimental acidification of the culture. How would the impact the microstructure be distinguishable compared to the acid etch of the microstructure of shells? Figure 4 and 5, can the authors please provide a specimen reference to which sample and treatment these images relate to? The manuscript suggests just two specimens #8005 and #9006 were used for these analyses. Can the authors conclude that these are representable as a sample population? Are these images available in supplementary information for comparison?

Carbonate stable isotopes analyses Likewise, the authors use 5% hydrochloric acid to clean shell prior to sample preparation for stable isotopes. Please detail why this was used, for example to remove organic material? If so why was a bleach or plasma ash treatment not chosen for this purpose to avoid potential issues with comparing experimental acidification treatments with hydrochloric acid treated shells? Section 3.3.2 During culturing Page 23, the authors state that 'The results from specimens (#43 and #63) grown under low pH conditions (pH3 and pH4) for a short time interval

of 214 days are difficult to interpret, as in this case, there is no direct control experiment sample to compare', can the authors confidently relate the changing microstructure and geochemistry here to acidification is the only comparison are those samples grown under natural conditions? It appears that the experimental treatments here are similar despite the pH used (Figure 9). I would question the relevance of this section, perhaps omit or justify how this is comparable.

Section 3.4 Stable isotopes Nice figure 10, it is clear to see trends between three specimens for significantly lighter stable carbon isotopes with experimental low-pH treatment compared to natural versus control treatments. Did the authors compare these data statistically?

Discussion Page 28, 'electron back scattering diffraction' should be electron backscatter diffraction. Line 20, 'May this indicate a greater amount of organic components in this part of the shell?' is this what the authors suggest? Please rephrase not as a question but a statement with references or omit. Lines 28, 'living organism' this should be living organisms. Page 34-35. The discussion of the depleted $\delta$18O, $\delta$13C is related to changes in percentage, can the authors present the statistical significance here of the changing isotope values? The authors state that there is individual specimen variability, does this remove the significance of the low pH treatment over the isotope depleted values? Or are the authors suggesting here that there are insufficient specimen numbers to make significant statements relating to the isotope data? Page 35, line 11 'More measurements are however needed to fully answer this.'? Page 35, line 16 'Thus, we think', perhaps the data suggest? The authors end in the statement 'secondary layer isotope record may reflect the environmental conditions supporting the interpretation of brachiopod shells as good archives of geochemical proxies, even when stressed by ocean acidification.'. This is also stated in the abstract as one of the main implications of this study. Following the current discussion on page 34-35 I would question whether the authors can make this statement, and whether there is sufficient evidence to support this, although Figure 10 does suggest this is the case. Please directly refer to the

data here; are there sufficient samples, what is the n-number? This will enable the reader to determine if the manuscripts data do support these conclusions. If this data is not available then the authors will need to remove this emphasis from the abstract and conclusion statements. Page 35 conclusions 'This was related to the source of carbon dioxide gas used in the culture setup', could this not be due to a change in the carbonate compositions as a result of adding $CO_2$ impacting the DIC? Did you test the carbon isotopes of the gas? I have seen this drop in carbon isotopes in the natural seawater samples where increasing $CO_2$ from run-off caused a lighter carbon isotope value. The authors should expand this discussion to the previous paragraph.

―――――――――――――――――――

---

## Short Comment (SC1) · 27 Sep 2018

Firstly, it is great to read that other researchers are using living specimens of this highly calcium-carbonate-dependent group to address outstanding questions of biological responses to ocean acidification. I look forward to future publications from this lab. I have a few minor comments in relation to correctly citing previous research on ocean acidification impacts on brachiopods:

Page 2, Line 21-23: Please add to this sentence about previous findings of ocean acidification impacts on brachiopods that Cross et al. (2018) also found that punctae have become thinner over the last 120 years, which partially explained the increase in

shell density over this time period.

Page 2, Table 1: Please specify that shell growth rates and shell repair frequencies of *Calloria inconspicua* were not affected by low pH in the row related to Cross et al. (2016) in the column stating "not affected by lower pH".

Page 3, Table 1: Please correct the number of specimens used in the row related to Cross et al. (2018). 389 adult specimens were used in the shell morphology analysis. A subsample of 40 brachiopods (2-5 specimens per decade over the last 120 years) were used for further shell analysis on shell density, punctal width, punctal density, shell dissolution, shell thickness and shell elemental composition. Please also add that no changes were found in shell dissolution over the last 120 years.

Page 3, Table 1: Please specify that shell growth rates and shell repair frequencies of *Liothyrella uva* were not affected by low pH and temperature in the row related to Cross et al. (2015) in the column stating "not affected by lower pH".

Page 31, Line 9: To avoid confusion, please add in that these specimens are from the same locality in New Zealand (Paterson Inlet, Stewart Island, New Zealand).

Page 31, Line 10: Please also add that the pH decrease by 0.1 pH units occurred over the last two decades whilst the $2°C$ increase in temperature occurred over the last 60 years.

Page 32, Line 5-8: Majority of the studies listed here did not investigate brachiopod growth rates. To avoid confusion and strengthen the authors point that there is a limited database on ocean acidification impacts on brachiopods, only include studies here on brachiopods.

———————————————————

---

## Author Comment (AC2) · 18 Oct 2018

Dear Anonymous Referee #2, We want to thank you very much for the very constructive, helpful and valuable comments and corrections, which helped us to ameliorate the manuscript. The details responses are presented as a Supplement PDF form (attached)

Please also note the supplement to this comment:
https://www.biogeosciences-discuss.net/bg-2018-332/bg-2018-332-AC2-supplement.zip

---

## Author Comment (AC3) · 18 Oct 2018

Dear Dr. Emma Cross,
We want to thank very much Emma Cross for the very constructive, helpful and valuable comments and corrections, which helped us to ameliorate the manuscript. Please find below my responses to all comments:

Title: Variation in brachiopod microstructure and isotope geochemistry under low pH–ocean acidification–conditions
Authors: Facheng Ye, Hana Jurikova, Lucia Angiolini, Uwe Brand, Gaia Crippa, Daniela Henkel, Jürgen Laudien, Claas Hiebenthal, and Danijela Šmajgl
MS No.: bg-2018-332
MS Type: Research article

The responses to referees are structured following this sequence using different colors:
(1) Comments from Referees;
(2) Author's response;
(3) Changes in the manuscript: original sentences/revised sentences

Comments from E. Cross

e.l.cross@cantab.net

Firstly, it is great to read that other researchers are using living specimens of this highly calcium-carbonate-dependent group to address outstanding questions of biological responses to ocean acidification. I look forward to future publications from this lab. I have a few minor comments in relation to correctly citing previous research on ocean acidification impacts on brachiopods:

1) Page 2, Line 21-23: Please add to this sentence about previous findings of ocean acidification impacts on brachiopods that Cross et al. (2018) also found that punctae have become thinner over the last 120 years, which partially explained the increase in shell density over this time period.

Answer:

We have added this sentence in the revised manuscript.

Page 2, Line 21-23:

The few studies that examined brachiopods or brachiopod shells suggest that the latter suffered increased dissolution under lower seawater pH conditions, whereas the organism either exhibited no changes, or an increase in shell density [calculated as dry mass of the shell (g)/shell volume (cm$^3$)], but otherwise no changes in shell morphology and trace chemistry (Table 1).

changed to

The few studies that examined brachiopods or brachiopod shells suggest that the latter suffered increased dissolution under lower seawater pH conditions, whereas the organism either exhibited no changes, or an increase in shell density [calculated as dry mass of the shell (g)/shell volume (cm$^3$)], but otherwise no changes in shell morphology and trace chemistry (Table 1). Cross et al. (2018) also found that punctae have become thinner over the last 120 years, which partially explained the increase in shell density over this time period.

2) Page 2, Table 1: Please specify that shell growth rates and shell repair frequencies of *Calloria inconspicua* were not affected by low pH in the row related to Cross et al. (2016) in the column stating "not affected by lower pH".

Answer:

We have indicated it in the revised manuscript.

Page 2, Table 1 column 3 row 2:

not affected by lower pH

changed to

shell growth rates and shell repair frequencies were not affected by low pH

3) Page 3, Table 1: Please correct the number of specimens used in the row related to Cross et al. (2018). 389 adult specimens were used in the shell morphology analysis. A subsample of 40 brachiopods (2-5 specimens per decade over the last 120 years) were used for further shell analysis on shell density, punctal width, punctal density, shell dissolution, shell thickness and shell elemental composition.

Answer:

We have corrected it in the revised manuscript.

Page 2, Table 1 column 1 row 3:

N = 389 (adults)

changed to

N = 389 (adults) for shell morphology analyses*.

And we have added the note below the table:

*A subsample of 40 brachiopods (2-5 specimens per decade over the last 120 years) were used for further shell analysis on shell density, punctal width, punctal density, shell dissolution, shell thickness and shell elemental composition.

4) Please also add that no changes were found in shell dissolution over the last 120 years.
Answer:
We have added it in the revised manuscript.
Page 3, Table 1 column 3 row 3:
add:
no changes were found in shell dissolution over the last 120 years.

5) Page 3, Table 1: Please specify that shell growth rates and shell repair frequencies of *Liothyrella uva* were not affected by low pH and temperature in the row related to Cross et al. (2015) in the column stating "not affected by lower pH".
Answer:
We have specified it in the revised manuscript.
Page 3, Table 1 column 2 row 4:
not affected by lower pH
changed to
not affected by lower pH and temperature
Page 3, Table 1 column 3 row 4:
not affected by either low pH conditions or temperature
changed to
shell repair frequencies were not affected by low pH and temperature

6) Page 31, Line 9: To avoid confusion, please add in that these specimens are from the same locality in New Zealand (Paterson Inlet, Stewart Island, New Zealand).
Answer:
We have added it in the revised manuscript.
Page 31, Line 9:
One response, however, appears to reinforce the shells of C. inconspicua by laying down a denser shell compared to specimens from New Zealand over the last 120 years while subjected to a slight decrease in pH (by 0.1) and 2°C increase in temperature over the last two decades (Cross et al., 2018).
changed to
One response, however, appears to reinforce the shells of *C. inconspicua* by laying down a denser shell compared to specimens from the same locality in New Zealand (Paterson Inlet, Stewart Island, New Zealand) over the last 120 years while subjected to a slight decrease in pH (by 0.1) and 2°C increase in temperature over the last two decades (Cross et al., 2018).

7) Page 31, Line 10: Please also add that the pH decrease by 0.1 pH units occurred over the last two decades whilst the 2°C increase in temperature occurred over the last 60 years.
Answer:
We have added it in the revised manuscript.
Page 31, Line 10:
while subjected to a slight decrease in pH (by 0.1) and 2°C increase in temperature

over the last two decades (Cross et al., 2018).
changed to
while subjected to a slight decrease in pH (by 0.1) occurred over the last two decades whilst the 2°C increase in temperature occurred over the last 60 years. (Cross et al., 2018).

8) Page 32, Line 5-8: Majority of the studies listed here did not investigate brachiopod growth rates. To avoid confusion and strengthen the authors point that there is a limited database on ocean acidification impacts on brachiopods, only include studies here on brachiopods.
Answer:
As we mainly discussed the effects of acidification on the growth rates of marine calcifiers, we have corrected the word 'brachiopod' to 'marine calcifiers' in the revised manuscript.
Page 32, Line 5-8:
show no or little impact of acidification on brachiopod growth rates
changed to
show no or little impact of acidification on the growth rates of marine calcifiers

---

## Author Response (AR1)

Dear Editor,

Please find below my responses to all comments (referee comments and short comment):

Title: Variation in brachiopod microstructure and isotope geochemistry under low pH–ocean acidification–conditions

Authors: Facheng Ye, Hana Jurikova, Lucia Angiolini, Uwe Brand, Gaia Crippa, Daniela Henkel, Jürgen Laudien, Claas Hiebenthal, and Danijela Šmajgl

MS No.: bg-2018-332

MS Type: Research article

The responses to referees are structured following this sequence using different colours:

(1) Comments from Referees;

(2) Author's response;

(3) Changes in the manuscript: original sentences/revised sentences

We want to thank very much Karem Azmy, the anonymous reviewer 2 and Emma Cross for the very constructive, helpful and valuable comments and corrections, which helped us to ameliorate the manuscript.

**1. Comments from K. Azmy (Referee):

I have read the manuscript by Ye et al entitled "Variation in brachiopod microstructure and isotope geochemistry under low pH–ocean acidification–conditions", which test the ultrastructural and stable isotope geochemical variations in the shell of M. venosa shells that were partially cultured under different pH conditions. The subject of the manuscript fits well within the scope of BG and the paper present a novel application to brachiopods that are known to archive the climatic and oceanographic conditions of the ambient seawater.

The title reflects the paper contents and the abstract is concise and to the point. The authors used the proper methodology and utilized the most suitable tools to perform their investigation that lead to clear and reliable and good results that will contribute significantly to the subject of study. The experimental procedure is clear and the statistical analyses utilized are suitable. The references are up to date and the general structure of the manuscript is appropriate.

I have some minor comments that I list hereby below.

1) Page 7 Line 6: How were samples transported? in what type of containers and under what conditions of transportation. Provide couple of lines that describe the conditions during transportation and how they were maintained to simulate the original environment as close as possible.

Answer:

We agree with the referee's suggestion and we have added the requested information in the revised manuscript.

Page 7 Line 6:

In summary, *M. venosa* individuals sampled in Chile were transported to Germany and cultured under controlled environmental setting in a climate laboratory.

changed to

In summary, individuals of *M. venosa* were collected alive in Chile and transported to GEOMAR, Germany in plastic bags filled with seawater, and maintained under controlled conditions in a climate laboratory.

2) Page 11 Line 7: "Four sub-zones (C2, A1, A2, A3)", a figure is needed to be cited here to show the positions of those sub-zones.

Answer:

We have added the legend showing positions in the revised figure 4.

Page 11 Line 7:

Four sub-zones (C2, A1, A2, A3) were defined according to their position along the posterior-anterior direction,

changed to

Four sub-zones (C2, A1, A2, A3) were defined according to their position along the posterior-anterior direction (Figure 4),

3) Page 15 Line 3: after "microdrill" add here at "low speed"

Answer:

We have added it in the revised manuscript.

Page 15 Line 3:
microdrill with tungsten–carbide milling bit.
changed to
microdrill at low speed with tungsten–carbide milling bit.

4) Page 16 caption of Table 5: "are marked in bold style". There nothing is marked in bold font in the table. Mark those values in bold font. Also replace "style" by "font" or you can delete the entire word "style" and leave it as "marked in bold". Change to "N" into "n" in italic font and fix it in the rest of table headings. This way you do not need to repeat it in the following table captions
Answer:
We have made all these corrections in the caption and in Table 5.
Page 16 caption of Table 5:
are marked in bold style
changed to
are marked in bold

5) Page 17 Line 12: "worth noting" change to "noteworthy"
Answer:
We have changed it in the revised manuscript.
Page 17 Line 12:
However, it is worth noting that in the most anterior part
changed to
However, it is noteworthy that in the most anterior part

6) Caption of Table 6: What does this mean? " "â´S˘a: Specific zones see to Figure 7". Something is wrong. Delete "to" Also no specific zones marked on Figure 7. This is confusing and similarly in the Caption of Table 7.
Answer:
We have made this correction in the revised manuscript.
specific zones marked on Figure 7
changed to
position of zones before culturing: C2, and post culturing: A1, A2, A3 (cf. Figure 7)

7) Is STD 1 sigma or 2 sigma. Replace by the correct value and use the Greek proper letter. Fix it in the rest of the manuscript and table captions when applicable If you used exel for your calculations, then this is 1ïA¿s¸.
Answer:
We have made the correction in the table caption, and fixed it in the rest of the manuscript and table captions.
We have added this sentence in the captions of Tables 5, also changed STD to σ in the other tables (6-10):
Population standard deviation (σ) was calculated using the Excel STDEV.P function.

8) Page 28 Line 27: "which in", delete "in" and replace by "during". Also add "s" to "expansion"

Answer:

We have made this correction in the revised manuscript.

Page 28 Line 26:

Endopunctae, which in life are filled with mantle expansion,

changed to

Endopunctae, which during life are filled with mantle expansions,

9) Page 31 Line 6, add "," after "inconspicua"

Answer:

We have added it in the revised manuscript.

Page 31 Line 6:

For the Antarctic brachiopod *Liothyrella uva* and the New Zealand brachiopod *Calloria inconspicua* no ocean acidification effects on shell growth were detected by Cross et al. (2015, 2016, 2018),

changed to

For brachiopods, in the *Liothyrella uva* (Antarctic) and *Calloria inconspicua* (New Zealand), no ocean acidification effects on shell growth were detected by Cross et al. (2015, 2016, 2018),

10) Page 33 Line 22: replace "similarly" by "and similar"

Answer:

We have made this correction in the revised manuscript.

Page 33 Line 22:

Thus, in bivalves, similarly to our observations,

changed to

Thus, in bivalves, and similar to our observations,

11) Page 34 Line 9: "We exclude: : :: : :: : :: : :" The structure of this sentence is wrong. Rewrite it.

Answer:

We have rewritten this sentence:

Page 34 Line 9:

We exclude that this drop may be produced by shell material added later, during the during–culturing shell thickening, as the samples were taken from the mid-shell layer and not from the shell interior.

changed to

Since the samples were taken from the mid-shell layer and not from the shell interior, we can exclude that the isotope negative shift was produced by shell material added during the during–culturing shell thickening. This small drop may be an artefact of both sampling and analytical uncertainties. However, it does not distract from the isotopic drop observed with culturing.

12) Page 34 Line 11: add "a" before "similar"

Answer:

We have added it in the revised manuscript.

Page 34 Line 11:

negative isotope excursions of similar magnitude were recorded

changed to

negative isotope excursions of a similar magnitude were recorded

13) Page 34 Line 16: delete "also" and put it after "been" and delete the other "also in" after "observed"

Answer:

We have made these corrections in the revised manuscript.

Page 34 Line 16:

Negative shifts in both, $\delta^{13}$C and $\delta^{18}$O values during ontogeny have also been observed also in in the brachiopod *Terebratella dorsata*,

changed to

Negative shifts in both, $\delta^{13}$C and $\delta^{18}$O values during ontogeny have been also observed in the brachiopod *Terebratella dorsata*,

Comments from Anonymous Referee #2,
Dear Editor,
The manuscript 'Variation in brachiopod microstructure and isotope geochemistry under low pH–ocean acidification–conditions' examines the change to micro-structure and biogeochemistry of the shell of the brachiopod Magellania venosa in natural conditions versus experimentally cultured brachiopods under low pH environments. The authors apply scanning electron microscopy (SEM) to understand changes to the microstructure in the form of the distribution of endopunctae in the anterior margin and the thickness of the primary layer. In addition, the study applies stable carbon and oxygen isotopes ($^{18}O$, $^{13}C$) to relate the shell growth to the surrounding seawater chemistry in which the brachiopods were grown. This is a thorough investigation of the changes to the brachiopod shell formation under low pH environments. The authors comment from a biogeochemistry point of view relating the changing microstructure and isotope geochemistry of the shell to environmental carbon isotopes therefore providing further evidence for brachiopods to be useful as an ocean acidification proxy in geological samples. A very good multi-disciplinary approach to determine the impacts on the brachiopods shell growth.

1) I do have a few concerns in the way the samples were prepared using 5% acid etching, and if this could possibly mask the impacts of the experimental acidification on the microstructure.
Answer:
The time of 5% acid etching was so rapid (3 seconds) that it did not affect the microstructure, as shown by a first screening and by previous published studies on preparation methods (e.g. Zaky et al., 2015; Crippa et al., 2016). In any case all the samples experienced the same treatment, with exactly the same possible effects; so for the comparative goals of the manuscript, this would have been negligible. Furthermore, if this treatment could have affected the microstructure- which was excluded by our screening and previous studies -, it could have only slightly affected the outline of the structural units (fibre), but not their size, and not the density of the endopunctae.

2) However, these are minor concerns outlined below for the authors. The use of calcein staining to mark the new growth of the shell distinguishes where natural and low pH environments impact the shell growth in the brachiopod. This should ensure that the authors can identify any similarities in the low pH treatment versus the acid etching. However, this should be discussed in the manuscript and perhaps guide the reader to the nice figures representing this. For example, can the authors provide figures 4 for each treatment for comparison? These are very nice visual representations of how the brachiopod microstructure is affected under low pH treatments versus the natural growth ahead of the calcein staining.
Answer:
There are: Figure 2 to show the growth lines marked with calcein, and Figure 11 to summarize the visible microstructure difference under different pH treatments. Also,

the onset of culturing could be seen by a 'break' in the shell structure-quite visible on the surface. Calcein has been widely used for staining calcium carbonate structures. Calcein has been shown to be incorporated passively into growing calcium carbonate of various taxa (e.g. Moran and Marko 2005; Riascos et al., 2007; Herrmann et al., 2009), including brachiopods (Rowley and Mackinnon 1995). None of the authors reported enhanced mortality or other negative influences on life histories or physiology.
Following the request of the reviewer, an additional plate was added in the supplementary material (supplementary figure 1) to show how the brachiopod microstructure is affected under different treatments.

The manuscript is appropriate and well-suited for publication in Biogeosciences Discussions, I would recommend for publication with minor edits as detailed below. Minor comments to the authors.
3) Introduction Please re-phrase, 'pH has dropped by 0.1 pH units and will probably drop another 0.3-0.5', these are projections based on modelling of historic data, I would suggest predicted or projected instead of probably.
Answer:
We have corrected them in the revised manuscript
Page 2 Line 2:
pH has dropped by 0.1 pH units and will probably drop another 0.3-0.5 units
changed to
pH has dropped by 0.1 units and predicted to drop another 0.3–0.5 units

4) Line 15 states 'calcifying organisms', the table 1 refers only to a few brachiopod studies, please change to brachiopods. I could not comment on the supplementary table here. I would instead suggest including a sentence referencing some of the key papers outlining the consequences of experimental acidification on biomineral formation in other calcifying organisms.
Answer:
Supplementary Table 1 contains all the information on calcifying organisms.
We are sorry that the reviewer could not find the supplementary table, which we report also here in the attached file.
The sentence referencing to some of the key-papers on experimental acidification on biominerals is already written in the manuscript, just below at Page 2 Line 18: Only a few studies deal with the effect of acidification on microstructure (Beniash et al., 2010; Hahn et al., 2012; Stemmer et al., 2013; Fitzer et al., 2014a, b; Milano et al., 2016), and all of them focused on bivalves and show that neither microstructure, nor shell hardness seem to be affected by seawater pH.

5) There are such studies examining acidification impact on the microstructure of the sea urchin spicules for example Bray et al., 2014 (Med. Mar. Sci.), PUPA Gilberts group including studies by Politi et al., the authors only list here studies applied to molluscs.

Answer:

We have added the suggested citations to the supplementary table 1: Bray et al., 2014; Wolfe et al., 2013.

However, the paper of Politi et al., 2008 is about the "Transformation mechanism of amorphous calcium carbonate into calcite in the sea urchin larval spicule" so it is not very relevant for this aim.

6) Materials and methods. Page 8, line 2, how long were the brachiopods acclimated for prior to calcein staining and $CO_2$ induced acidification?
Answer:

We have added the time of the acclimatisation and changed the sentence:

Brachiopods were first left to acclimatize,

changed to

The brachiopods were first acclimatized under control conditions for five weeks,

7) Low-pH culture of several brachiopods was done under two phases, what was the justification for this? Were these two phases comparable in their treatments?
Answer:

Regarding these two phases, basically we had a control and low pH aquarium. The two phases were comparable in their treatments. One reason for increasing the $CO_2$ treatment from 2000ppm to 4000ppm was that the brachiopods have been done obviously well under the 2000ppm treatment and we aimed to increase the chemical impact as much as possible but with considering the survival of the specimen. Before the 4000ppm treatment has been started individuals of *M. venosa* have been stained again with calcein in order to mark the new material grown under 4000ppm.

8) In general, the treatments appear clear in the table 2 and 3, however it is difficult to understand the experimental design without the details which are currently not obtainable from Jurikova et al., in review.
Answer:

Jurikova et al. manuscript (submitted to Geochimica et Cosmochimica Acta) was revised and sent to the Editor for final decision. We will update the information as soon as Hana Jurikova receives the notification about acceptance.

9) Microstructural analyses – this section is much easier to understand with sufficient detail for the reader to reproduce. The authors used 5% hydrochloric acid for etching the shell prior to SEM analyses. Although a standard protocol for SEM imaging, the authors should comment on how they can be sure that this has not affected the microstructure in comparison to the experimental acidification of the culture. How would the impact the microstructure be distinguishable compared to the acid etch of the microstructure of shells?
Answer:

As explained above, the time of 5% acid etching was so rapid (3 seconds) that it did not affect the microstructure, as shown by our screening as well as in previous

published studies on preparation methods (e.g. Zaky et al., 2015; Crippa et al., 2016). In any case, all the samples experienced the same treatment, with exactly the same possible effects; so for the comparative goals of the manuscript, this would have been negligible. Furthermore, if this treatment could have affected the microstructure- which was excluded by our screening and previous studies -, it could have only affected the outline of the structural units (fibre), but not their size, and not the density of the endopuncta.

In any case, in the revised manuscript we have added a sentence stating that the effect of 5% acid etching was negligible.

Page 10, line 5

The sectioned surfaces were manually smoothed with 1200 grit 5 sandpaper, then quickly (3 seconds) cleaned with 5% hydrochloric acid (HCl), immediately washed with tap water and air–dried.

changed to

The sectioned surfaces were manually smoothed with 1200 grit sandpaper, then quickly (3 seconds) cleaned with 5% hydrochloric acid (HCl), immediately washed with water and air–dried. The time of acid etching was kept short so not affect the microstructure (Crippa et al., 2016b).

10) Figure 4 and 5, can the authors please provide a specimen reference to which sample and treatment these images relate to?
Answer:
We have added the information to the figures in the revised manuscript.
Figure 4a: #9006dv; 4b: #8005dv
Figure 5: #43vv

11) The manuscript suggests just two specimens #8005 and #9006 were used for these analyses. Can the authors conclude that these are representable as a sample population? Are these images available in supplementary information for comparison?
Answer:
**43, #63, #158, #223, #8005 and #9006 were analysed for the microstructure of secondary layer, additionally, #8005 and #9006 were also analysed for the thickness of primary layer and the size/density of endopuncta. All the measurement data are available in the Appendix dataset.**

12) Carbonate stable isotopes analyses Likewise, the authors use 5% hydrochloric acid to clean shell prior to sample preparation for stable isotopes. Please detail why this was used, for example to remove organic material? If so why was a bleach or plasma ash treatment not chosen for this purpose to avoid potential issues with comparing experimental acidification treatments with hydrochloric acid treated shells?
Answer:
As explained in the method description (paragraph 2.3), we used 10 % HCl to remove the primary shell layer and surface contaminants; then we immediately rinsed with

distilled water and air–dried. This is an important and fundamental step before doing isotope analyses. As already proved by previous studies (e.g., Veizer, 1992; Carpenter and Lohmann, 1995; Brand et al., 2003, 2013), the primary layer is not secreted in isotope equilibrium with the seawater in which the brachiopod lives, so we have to remove it in order to avoid contamination when sampling the shell. In this way we analysed only the in-equilibrium secondary layer (Parkinson et al., 2005; Cusack et al., 2012; Brand et al., 2013, 2015). To remove the organic material in the shell, as written in paragraph 2.1 we used 36 volume hydrogen peroxide. This in another important step to get clear images of recent brachiopods at the SEM (see Crippa et al., 2016b). We have added a sentence to explain wht we have removed the primary layer before isotope analyses.

.

Page 14 Line 10:
For specimens #8005 and #9006, the primary layer and surface contaminants were manually and chemically removed by leaching with 10 % HCl, rinsed with distilled water and air–dried

changed to

For specimens #8005 and #9006, surface contaminants and the primary layer were first manually and then chemically removed by leaching with 10 % HCl, rinsed with distilled water and air–dried. As the primary layer is not secreted in equilibrium with ambient seawater (e.g., Carpenter and Lohmann, 1995; Brand et al., 2003, 2013), it is important to chemically remove it in order to avoid cross-contamination of results.

13) Section 3.3.2 During culturing Page 23, the authors state that 'The results from specimens (#43 and #63) grown under low pH conditions (pH3 and pH4) for a short time interval of 214 days are difficult to interpret, as in this case, there is no direct control experiment sample to compare', can the authors confidently relate the changing microstructure and geochemistry here to acidification is the only comparison are those samples grown under natural conditions?
Answer:
As written in the manuscript, to assess the change in microstructure and geochemistry, we compared both the differences between parts produced before-culturing and during-culturing, as well as the differences between low-pH treated specimens and control specimens. For specimens (#43 and #63), a control specimen was not available so we were not confident in interpreting the results we obtained and we preferred to underline it in the manuscript

14) It appears that the experimental treatments here are similar despite the pH used (Figure 9). I would question the relevance of this section, perhaps omit or justify how this is comparable.
Answer:
We think that these data are very important to be represented as they show the results of the different methods used. Also this diagram answers to reviewer question 13 and it shows that specimen cultured under low–pH conditions had smaller fibres when

compared to that of control. It also shows the differences in fibre size among different subzones of the same specimens. In conclusion, this section clearly syntheses and displays the numerous measurements taken in this this study, so it is very important for the reader.

15) Section 3.4 Stable isotopes Nice figure 10, it is clear to see trends between three specimens for significantly lighter stable carbon isotopes with experimental low-pH treatment compared to natural versus control treatments. Did the authors compare these data statistically?
Answer:
We thank the reviewer for this suggestion. We have done additional $t$-tests to compare the data from different pH treatments, and added them in the revised manuscript (Supplementary table 2).

16) Discussion Page 28, 'electron back scattering diffraction' should be electron backscatter diffraction.
Answer:
We have corrected it in the revised manuscript
Page 28 Line 9
Analyses of electron back scattering diffraction
changed to
Analyses of electron backscatter diffraction

17) Line 20, 'May this indicate a greater amount of organic components in this part of the shell?' is this what the authors suggest? Please rephrase not as a question but a statement with references or omit.
Answer:
We have deleted this sentence in the revised manuscript, because there is no conclusive evidence about it in the literatures.

18) Lines 28, 'living organism' this should be living organisms.
Answer:
We have corrected it in the revised manuscript
Page 28 Lines 28:
living organism
Changed to
living organisms

19) Page 34-35. The discussion of the depleted $^{18}O$, $^{13}C$ is related to changes in percentage, can the authors present the statistical significance here of the changing isotope values?
Answer:
We thank the reviewer for this suggestion. We have done additional $t$-tests to compare the data from different pH treatments, and added them in the revised manuscript

(Supplementary table 2).

20) The authors state that there is individual specimen variability, does this remove the significance of the low pH treatment over the isotope depleted values? Or are the authors suggesting here that there are insufficient specimen numbers to make significant statements relating to the isotope data? Page 35, line 11 'More measurements are however needed to fully answer this.'?
Answer:
We think that analyses on more specimens could help to understand in greater details the $d^{13}C$ variability, but this does not undermine that we have robust data to support the fact that brachiopod produce shells near brachiopod equilibrium even in changing external conditions. We have results from both dorsal and ventral valves, if there was uncertainty it might show up in one but hardly both, the matching and concurrent isotope results speak clearly to the robustness of the values and trends.

21) Page 35, line 16 'Thus, we think', perhaps the data suggest?
Answer:
We have changed it in the revised manuscript
Thus, we think that large part of the secondary layer isotope record
changed to
Thus, the data suggest that large part of the secondary layer isotope record

22) The authors end in the statement 'secondary layer isotope record may reflect the environmental conditions supporting the interpretation of brachiopod shells as good archives of geochemical proxies, even when stressed by ocean acidification.'. This is also stated in the abstract as one of the main implications of this study. Following the current discussion on page 34-35 I would question whether the authors can make this statement, and whether there is sufficient evidence to support this, although Figure 10 does suggest this is the case. Please directly refer to the data here; are there sufficient samples, what is the n-number? This will enable the reader to determine if the manuscripts data do support these conclusions. If this data is not available then the authors will need to remove this emphasis from the abstract and conclusion statements.
Answer:
We think that our data are robust because we have analysed 9 specimens, 6 specimens for microstructure analyses, 5 for isotope geochemistry. We measured the size and shape of 540 fibres plus 1392 fibre at the anterior margin, we selected and measured 388 sub-zones for boundary calculations; we took 170 measurements for primary layer thickness; we selected and measured 29 zones for endopunctae density, 227 for diameter of endopunctae; we analysed 79 samples for isotope geochemistry. Following the suggestions of the reviewer, we have added these numbers in the conclusions.
Page 35 lines 19-20:
This study combined the analysis of shell microstructure and stable isotope

geochemistry on brachiopods cultured at low pH conditions for different time intervals, and suggests the following conclusions.

changed to

This study combines the analysis of shell microstructures on 6 specimens consisting of 1932 fibre size measurements, 170 primary layer thickness measurements, 256 punctal density and diameter measurements and stable isotope geochemistry on 5 specimens of 79 sample analyses, on brachiopods cultured at low pH conditions for different time intervals. The results suggest the following conclusions.

23) Page 35 conclusions 'This was related to the source of carbon dioxide gas used in the culture setup', could this not be due to a change in the carbonate compositions as a result of adding $CO_2$ impacting the DIC? Did you test the carbon isotopes of the gas?
Answer:
We did not test the $d^{13}C$ of the gas, but we did measure the $d^{13}C$ in water, as we written in the manuscript Page 34 Line 31: the $\delta^{13}C_{DIC}$ in the water during the cultivation process of our specimens was low ($\delta^{13}C$ VPDB: -23.63 ‰ for the low pH conditions and -2.03 ‰ for the control conditions, which corresponds to the $pH_2$ phase), This is essentially the $d^{13}C$ of the DIC which comes from the source so it is pretty much almost the same.

24) I have seen this drop in carbon isotopes in the natural seawater samples where increasing $CO_2$ from run-off caused a lighter carbon isotope value. The authors should expand this discussion to the previous paragraph.
Answer:
As the main goal of the paper is to understand the impact of acidification on cultured specimens, we did not expand the discussion on what happened before culturing in the natural environment. However, this was the object of a paper by Romanin et al. (2018). In fact, as we have already written in manuscript at Page 34 Line 14, Romanin et al. (2018), who also analysed specimens collected from Comau Fjord, attributed the negative isotope excursion to environmental perturbations, in particular, to changes in seawater productivity and temperature, and/or to anthropogenic activities. Here, we follow the interpretation of Romanin et al. (2018) to explain the mid–shell excursion observed in our specimens.

Comments from E. Cross

e.l.cross@cantab.net

Firstly, it is great to read that other researchers are using living specimens of this highly calcium-carbonate-dependent group to address outstanding questions of biological responses to ocean acidification. I look forward to future publications from this lab. I have a few minor comments in relation to correctly citing previous research on ocean acidification impacts on brachiopods:

1) Page 2, Line 21-23: Please add to this sentence about previous findings of ocean acidification impacts on brachiopods that Cross et al. (2018) also found that punctae have become thinner over the last 120 years, which partially explained the increase in shell density over this time period.

Answer:

We have added this sentence in the revised manuscript.

Page 2, Line 21-23:

The few studies that examined brachiopods or brachiopod shells suggest that the latter suffered increased dissolution under lower seawater pH conditions, whereas the organism either exhibited no changes, or an increase in shell density [calculated as dry mass of the shell (g)/shell volume ($cm^3$)], but otherwise no changes in shell morphology and trace chemistry (Table 1).

changed to

The few studies that examined brachiopods or brachiopod shells suggest that the latter suffered increased dissolution under lower seawater pH. In other studies, the organism either exhibited no changes or an increase in shell density [calculated as dry mass of the shell (g)/shell volume (cm3)], but otherwise no changes in shell morphology and trace chemistry (Table 1). Cross et al. (2018) found that punctae became narrower over the past 120 years, which partially explained the increase in shell density over this period.

2) Page 2, Table 1: Please specify that shell growth rates and shell repair frequencies of *Calloria inconspicua* were not affected by low pH in the row related to Cross et al. (2016) in the column stating "not affected by lower pH".

Answer:

We have indicated it in the revised manuscript.

Page 2, Table 1 column 3 row 2:

not affected by lower pH

changed to

shell growth rates and shell repair frequencies were not affected by low pH

3) Page 3, Table 1: Please correct the number of specimens used in the row related to Cross et al. (2018). 389 adult specimens were used in the shell morphology analysis. A subsample of 40 brachiopods (2-5 specimens per decade over the last 120 years) were used for further shell analysis on shell density, punctal width, punctal density, shell dissolution, shell thickness and shell elemental composition.

Answer:

We have corrected it in the revised manuscript.
Page 2, Table 1 column 1 row 3:
N = 389 (adults)
changed to
N = 389 (adults) for shell morphology analyses*.
And we have added the note below the table:
*A subsample of 40 brachiopods (2-5 specimens per decade over the last 120 years) were used for further shell analysis of shell density, punctal width, punctal density, shell dissolution, shell thickness and shell elemental composition.

4) Please also add that no changes were found in shell dissolution over the last 120 years.
Answer:
We have added it in the revised manuscript.
Page 3, Table 1 column 3 row 3:
add:
no changes were found in shell dissolution over the last 120 years.

5) Page 3, Table 1: Please specify that shell growth rates and shell repair frequencies of *Liothyrella uva* were not affected by low pH and temperature in the row related to Cross et al. (2015) in the column stating "not affected by lower pH".
Answer:
We have specified it in the revised manuscript.
Page 3, Table 1 column 2 row 4:
not affected by lower pH
changed to
not affected by lower pH and temperature
Page 3, Table 1 column 3 row 4:
not affected by either low pH conditions or temperature
changed to
shell repair frequencies were not affected by low pH and temperature

6) Page 31, Line 9: To avoid confusion, please add in that these specimens are from the same locality in New Zealand (Paterson Inlet, Stewart Island, New Zealand).
Page 31, Line 10: Please also add that the pH decrease by 0.1 pH units occurred over the last two decades whilst the 2 ℃ increase in temperature occurred over the last 60 years.
Answer:
We have added it in the revised manuscript.
Page 31, Line 9:
One response, however, appears to reinforce the shells of C. inconspicua by laying down a denser shell compared to specimens from New Zealand over the last 120 years while subjected to a slight decrease in pH (by 0.1) and 2 ℃ increase in temperature over the last two decades (Cross et al., 2018).

changed to

However, C. inconspicua from the same locality in New Zealand (Paterson Inlet, Stewart Island) laid down a denser shell over the last 120 years, with nearby environmental conditions increased by 0.6 ℃ from 1953 to 2016, and slightly increased by 35.7 μatm in pCO2 from 1998 to 2016 (Cross et al., 2018). These changes are in line with global trends of ocean pH and temperature since the industrial revolution (Caldeira and Wickett, 2005; Orr et al., 2005; IPCC, 2013).

7) Page 32, Line 5-8: Majority of the studies listed here did not investigate brachiopod growth rates. To avoid confusion and strengthen the authors point that there is a limited database on ocean acidification impacts on brachiopods, only include studies here on brachiopods.
Answer:
As we mainly discussed the effects of acidification on the growth rates of marine calcifiers, we have corrected the word 'brachiopod' to 'marine calcifiers' in the revised manuscript.
Page 32, Line 5-8:

[revised manuscript text omitted]